# Specialized RNA decay fine-tunes monogenic antigen expression in *Trypanosoma brucei*

Lianne I. M. Lansink [1,2], Htay Mon Aye[1,2], Leon Walther [1,2,3,4],
Sophie Longmore [1,2], Madeleine Jones [1,2], Adam Dowle [1,2],
João L. Reis-Cunha[1,2] & Joana R. C. Faria [1,2] ✉

Antigenic variation is an immune evasion strategy used by pathogens, including *Trypanosoma brucei*. This parasite expresses a single variant surface glycoprotein (VSG) from a large genetic repertoire, which it periodically switches throughout an infection. *VSGs* are co-transcribed with expression-site-associated genes (*ESAGs*) within a specialized nuclear body, but there is substantial differential expression and the regulatory mechanisms remain unclear. Here we applied TurboID-mediated proximity labelling mass spectrometry to map the subnuclear expression-site body (ESB) post-transcriptional network. We identify and characterize three previously undescribed components: ESB-associated protein 1 (ESAP1) and ESB-specific proteins 2 and 3 (ESB2 and 3). These proteins form discreet subnuclear condensates that are developmentally regulated. ESB2 is an active RNA endonuclease that negatively regulates *ESAG* transcripts. Its recruitment depends on a hierarchy involving VEX2, ESAP1 and ESB3, a constant flux of active transcription and RNA processing, and its own nuclease activity. Overall, we uncover a molecular mechanism that fine-tunes expression of virulence genes through specialized RNA decay in *T. brucei*.

The molecular mechanisms regulating expression of a single gene out of a large gene family remain a major mystery in eukaryotic biology. This phenomenon drives the fascinating 'arms race' between host immunoglobulins and pathogen surface antigens, including those of malaria- and sleeping-sickness-causing parasites[1]. In mammalian hosts, the African trypanosome replicates freely in the bloodstream and extravascular spaces, fully exposed to the immune system, relying on the stochastic switching of a variant surface glycoprotein (VSG) for immune evasion[2]. Remarkably, one antigen is expressed in a monogenic fashion from >2,000 genes, forming a protective coat comprising 5–10% of total cellular mRNA and protein[1,3].

*VSG* genes are exclusively expressed from 1 of ~15 sub-telomeric, polycistronic *VSG* expression sites (*VSG*-ESs), transcribed by RNA polymerase-I (Pol-I) within a subnuclear expression-site body (ESB), while the remaining sites are transcriptionally repressed[4,5]. Each *VSG*-ES also contains several expression-site-associated genes (*ESAGs*); some

of their products are surface exposed and involved in host–parasite interactions, including protecting the parasite from human serum lytic effects, modulating the host's innate immune response, uptake of essential nutrients and contribution to antigenic variation[6–8]. Numerous factors have been implicated in *VSG* gene silencing[9,10] (reviewed in refs. [1,11]); the ESB, however, has remained largely mysterious for over 20 years. Only two ESB-specific components have been identified to date: ESB-specific protein 1 (ESB1), a transcriptional activator, and VSG-exclusion protein 2 (VEX2), which prevents activation of silent *VSG*-ESs[11,12].

Beyond allelic exclusion, VEX2 enhances *VSG* mRNA processing by bridging the active-*VSG*-ES to one of the *Spliced Leader* (*SL*) arrays via VEX1 (refs. [13–15]). *SL* arrays encode for the *SL*-RNA required for *trans*-splicing, a process that adds a common sequence to each pre-mRNA and is coupled to polyadenylation in kinetoplastids[16]. Indeed, high *VSG* expression results from transcription by Pol-I and

[1]Biology Department, University of York, York, UK. [2]York Biomedical Research Institute, University of York, York, UK. [3]Chemistry Department, University of York, York, UK. [4]York Structural Biology Laboratory, University of York, York, UK. ✉e-mail: joana.correiafaria@york.ac.uk

post-transcriptional mechanisms. Spatially, the ESB clusters with RNA processing bodies: an *SL*-array-associated body (SLAB), a nuclear FMR1 interacting protein 1 (NUFIP) body and a 'Cajal-like' body[13,17], assembling a 'nuclear expression factory'. Simultaneously, the *VSG* mRNA is stabilized by a conserved 3′ untranslated region (UTR) '16-mer' motif, which recruits CFB2, a cyclin-like F-box protein, and promotes a protective N6-methyladenosine (m6A) modification of the poly(A) tail[18,19]. It remains a mystery how co-transcribed *ESAGs* are maintained at levels >140-fold lower than the active *VSG*, implying a robust, unknown post-transcriptional filter.

Here we applied TurboID-mediated proximity labelling mass spectrometry (PL–MS) to map the ESB post-transcriptional network, identifying three new components: ESB-associated protein 1 (ESAP1) and ESB-specific proteins 2 and 3 (ESB2 and 3). We characterize ESB2 as an RNA endonuclease that negatively regulates *ESAG* transcripts. Crucially, we show that ESB2 recruitment depends on both its own catalytic activity and a hierarchy involving VEX2, ESAP1 and ESB3. Overall, we unravelled a molecular mechanism that fine-tunes expression of virulence genes through specialized RNA decay in *Trypanosoma brucei*.

## VEX PL–MS identifies new components of the ESB and surrounding nuclear bodies

Based on previous studies, we hypothesized that the ESB is likely to have separately assembled 'transcriptional and processing subdomains', which are ESB1 and VEX-dependent, respectively[11,13,14]. To identify the protein network within the ESB 'processing subdomain' and at the interface with the SLAB, we performed PL–MS using VEX2 and VEX1 as baits, markers for the ESB and the SLAB, respectively (Fig. 1a). We fused the proteins of interest with TurboID, which labels proteins in living cells[20]. In the case of VEX2, given its large size, we generated fusions at both termini to increase spatial resolution. We confirmed that all protein fusions showed the predicted molecular weight (MW) and localized to the expected subnuclear compartments (Fig. 1b and Extended Data Fig. 1a–e). Following the addition of exogenous biotin, cells expressing VEX2 N- and C-TurboID showed a primary focus of biotinylation at the ESB but also notable labelling within the nucleolus (Fig. 1c), probably indicative of shared molecular components between them, such as Pol-I. VEX1-TurboID localized to one or both SLABs, with one frequently positioned adjacently to the ESB, as expected[13,14]. The diffuse biotinylation signal near the ESB partially extended into this compartment (Fig. 1d and Extended Data Fig. 1d–f), consistent with VEX1 interacting with distal active-VSG-ES chromatin despite its primary *SL*-array association[12,13]. Subnuclear biotinylation was observed within 30 min of biotin supplementation and maintained across longer time

points (18 h; Fig. 1c,d and Extended Data Fig. 1f,g). Biotinylated material was then affinity purified and submitted to MS analysis (Fig. 1e–g, Extended Data Fig. 1h–l and Supplementary Data Sheets 1–3).

We integrated the three TurboID datasets (VEX1, VEX2 N- and C-terminal) and identified 70 significantly enriched proteins in comparison with the parental control ($\log_2$(fold change (FC)) > 1.5; adjusted (adj) $P < 0.05$; false discovery rate (FDR) 1%; Fig. 1e–h and Supplementary Data Sheets 1–3). As expected, the baits and known interactors, including the histone chaperone chromatin assembly factor (CAF)-1a–c, were significantly enriched ($4.34 \times 10^{-9} <$ adj $P < 1.09 \times 10^{-3}$; $1.79 < \log_2$(FC) < 6.90) in the TurboID 'plus biotin' samples in all three datasets (Fig. 1e–g). In addition, Pol-I subunits, the Pol-I elongator TDP1 (ref. 21), all known telomere binding proteins[9,22] and NUFIP body components[17] were also enriched in the TurboID 'plus biotin' samples (Extended Data Fig. 2a–c). We also detected SIZ1 in proximity to VEX2 (N-TurboID $\log_2$(FC) 2.55; C-TurboID $\log_2$(FC) 2.79), a known small ubiquitin-like modifier (SUMO) E3 ligase, consistently with the ESB lying within a highly SUMOylated focus[23] (Extended Data Fig. 2b,c). The 70 significantly enriched proteins were subsequently refined to a set of 45 candidates by excluding known ESB, SLAB and NUFIP components and telomere-binding and non-nuclear proteins[24,25] (Fig. 1h,i). Gene ontology (GO) enrichment analysis revealed that 12 of those 45 protein candidates were probably involved in RNA processing or metabolism (Extended Data Fig. 2d,e and Supplementary Data Sheet 4). These were further refined to a final set of 19 prioritized candidates based on adherence to at least one of the criteria designed to identify proteins crucial for ESB-related activities (detailed in Methods; Fig. 1i,j). Among these, eight were proteins of unknown function, which, at first, we designated VEX Proximity Labelling X, V2PLx (Supplementary Data Sheet 5). We endogenously tagged 17 out of the 19 'high-priority hits', as 3 belonged to the same protein complex, and assessed their localization using fluorescence microscopy.

We identified three new ESB components, which we designated ESB-specific proteins 2 and 3 (ESB2, Tb927.1.1820, V2PLO2; ESB3, Tb927.9.15850, V2PL01) and ESB-associated protein 1 (ESAP1, Tb927.10.11600, V2PL04) (Figs. 1k,l and 2). We found four proteins that were accumulated at the ESB, albeit not exclusively: CPSF3, a component of the cleavage and polyadenylation specificity factor[26]; BRCA2, a protein involved in homologous recombination-mediated DNA repair[27]; Tb927.10.12030 (V2PL05), a protein of unknown function; and BDF2, a bromodomain protein involved in epigenetic regulation, previously shown to interact with the chromatin at the active-*VSG*-ES[28] (Extended Data Fig. 3a–i). Putative components of other assemblies within the cleavage and polyadenylation complex[29]

**Fig. 1 | VEX1 and VEX2 PL–MS identified novel ESB-specific, SLAB and NUFIP body components. a**, Schematic of PL–MS. VEX1 and VEX2, fused to the biotin ligase TurboID, serve as markers for the SLAB and ESB, respectively. Proximal proteins were biotinylated, affinity purified using streptavidin and identified via MS. MS analysis showed shared enrichment of known SLAB, ESB and NUFIP components in both VEX1 and VEX2 samples, suggesting dynamic interplay between these nuclear bodies. Protein candidates from combined VEX datasets were selected and prioritized as detailed in Methods and subsequently endogenously tagged to confirm their localization via colocalization fluorescence microscopy with known protein markers. **b–d**, Fluorescence microscopy analysis of [3myc]TurboID-VEX2/Pol-I (**b**), [3myc]TurboID-VEX2/ biotinylation (**c**), and biotinylation/Pol-I in a VEX1-TurboID[3myc] cell line (**d**). Biotin was applied at 50 μM for 30 min; biotinylated material was detected using Streptavidin-A488. Images are representative of at least two independent experiments, acquired using a Zeiss LSM980 Airyscan 2 and correspond to 3D projections by brightest intensity of 0.1-μm stacks. No, nucleolus. DNA was stained with DAPI (cyan). **e–g**, Radial plots representing the proximity proteomes of [3myc]TurboID-VEX2 (**e**, $n = 6$), VEX2-TurboID[3myc] (**f**, $n = 4$) and VEX1-TurboID[3myc] (**g**, $n = 4$). $\log_2$(FC), which increases clockwise, represents the fold change in protein abundance between the cell lines in which VEX1 and VEX2 were fused with TurboID and the parental line, both treated with 50 μM of biotin for 18 h.

The $y$ axis ($-\log_{10}(P)$) represents statistical significance, which increases towards the centre. Significance was assessed via two-sided moderated $t$-tests with Benjamini–Hochberg FDR correction (limma). VEX1, VEX2 and CAF-1a–c are known interactors, highlighted in cyan, burgundy and yellow, respectively. Datapoints highlighted in lilac represent proteins that are significantly enriched in the TurboID 'plus biotin' samples ($P < 0.05$, FDR 1%). WT, wild type. **h–j**, Venn diagrams representing the overlap between VEX1-TurboID[3myc], [3myc]TurboID-VEX2 and VEX2-TurboID[3myc] datasets. Proteins enriched in TurboID 'plus biotin' samples over parental controls ($\log_2$(FC) > 1.5 and $P < 0.05$) are depicted (**h**). Hits (**i**) were selected by subtracting known nuclear body components and clear contaminants and then prioritized (**j**) based on several parameters (detailed in Methods). **k**, Heatmap depicting the $\log_2$(FC) for the newly identified components of the ESB, SLAB and NUFIP body in the [3myc]TurboID-VEX2 and VEX2-TurboID[3myc] datasets (versus the parental line control). **l**, Schematic summarizing the main nuclear compartments in *T. brucei* bloodstream forms. Known components of each nuclear body (ESB, NUFIP, SLAB or Cajal-like) are in grey; new components identified in this study are in salmon. Those are also represented in salmon in the radial plots in **e–g**. Schematics created in BioRender: **a**, Faria, J. https://BioRender.com/83ovip3 (2026); **l**, Faria, J. https://BioRender.com/kpyp9lj (2026).

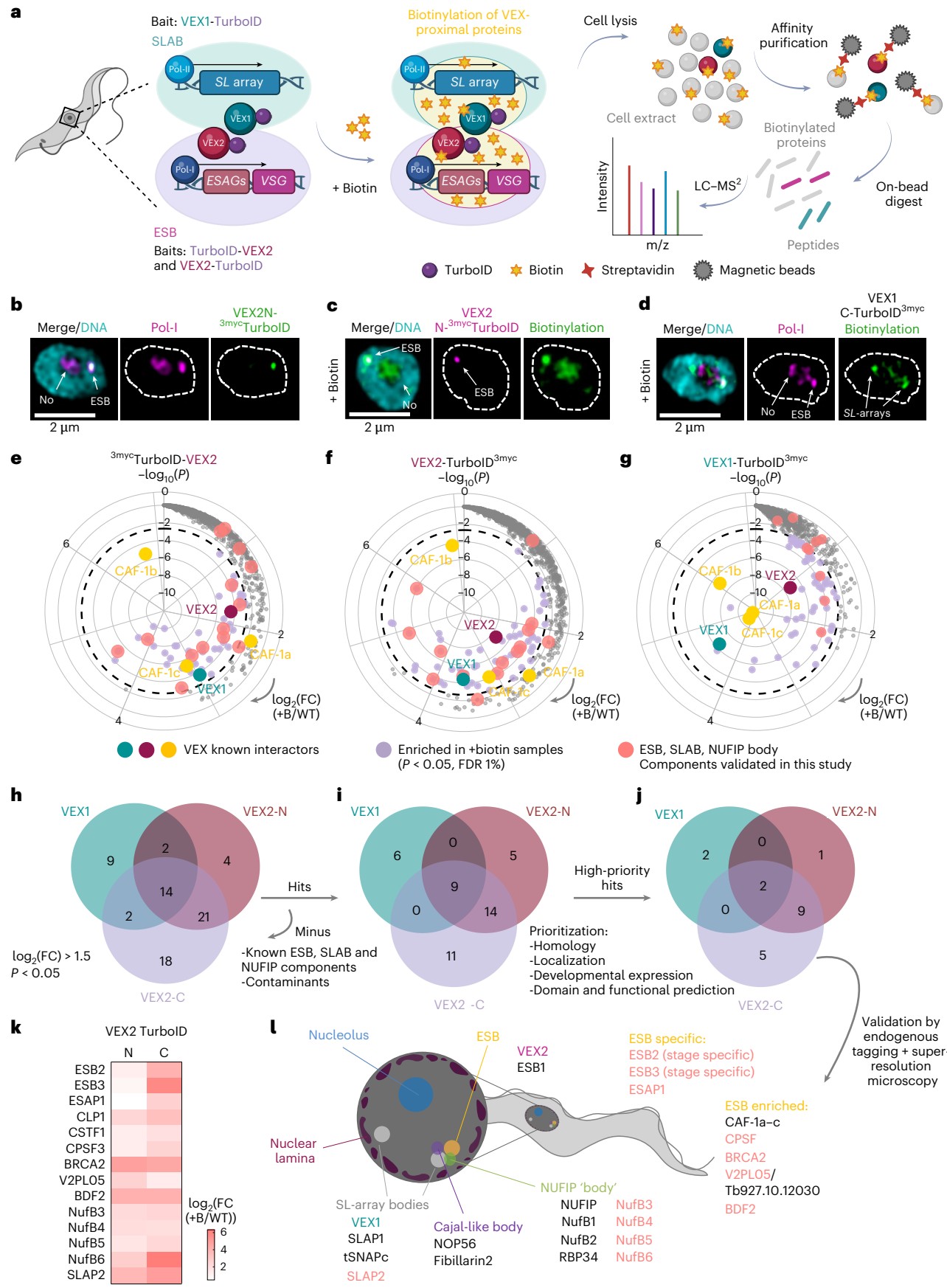

were also significantly enriched in the VEX proximity proteomes: CLP1 (Tb927.11.9760) and CstF1 (Tb927.10.15740) (Fig. 1k). Similarly, most of the BDF2 known interactors, another bromodomain containing protein, BDF6, and its known interactors, EAF6 (component of the NuA4 histone acetyltransferase complex) and Tb927.10.14190 (V2PL03)[30] were enriched in the TurboID 'plus biotin' samples (Extended Data Fig. 3h). Unlike BDF2, BDF6, EAF6 and V2PL03 localized to the nucleoplasm with a speckle-like profile without notable enrichment at the ESB (Extended Data Fig. 3h–j).

In addition, we identified four new NUFIP body components (NufB3–6), previously identified as KKT (kinetochore components) interacting proteins (KKIP 2, 3, 4 and 11)[31] and a new SLAB component, *SL*-associated protein 2 (SLAP2, Tb927.4.3360 (V2PL07); Extended Data Fig. 4).

Notably, reciprocal enrichment of ESB and SLAB components via VEX1 and VEX2 PL–MS supports the proposed VEX-mediated interchromosomal bridge linking these compartments[13,14]. Overall, we identified new SLAB, NUFIP and ESB factors (Fig. 1e–g,l) and then focused our functional studies on the ESB-specific ones.

## ESB2, ESB3 and ESAP1 are developmentally regulated

ESB2, ESB3 and ESAP1 form a single discrete protein condensate in G1 cells in bloodstream forms (BSFs), which colocalizes with the ESB in >95% of the nuclei (Fig. 2a–c), as confirmed by high-resolution imaging (Fig. 2d–f). We further validated exclusive ESB localization using double allele tagged lines, which retained identical subnuclear focal distribution (Fig. 2g, ESB2 depicted). Structural analysis revealed a C-terminal PilT N-terminus (PIN) domain in ESB2 (Fig. 2d), a motif typically associated with single-stranded RNA cleavage[32]. ESAP1 contains an RNA recognition motif (RRM)-like domain and C-terminal arginine–serine (RS) repeats (Fig. 2f), common in splicing regulators[33], whereas ESB3 lacks identifiable domains (Fig. 2e).

Next, we assessed the expression and localization of ESB2, ESB3 and ESAP1 in procyclic forms (PCFs, insect stage) of *T. brucei* (Fig. 2h–j and Extended Data Fig. 5), where the ESB is absent and *VSG* expression is repressed. Compared with PCFs, BSFs showed modest mRNA upregulation of ESB1 (~2.5-fold), ESB2 (~1.4-fold) and ESB3 (~3.5-fold), while VEX1, VEX2 and ESAP1 levels remained stable (Extended Data Fig. 5b). At the protein level, ESB2 and ESB3 were undetectable in PCFs, whereas ESAP1 was downregulated and redistributed as nucleoplasmic speckle (Fig. 2h–j and Extended Data Fig. 5a), supported by previous comparative transcriptomics and proteomics studies[34,35] (Extended Data Fig. 5c,d). Furthermore, RNA interference (RNAi) depletion of ESB2 and ESAP1 in PCFs yielded no fitness cost or transcriptomic changes (Extended Data Fig. 5e–g and Supplementary Data Sheet 6). While expected for the unexpressed ESB2, the function of ESAP1 in PCFs remains unclear.

Altogether, ESB2 and ESB3 are stage-specific like ESB1, while ESAP1 is expressed in both developmental stages but ESB-associated only in BSFs, similarly to VEX2. These attributes are reflected in their nomenclature. Notably, developmental regulation of all known ESB components appears to occur primarily at the protein level (Fig. 2k).

## ESB2, ESB3 and ESAP1 depletion leads to upregulation of *ESAGs* in BSFs

To investigate their role in the mammalian stage, we generated tetracycline-inducible RNAi cell lines for ESB2, ESB3 and ESAP1 and observed a pronounced fitness cost following induction in all three knockdowns, particularly for ESB2 (Fig. 3a). Successful protein depletion was confirmed by protein blotting, which also revealed that ESB2 migrates predominantly as a higher-MW species, distinct from its predicted size (~60 kDa), suggesting post-translational modification (Fig. 3b). Induced and non-induced cells homogeneously expressed VSG-2 (active-VSG in our strain) (Fig. 3c).

We then conducted RNA sequencing (RNA-seq) analysis at 24 h post-induction (Fig. 3d–k, Extended Data Fig. 6 and Supplementary Data Sheet 7). ESB2, ESB3 and ESAP1 depletion when compared with the parental cell line (5.5-, 20.5- and 3.4-fold decrease; $P = 7.79 \times 10^{-24}$, $P = 6.62 \times 10^{-94}$ and $P = 1.56 \times 10^{-49}$, respectively) led to a specific upregulation of all *ESAG* transcripts from the active *VSG*-ES (up to 11-fold, $1.04 \times 10^{-269} < P < 6.90 \times 10^{-3}$), accompanied by a modest reduction (1.1–1.2-fold decrease, $0.0002 < P < 0.1839$) in the active-*VSG* transcript. In addition, we observed a significant increase in some *ESAG* transcripts originating from 'silent' *VSG*-ESs, without any major changes in the mRNA levels of the corresponding *VSGs*.

Overall, ESB2, ESB3 and ESAP1 somehow negatively regulate *ESAGs*. The uncoupling of this upregulation from unchanged *VSG* levels argues against increased transcription initiation, pointing instead to specific post-transcriptional regulation, which we later explore. Furthermore, the remarkably similar transcriptional profiles (Fig. 3g–k) suggested a shared pathway, prompting us to investigate co-dependencies between these and other ESB factors.

## ESB2 recruitment hierarchically depends on VEX2, ESAP1 and ESB3

Firstly, we assessed how ESB2, ESB3 and ESAP1 reciprocally impacted their relative mRNA and protein abundances as well as localization to the ESB (Fig. 4a–c and Extended Data Fig. 7a–e). ESAP1 localization was not affected by ESB2 or ESB3 knockdown, neither was its protein abundance (Fig. 4b,c and Extended Data Fig. 7a,c). Next, ESB3 localization was shown to be ESAP1 dependent, with a reduction of ~66.7% in the percentage of nuclei where an ESB3 focus could be detected following ESAP1 RNAi, but without significant changes in expression. Furthermore, ESB3 localization was ESB2 independent, but its protein levels were significantly reduced (Fig. 4b,c, and Extended Data Fig. 7a,d). Lastly, we observed that ESB2 localization depended on both ESAP1 and ESB3, as following either knockdown, its foci could no longer be detected in ~82.5% or ~73.5% of nuclei, respectively. In addition, in the absence of ESB3, ESB2 protein levels were

---

**Fig. 2 | ESB2, ESB3 and ESAP1 are developmentally regulated.**
**a,d,e,f,** Fluorescence microscopy analysis of ESB2[12myc] (**a,d**), [6myc]ESB3 (**a,e**), [6myc]ESAP1 (**a,f**) and Pol-I RPA1[10ty] in bloodstream forms. **d–f,** Histograms on the right depict the fluorescence signal distribution across the distance highlighted by the yellow lines. A schematic of the number of amino acids, identifiable domains and sequence motifs is provided for each protein. PIN, RNA nuclease; RRM-like, RNA binding. **b,c,** Percentage of G1 nuclei with 1, 2 or >2 ESB2, ESB3 and ESAP1 foci (**b**) and percentage of G1 nuclei where ESB2, ESB3 and ESAP1 are overlapping, immediately adjacent or separate from the ESB (**c**). The graphs depict mean values of two biological replicates; >100 G1 cells per condition were analysed. For more details, see Microscopy and Image Analysis. **g,** Fluorescence microscopy analysis of [GFP-myc]ESB2 localization throughout the cell cycle. N, nucleus; K, kinetoplast. **h,i,** Fluorescence microscopy analysis of ESB2[12myc] (**h**), [6myc]ESB3 (**h**) and [6myc]ESAP1 (**i**) in procyclic forms. **j,** Protein blotting analysis of ESB2[12myc], [6myc]ESB3 and [6myc]ESAP1 expression in bloodstream (B) and procyclic (P) forms. EF1α was used as a loading control. The images in **d–f** and **i**, and **a**, **g** and **h**, were acquired using a Zeiss LSM980 Airyscan 2 and a Zeiss AxioObserver, respectively, and correspond to 3D projections by brightest intensity of 0.1-μm stacks. DNA was stained with DAPI (cyan, grey or purple). All cell lines were single allele tagged with the exception of **g** where ESB2 was double allele tagged. In **d–f** and **h–j**, the data are representative of two independent experiments. **k,** Diagram summarizing how VEX2, ESAP1, ESB3 and ESB2 are developmentally regulated between insect (cyan) and mammalian (purple) stages of *T. brucei* at both RNA and protein levels. In the insect stage, VEX2 and ESAP1 can be found throughout the nucleus in a speckle-like manner. All proteins form a single protein focus that colocalizes with the ESB in the mammalian form of the parasite. Diagram in **k** created in BioRender; Faria, J. https://BioRender.com/zevyuz1 (2026).

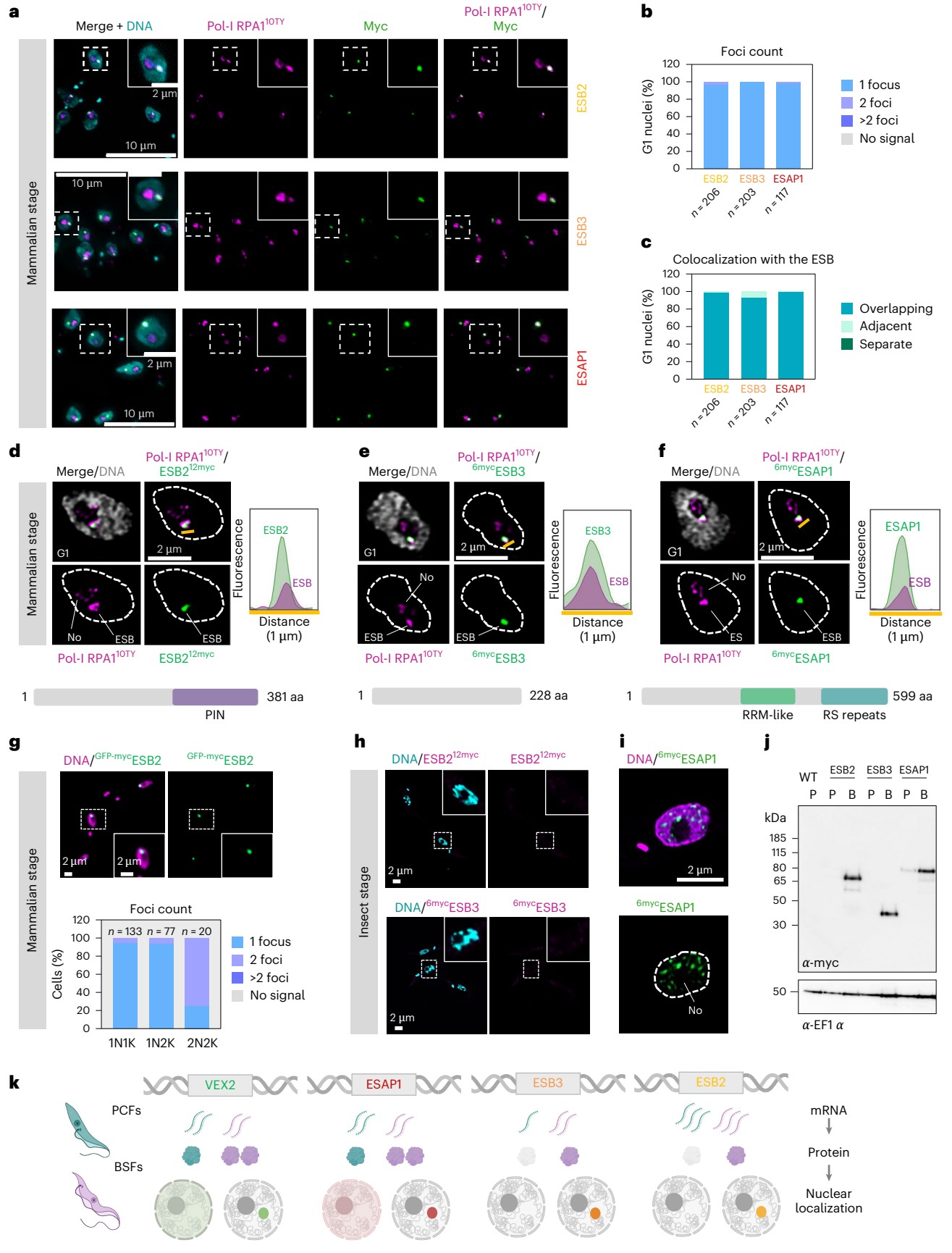

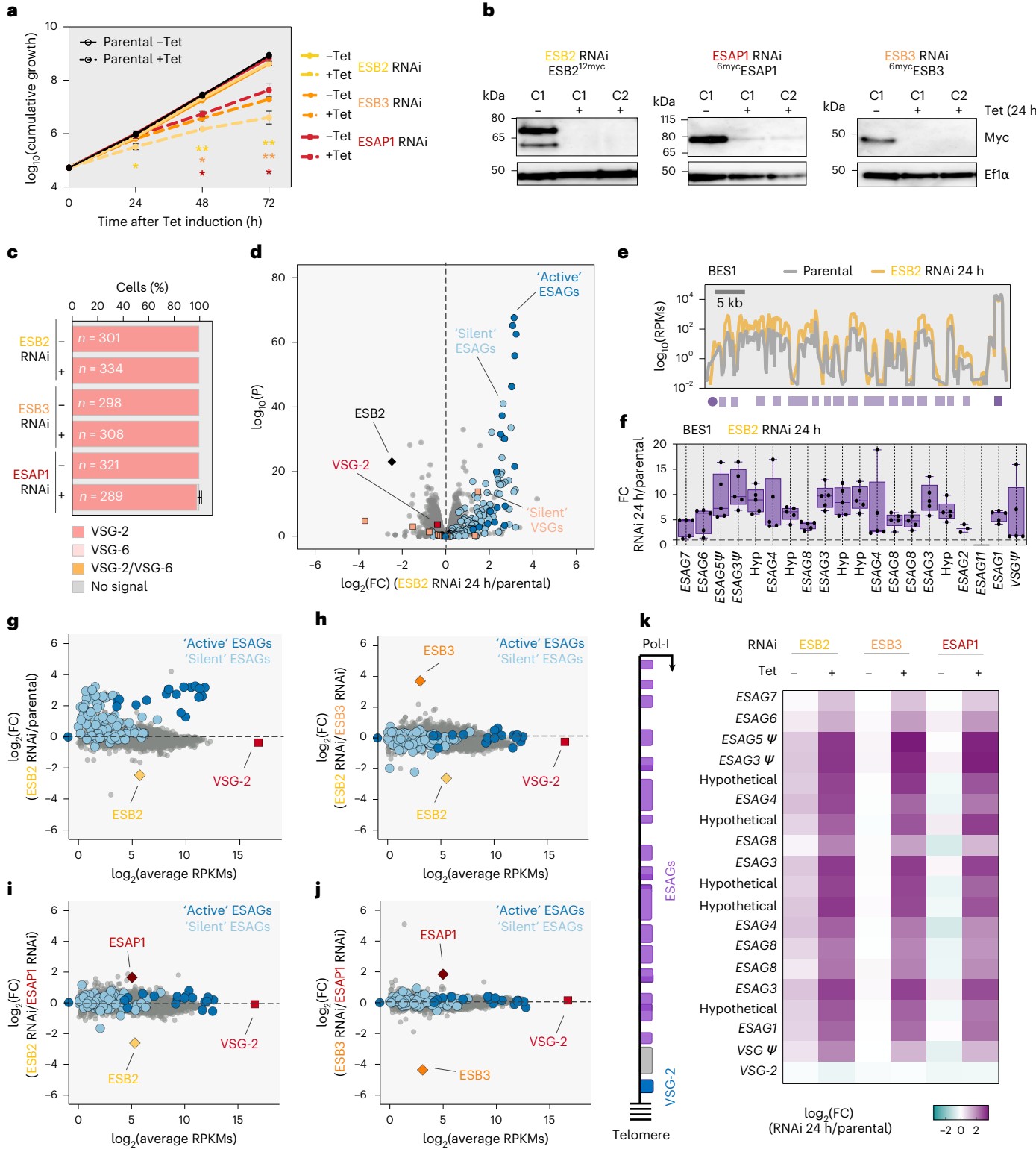

significantly reduced ($P < 0.001$; Fig. 4a–c and Extended Data Fig. 7e); transcriptomic data showed no significant changes at the mRNA level (Extended Data Fig. 7b).

Secondly, we investigated whether the ESB integrity was impacted by ESB2, ESB3 and/or ESAP1 depletion; we used VEX2 and ESB1 knockdowns as controls (Fig. 4d,e and Extended Data Fig. 7f). ESB1 knockdown caused mild ESB loss (~22%), whereas VEX2 depletion resulted in >90% of cells expressing at least two VSGs and severely disrupted Pol-I

localization by reducing detectable ESBs to <30%, consistent with previous reports[11,12]. By contrast, ESB2, ESB3 and ESAP1 depletion did not significantly impact ESB integrity (Fig. 4e and Extended Data Fig. 7f).

Thirdly, as (1) VEX2 has a major role in maintaining, possibly establishing, the ESB structure and (2) ESB2, ESB3 and ESAP1 were found in proximity to VEX2, we next asked whether their localization was VEX2 dependent. VEX2 knockdown led to a pronounced nuclear dispersion of ESB2, ESB3 and ESAP1 in >64% of nuclei (Fig. 4f,g and

**Fig. 3 | ESB2, ESB3 and ESAP1 knockdown leads to upregulation of *ESAG* transcripts in BSFs. a**, Cumulative growth following tetracycline-induced ESB2 (yellow), ESB3 (orange) and ESAP1 (red) knockdown. Regarding the statistical analysis, the colours indicate which cell line they refer to and correspond to induced versus non-induced conditions for each time point; significance was determined using two-tailed paired *t*-tests with multiple-comparison correction using the Holm–Šídák method; *$P < 0.05$; **$P < 0.01$; **$P < 0.001$. **b**, Protein blotting analysis of ESB2, ESB3 and ESAP1 expression following their respective knockdowns. C1, clone 1; C2, clone 2; EF1α, loading control. **c**, Immunofluorescence analysis of VSG expression following ESB2, ESB3 and ESAP1 knockdown (48 h); >100 cells per condition were analysed. – and + indicate the absence and presence of tetracycline, respectively. **a,c**, Data correspond to mean values of three biological replicates ± s.d. (not visible in **c**) and representative of independent experiments. **d–k**, RNA-seq analysis of ESB2 ($n = 5$), ESB3 ($n = 3$) and ESAP1 ($n = 3$) knockdowns at 24 h post-induction; *n* values correspond to biological replicates. **d**, Volcano plot depicting upregulation of 'active' and 'silent' *ESAGs* following ESB2 RNAi. Differential expression determined by two-sided negative binomial generalized linear models (edgeR)

with Benjamini–Hochberg FDR correction. **e**, Transcript abundance at the active *VSG-2* locus. Purple circle, promoter; purple box, *VSG-2*; lilac boxes, *ESAGs*. RPM, reads per million. Bin size, 0.1 kb. **f**, Box plots depicting *ESAG* transcript abundance at the active ES in the parental line versus ESB2 RNAi represented as fold change between both conditions. Boxes span between the 25th and 75th percentiles, the lines correspond to the mean values and whiskers span between minimum and maximum values; individual biological replicates are depicted. *ESAG2* could be detected only in 3 out of 5 replicates, and *ESAG11* in none when using only uniquely mapped reads. **g–j**, Scatter plots comparing average gene expression in the parental line (in RPKMs) with fold change in expression in ESB2 RNAi 24 h/parental (**g**), ESB2 RNAi 24 h/ESB3 RNAi 24 h (**h**), ESB2 RNAi 24 h/ESAP1 RNAi 24 h (**i**) and ESB3 RNAi 24 h/ESAP1 RNAi 24 h (**j**). **k**, Heatmap depicting the $\log_2(FC)$ within the active ES for induced and uninduced samples of ESB2, ESB3 and ESAP1 RNAi normalized against the parental line. ESB2 RNAi non-induced samples present some leaky expression. From top to bottom, genes are ordered as they appear in BES1 (active ES). *ESAG2* and *ESAG11* were omitted as their expression could not be detected in all replicates in the parental line when considering uniquely mapped reads only. In **f** and **k**, Ψ, pseudo; hyp, hypothetical.

---

Extended Data Fig. 7g). Conversely, ESB2, ESB3 and ESAP1 RNAi did not impact VEX2 localization (Fig. 4h,i and Extended Data Fig. 7h).

Altogether, these observations allowed us to define a complex network of co-dependencies at the ESB (Fig. 4j,k), whereby its integrity depends on both VEX2 and ESB1, albeit by independent mechanisms. In addition, VEX2 liaises the ESB to the SLAB via VEX1, and is required for the sequential recruitment of ESAP1, ESB3 and ESB2 to the ESB. Given that ESB2 and ESB3 depletion affected each other's abundance, we further analysed the interplay between them. As expected, $^{GFP\text{-}myc}$ESB2 and $^{6HA}$ESB3 colocalized (Fig. 4l), but also co-immunoprecipitated (Fig. 4m). Notably, this direct interaction is consistent with a model whereby ESB2 mislocalization following ESB3 depletion is a consequence of failed recruitment, not reduced protein levels (Fig. 4a–c).

Lastly, we performed a comparative transcriptomic analysis (Fig. 4n and Extended Data Fig. 7i–l) using the data generated in this study and previously published data for VEX2 RNAi[12]. The most significant changes following VEX2 depletion were the upregulation of 'silent' *ESAGs* (32-fold) and 'silent' *VSGs* (dual promoter 344-fold; single promoter 42-fold), reflecting activation of silent *VSG*-ESs, whereas the most significant change following ESB2, ESB3 and ESAP1 knockdown was the upregulation of 'active' *ESAGs* (Fig. 4n). Considering the hierarchy of co-dependencies we established (Fig. 4k), the modulation of the mRNA abundance of 'active' *ESAGs* is likely to be mediated by ESB2; similar changes observed following VEX2, ESB3 and ESAP1 knockdown probably stem from ESB2 mislocalization and/or downregulation. Thus, we then focused our subsequent efforts on ESB2.

## ESB2 is a functional RNA endonuclease and inactivation of key catalytic residues impacts localization to the ESB

While ESB2 has orthologues across kinetoplastids, ESAP1 is restricted to trypanosomes, both sharing the highest sequence similarity with orthologues in *Trypanosoma brucei gambiense*, *Trypanosoma equiperdum* and *Trypanosoma evansi*—interestingly, the only species in which ESB3 and *ESAGs* can be found[36] (Extended Data Fig. 8a–c). For ESB2, we found the closest experimentally determined structural relative to be human SMG6 (Fig. 5a,b and Extended Data Fig. 8f,g,i,j), an endonuclease and critical component of the nonsense-mediated decay (NMD) pathway, mostly known for the degradation of aberrant transcripts[37]. Notably, while these two proteins are not homologues, the catalytic residues required for the nuclease activity are strictly conserved[38,39] (Fig. 5b and Extended Data Fig. 8i,j). To confirm the enzymatic function, we purified the ESB2 nuclease domain in *Escherichia coli* (amino acids (aa) 95-C) (Extended Data Fig. 9a). As isolated nuclease domains rarely retain sequence specificity, which is typically determined by other domains or binding partners, we performed in vitro assays using a non-*ESAG*-related RNA substrate (Fig. 5c). We observed concentration-, time- and magnesium-dependent endonuclease activity, evidenced by the generation of distinct lower MW bands indicative of internal cleavage.

Next, to assess the importance of the nuclease activity in trypanosomes, we used clustered regularly interspaced short palindromic repeats–CRISPR-associated protein 9 (CRISPR–Cas9)-mediated

---

**Fig. 4 | A hierarchy of co-dependencies at the ESB negatively regulates *ESAG* and silent *VSG* expression in an ESB2- and VEX2-dependent manner, respectively. a,b**, Fluorescence microscopy analysis of ESB2$^{12myc}$ localization following ESB3 and ESAP1 RNAi (**a**, **b**), $^{6myc}$ESB3 localization following ESB2 and ESAP1 RNAi (**b**) and $^{6myc}$ESAP1 localization following ESB2 and ESB3 RNAi (**b**). **c**, ESB2, ESB3 and ESAP1 protein levels following VEX2, ESB2, ESB3 and ESAP1 RNAi. The values were derived from protein blotting analysis of three biological replicates per cell line and are represented in a floating bar graph in which the box spans between minimum and maximum values, the centre line is the median, and all datapoints are depicted. Two-tailed unpaired *t*-tests were applied; NS, non-significant; **$P < 0.01$. **d–i**, Fluorescence microscopy analysis of Pol-I localization following VEX2, ESB1, ESB2, ESB3 and ESAP1 knockdown (**d,e**); ESB1$^{12myc}$, ESB2$^{12myc}$, $^{6myc}$ESB3 and $^{6myc}$ESAP1 localization following VEX2 knockdown (**f,g**); $^{6myc}$VEX2 localization following ESB2, ESB3 and ESAP1 knockdown (**h,i**). The graphs in **b**, **e**, **g** and **i** depict mean values of two biological replicates; >100 G1 cells per condition were analysed; all analyses were performed at 24 h post-induction. **j**, Heatmap summarizing the microscopy analyses; C1, clone 1; C2, clone 2. **k**, Diagram depicting the complex network of co-dependencies at the ESB and interface with the SLAB. The direction of the arrows indicates that the protein upstream

is required for the localization of the protein downstream to its corresponding nuclear compartment. **l**, Fluorescence microscopy colocalization analysis of $^{GFP\text{-}myc}$ESB2 and $^{6HA}$ESB3. The images in **a**, **d** (left), **h** and **l** were acquired using a Zeiss AxioObserver, whereas the images in **d** (right) and **f** were acquired using a Zeiss LSM980 Airyscan 2. All correspond to 3D projections by brightest intensity of 0.1-μm stacks. DNA was stained with DAPI (cyan or grey). Scale bars, 2 or 5 μm. **m**, Immunoprecipitation of $^{GFP\text{-}myc}$ESB2 using GFP nanobody-coated beads followed by protein-blot analysis using an anti-HA antibody to detect the prey ($^{6HA}$ESB3) and an anti-GFP antibody to detect the bait ($^{GFP\text{-}myc}$ESB2). IP, immunoprecipitation. The protein blots are representative of four independent experiments; a $^{6HA}$ESB3 single tagged cell line was used as a negative control. **n**, Violin plots depicting a comparative transcriptomic analysis between VEX2 ($n = 3$), ESB2 ($n = 5$), ESB3 ($n = 3$) and ESAP1 ($n = 3$) RNAi cell lines for 'active' *ESAGs*; *n* values correspond to biological replicates. $\log_2(FC)$, fold change in transcript abundance between RNAi (24 h post-induction) and the parental cell line. The violins span between minimum and maximum values, centre lines correspond to the mean and all datapoints are shown. Statistical significance was determined by one-way ANOVA followed by Tukey's multiple-comparison test; ****$P < 0.0001$. Diagram in **m** created in BioRender; Faria, J. https://BioRender.com/oia6hdu (2026).

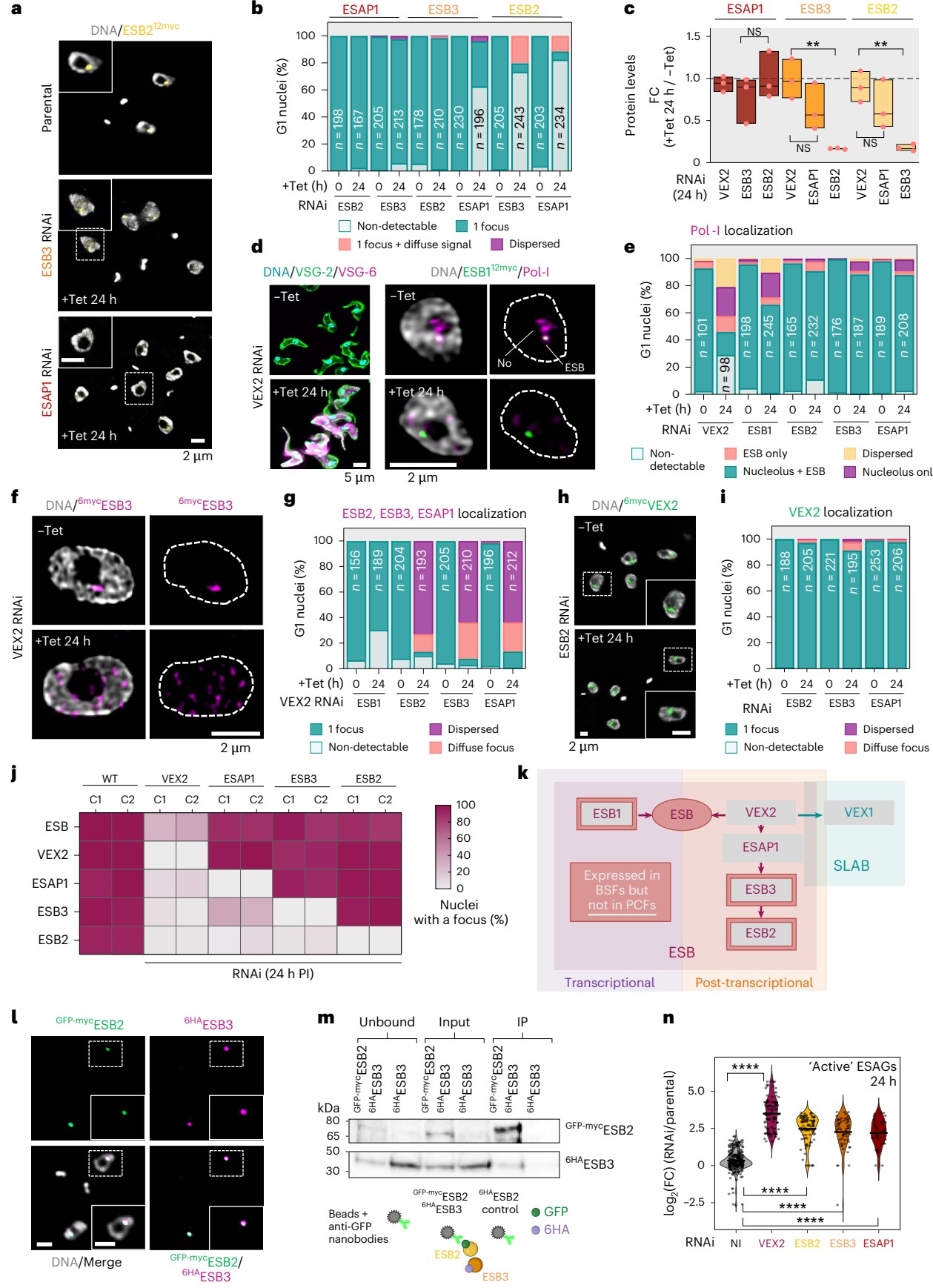

precision editing to introduce a point mutation in D240 (D1251 in human SMG6), which has been shown to render the nuclease inactive[38]. While we obtained transfectants in which both alleles contained a synonymous mutation, we were unable to generate mutants in which D240 had been replaced by alanine in both alleles (Fig. 5d and Extended Data Fig. 9b,c). Based on the fitness cost caused by its depletion, ESB2 is likely to be essential for parasite survival in a nuclease-dependent manner.

As we could not generate viable mutants when editing the endogenous alleles, we then expressed a tetracycline-inducible ectopic copy of ESB2, either wild type or a catalytic mutant (Fig. 5e–i and Extended Data Fig. 9d,e). We mutated three aspartic acids (D240A, D330A, D353A) to completely inactivate the nuclease[38,39]. Overexpression of wild-type ESB2 led to its accumulation at the ESB in 85.8% of nuclei often with additional nucleoplasmic signal; the focus at the ESB was larger than the one observed when the protein was expressed at endogenous levels (~2-fold increase, $P < 0.0001$; Fig. 5f,g and Extended Data Fig. 9d). To our surprise, ESB2 containing an inactive nuclease, when detected, localized solely to the cytosol (Fig. 5g and Extended Data Fig. 9d). Notably, the triple replacement abrogates the nuclease activity and/or RNA-binding capacity but does not compromise protein folding or stability[39,40]. It is conceivable that somehow these mutations led to a conformational change that either obstructed the nuclear localization signal (NLS) or the ability of ESB2 to interact with a binding partner that transports it into the nucleus through a piggyback mechanism or enables its retention at the ESB. We then fused the mutant version of ESB2 with a known NLS; subsequently, it could be detected in 68.4% of nuclei, but only in 24.2% of which, clearly accumulated at the ESB (Fig. 5e,g). A wild-type version fused with the same NLS was generated as a control, and no changes to its localization were observed (Fig. 5e,g). Both wild-type and mutant versions of ESB2 were expressed at similar levels (Fig. 5h,i); neither had an impact on the ESB (Extended Data Fig. 9e).

We then used a dual-inducible system to express RNAi-refractory wild-type or mutant ESB2 during knockdown of the endogenous protein (Fig. 5j). Ectopic copies were untagged; the mutant included an NLS to ensure nuclear localization. While the wild-type copy rescued the RNAi fitness cost, the catalytic mutant did not (Fig. 5k). Successful depletion and ectopic expression were confirmed at both RNA and protein levels (Fig. 5l,m).

Collectively, these experiments show that the nuclease activity of ESB2 (and/or RNA binding capacity) is required for function and recruitment to the ESB.

## ESB2 negatively regulates *ESAG* transcripts in a nuclease-dependent manner

Next, we ectopically expressed tetracycline-inducible untagged versions of either wild-type or mutant ESB2; no significant fitness cost was observed (Extended Data Fig. 10a,b). Furthermore, we analysed their transcriptomes and observed that overexpression of wild-type (3.2-fold increase, $P = 7.41 \times 10^{-34}$) or catalytically inactive ESB2 (3.4-fold increase, $P = 1.70 \times 10^{-41}$) led to a downregulation or upregulation of *ESAGs* at the active-*VSG*-ES, respectively (Fig. 6a–c, Extended Data Fig. 10c and Supplementary Data Sheet 8). Notably, the transcriptional profile obtained following overexpression of the catalytic mutant resembled that of the ESB2 knockdown, suggesting that the mutant competes with the endogenous wild-type protein at the ESB. The phenotype was less penetrant than the RNAi (~6.6- versus ~11-fold, respectively), presumably given the mutant's poor recruitment to the ESB (Fig. 5e,g). These results, supported by in vitro data (Fig. 5c), show that ESB2 negatively regulates *ESAG* transcripts through nuclease-dependent RNA decay. Therefore, to assess the impact of ESB2 on *ESAG* mRNA stability, we monitored RNA decay kinetics before and after its depletion, following transcription and splicing inhibition. However, in uninduced samples, ESB2 foci rapidly dispersed within 5 min of treatment (Fig. 6d), preventing a valid stability assessment as evidenced by unchanged *ESAG3* half-lives between conditions (~14–15 min; Fig. 6e). This sensitivity parallels the behaviour of VEX2 (refs. 11–13) and indicates that ESB2 is a dynamic ESB component, whose localization is dependent on transcription and RNA processing.

Finally, we sought to distinguish whether ESB2 targets *ESAGs* via sequence recognition or through a position-dependent mechanism

**Fig. 5 | ESB2 localization and function are dependent on its RNA nuclease activity. a,b**, ESB2 PIN domain analysis. Predicted ESB2 PIN domain using AlphaFold2 is superimposed with the PIN domain structure of human SMG6 (PDB ID: 2HWW). Root mean square deviation (RMSD) quantifies the average distance between corresponding atoms in two superimposed structures. Catalytic residues are highlighted in **b**. **c**, In vitro nuclease activity assay using a recombinant ESB2 PIN domain (aa 95-C) produced and purified from *E. coli*. The RNA content of the reaction mixtures was analysed by urea-PAGE. In the bottom panel, protein loading controls were resolved by SDS–PAGE and the triangle shows where gels were spliced to show the most relevant conditions (uncropped gels provided as source data). – absence; + presence. For ESB2 PIN specifically: +, 150 ng; ++, 1 µg. EDTA (metal chelator) and RNasin (RNase A, B and C inhibitor) were used as controls. **d**, CRISPR–Cas9-mediated precision editing of ESB2 D240 to alanine. No homozygous clones for the desired mutation could be generated; however, successful double-allele editing was achieved when using a repair template containing a synonymous mutation. A total of 15 clones were screened for both synonymous and non-synonymous mutations, respectively. The data are representative of two independent experiments. **e–i**, Tetracycline-inducible overexpression of ESB2[12myc] in BSFs. **e–g**, Fluorescence microscopy analysis of Pol-I and ectopically expressed ESB2[12myc]. Wild type or a catalytic mutant (D240A, D330A, D353A), with or without an La-NLS sequence, was analysed. Images were acquired using a Zeiss LSM980 Airyscan 2 and correspond to 3D projections by brightest intensity of 0.1-µm stacks. DNA was stained with DAPI (grey). The violin plot (**f**) shows data from >100 G1 cells and is representative of two biological replicates and independent experiments; the plot spans between minimum and maximum values, centre lines correspond to the mean, and all datapoints are shown. The stacked graph (**g**) depicts mean values of two biological replicates; >100 G1 cells per condition were analysed. Data in **e–g** correspond to 24 h post-induction. Yellow arrows point to the presence or absence of a visible ESB2 focus at the ESB. **h,i**, Protein blotting analysis of wild-type (**h**) or mutant (**i**) ESB2 expression. 'Native' corresponds to a cell line in which ESB2 was endogenously tagged. **j–m**, Tetracycline-inducible expression of ESB2[Mut-NLS] and ESB2[WT] following depletion of endogenous ESB2. **j**, Schematics summarizing the cell lines that were generated. The ectopic copies were recoded to be refractory to the RNAi target sequence and were untagged. **k**, Cumulative growth following tetracycline induction. The ESB2 RNAi cell line was used as a benchmark. Regarding the statistical analysis, the colours indicate which cell line they refer to and correspond to induced versus non-induced conditions for each time point. **l**, RT-qPCR analysis showing successful expression of *ESB2[Mut-NLS]* and *ESB2[WT]* ectopic copies following endogenous ESB2 depletion. Primers annealed in the *ESB2* 5'UTR and beginning of its coding sequence amplifying only endogenously expressed *ESB2* (left side) or in the recoded region amplifying only ectopically expressed *ESB2* (right side). The expression of the gene of interest was normalised against actin (housekeeper gene) and shown as fold change in relation to the parental cell line. Positive and negative values indicate up- and downregulation, respectively. Each experiment was conducted using three technical replicates per condition, which were averaged. In **k** and **l**, graphs depict mean values of four (*ESB2[Mut-NLS]*) or two (*ESB2[WT]*) biological replicates ± s.d. **m**, Protein blotting analysis showing the depletion of endogenous ESB2, which was C-terminally tagged with 12xmyc in the presence of ectopically expressed *ESB2[Mut-NLS]* or *ESB2[WT]*. Statistical analysis was performed as follows: two-tailed unpaired *t*-test (**f**), two-tailed paired *t*-tests with multiple-comparison correction using the Holm–Šídák method (**k**) and one-way ANOVA followed by Tukey's multiple-comparison test (**l**). *$P < 0.05$; **$P < 0.01$; ***$P < 0.001$; ****$P < 0.0001$. Error bars in **k** (non-visible) and **l** correspond to standard deviation. **h,i,m**, EF1α was used as loading control. Diagram in **d** created in BioRender; Faria, J. https://BioRender.com/iy0mp60 (2026).

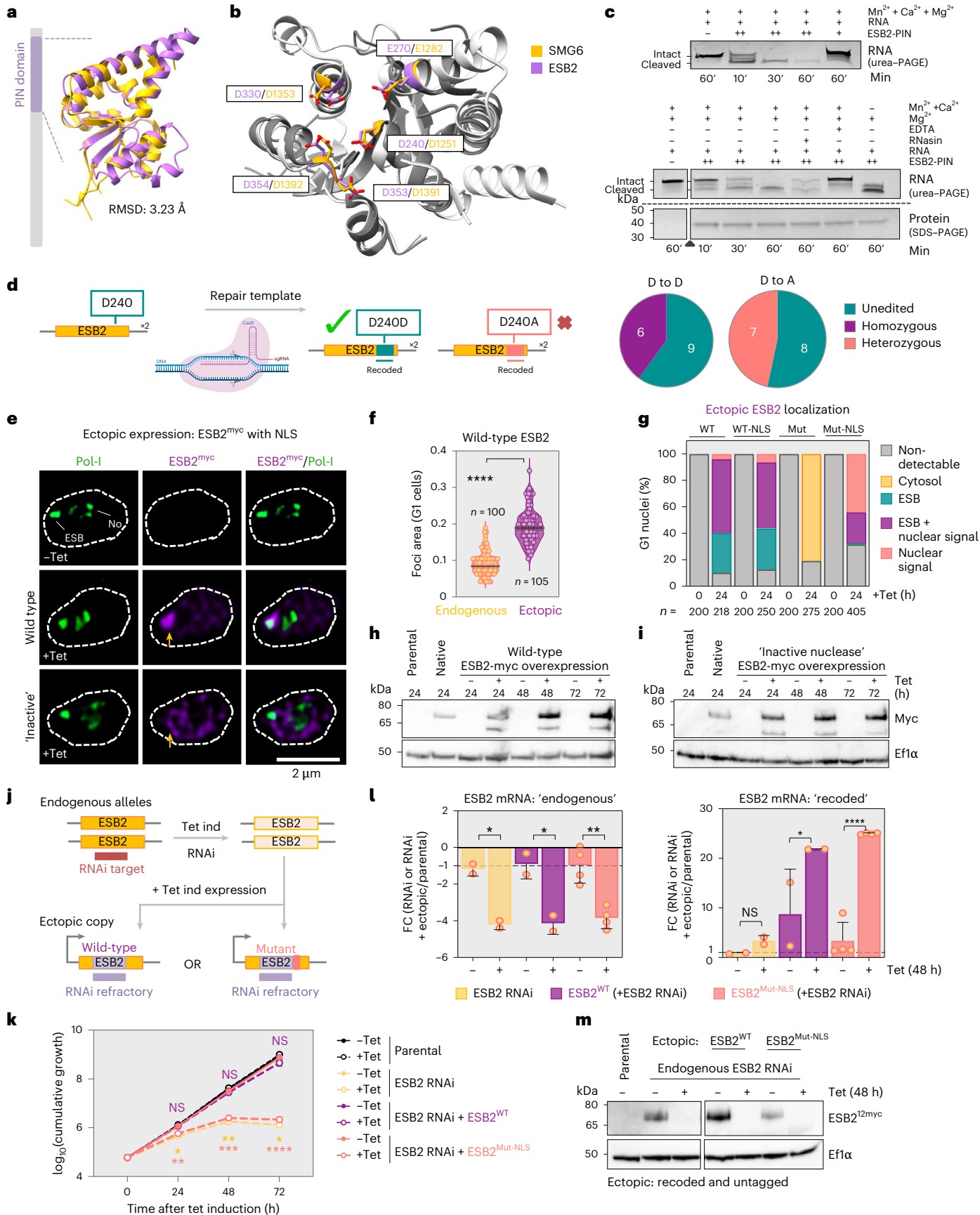

mediated by spatial restriction of its activity. We integrated an *RFP::PAC* reporter flanked by tubulin UTRs (unrelated DNA elements) either proximal to the active-ES promoter (~2 kb upstream of the first *ESAG−ESAG7*) or ~26.5 kb downstream (between *ESAG8* and *ESAG3*). Following ESB2 knockdown (~3-fold, $P < 0.05$), and consequent *ESAG7/ESAG3* upregulation (~2.5–4-fold, $P < 0.01$), *RFP* levels remained unchanged at the promoter-proximal site. Conversely, the downstream reporter was significantly upregulated (~3-fold, $P < 0.01$) (Fig. 6f,g). We further validated this using *ESAG3* UTRs. Tubulin UTRs conferred higher *RFP* mRNA amounts due to increased mRNA stability (Extended Data Fig. 10d), as previously reported[41]; however, the fold change following ESB2 depletion was comparable regardless of the flanking UTRs (~3-fold versus ~3.6-fold, Fig. 6g). While other post-transcriptional mechanisms may affect *RFP* transcripts, these results imply that the activity of ESB2 is spatially regulated rather than sequence specific. Consistent with this finding, MEME analysis[42] of *ESAGs* and their UTRs identified no discrete consensus sequences indicative of specific nuclease recognition (Extended Data Fig. 10e).

In summary, ESB2 functions as a spatially restricted endonuclease that post-transcriptionally regulates gene expression within the ESB (Fig. 6h,i and Extended Data Fig. 10j).

## Discussion

To this day, our understanding of the mechanisms underpinning antigenic variation in parasites that cause malaria and sleeping sickness is incomplete[1,43]. In *T. brucei*, *ESAGs*, involved in key host–parasite interactions, are mysteriously present in *VSG*-ESs, possibly to achieve mammalian-specific expression or alternatively for host adaptation. For instance, sequence variation in the transferrin receptor (an ESAG6 and ESAG7 heterodimer) was originally thought to enable uptake of transferrin from different mammals[4,44]. However, recent studies propose that this diversification is instead driven by a need for antigenic variation to prolong survival in the host[8]. Whether other *ESAGs* are involved in host adaptation and/or contribute to antigenic variation is yet to be determined. Overall, despite their critical roles, and co-transcription in the same polycistronic unit, *ESAGs* are required—and indeed expressed—at much lower levels than the active *VSG*. Here we identified the molecular machinery that negatively regulates *ESAG* transcripts. Together with processes responsible for enhanced *VSG* mRNA processing and stability[13,14,17–19], this mechanism yields a >140-fold differential expression between these genes.

Notably, over 20 years after its discovery, only two ESB components had been identified, ESB1 and VEX2 (refs. 11,12). Here we identified three new factors and a layer of post-transcriptional regulation

that was previously uncharacterized. Furthermore, ESB2, ESB3 and ESAP1 are developmentally regulated and integrate a complex network of co-dependencies consistent with a modular ESB architecture, composed of distinct transcriptional and post-transcriptional subdomains. While ESB1 acts as a transcriptional activator[11], VEX2 is a multi-functional factor with a central role in the exclusion of 'silent' *VSG*-ESs[12,14] and coordination of post-transcriptional regulatory processes. Specifically, it facilitates *VSG* mRNA processing via VEX1-mediated spatial proximity to the *SL* array[13,14], while simultaneously assembling the ESAP1–ESB3–ESB2 'module' to negatively regulate *ESAG* expression (Extended Data Fig. 10j; recently reviewed in ref. 45). The absence of this 'module' in previous VEX-affinity purifications[12,14] suggests its VEX2-dependent localization is linked to a structural role, implying spatial proximity over stable interaction. However, future work should clarify if distinct VEX2 sub-complexes operate along the active *VSG*-ES. Through the proximity proteome of VEX2, we also identified new components of the SLAB, the NUFIP body[17], and a remarkable enrichment of the CPSF complex[46] at the ESB. This restricted access to mRNA processing machinery aligns with the inefficient polyadenylation of transcripts from silent *VSG*-ESs[47] and supports the role of VEX2 in spatially integrating transcription and RNA processing in the 3D nuclear space[13,14].

Eukaryotic PIN domains mediate various RNA cleavage processes, including NMD—a pathway for degrading aberrant transcripts that also regulates genome homeostasis and specialized mRNA decay[40,48–50]. Although VEX2, ESAP1 and ESB2 share structural features with NMD regulators (Extended Data Fig. 8d–j and 10j), they are not homologues. Instead of NMD surveillance, which is dispensable in *T. brucei*[51,52], these proteins function distinctively in selective transcript tuning.

Nucleases show diverse specificity and activation states, yet substrate recognition by PIN-domain proteins remains poorly defined[40,53]. Therefore, while ESB2 appears to operate in a sequence-independent manner, with specificity stemming from spatial confinement instead, the precise mechanism regulating its activity along the active *VSG*-ES remains to be determined. As eukaryotic PIN domains tend to be part of larger protein complexes, it is conceivable that the interaction with a binding partner could either direct substrate binding and/ or activate the nuclease that is otherwise kept in an 'off state'. ESB3 could be one such molecule, as it is required for the recruitment of ESB2 to the ESB and found exclusively in African trypanosomes that possess *ESAGs*. In this model, stoichiometric dependence on limiting ESB3 would cap active complex formation, thereby explaining the modest *ESAG* reduction following ESB2 overexpression, while safeguarding against promiscuous off-target cleavage. Future work

---

**Fig. 6 | ESB2 negatively regulates *ESAG* transcripts, and specificity is achieved through spatial restriction of its nuclease activity. a–c**, RNA-seq analysis of ESB2[WT] and ESB2[Mut-NLS] overexpression at 72 h post-induction. Three biological replicates were used as well as parental and uninduced controls. **a,b**, Scatter plots comparing average gene expression in the parental line (in RPKMs) with fold change in expression in ESB2[WT] overexpression over parental (**a**) and ESB2[Mut-NLS] overexpression over parental (**b**). **c**, Heatmap depicting the log2(FC) within the active ES (BES1) for induced samples of ESB2[WT] and ESB2[Mut-NLS] overexpression compared with ESB2 RNAi—all normalized against the parental line. Hierarchical clustering—average linkage, Euclidean distance. *ESAG2* and *ESAG11* were omitted as their expression could not be detected in all replicates in the parental line when using uniquely mapped reads. **d**, Fluorescence microscopy analysis of ESB2[12myc] treated with actinomycin D (ActD) or sinefungin (Sinef), transcription and *trans*-splicing inhibitors, respectively, for 5 min or 30 min versus untreated controls (in an ESB2 RNAi background, non-induced samples). [6myc]VEX2 was used as a positive control (wild-type background). More than 100 cells were analysed per condition. In the left panel, DNA was stained with DAPI (grey). **e**, *ESAG3* mRNA turnover before and after ESB2 RNAi (24 h). Individual datapoints are depicted; mRNA half-life was determined using a one-phase exponential decay equation. **f,g**, Analysis of cell lines containing an *RFP::PAC* fluorescent reporter at the active *VSG*-ES before and after ESB2 RNAi. The schematics at the top show where the

reporters were integrated; the reporters were either flanked by *TUB* (tubulin) or *ESAG3* UTRs. *RFP*, *ESAG3* or *ESAG7* and *ESB2* mRNA levels were determined by RT-qPCR. In **f**, on the left side, fluorescence microscopy analysis is depicted. For the RT-qPCR experiments, the expression of the gene of interest was normalized against actin (**f,g**) or rRNA (**e**) and shown as fold change in relation to the parental or uninduced cell line or in relation to the initial mRNA levels, respectively. Positive and negative values indicate up- and downregulation, respectively. Each experiment was conducted using three technical replicates per condition, which were averaged. In **d–g**, graphs depict mean values of one (**d,e**), two (**g**) or three (**f**) independent clones of ESB2 RNAi and two technical replicates (**d**), or two (**f,g**) or three (**e**) independent experiments. In **f** and **g**, error bars correspond to standard deviation; statistical analysis was performed using one-way ANOVA followed by Tukey's multiple-comparison test. *$P < 0.05$; **$P < 0.01$; ***$P < 0.001$. **h–i**, Cartoon depicting the nuclear architecture of *T. brucei* BSFs, including all known components of the ESB and the VEX-mediated interaction with one of the *SL* arrays. ESB1 is a transcriptional activator required for transcription at the active ES. VEX2 is an exclusion factor, which prevents activation of silent *VSGs* and therefore a negative regulator. ESB2 negatively regulates *ESAG* expression; its recruitment to the ESB is sequentially dependent on VEX2, ESAP1 and ESB3. Diagrams in **h** and **i** created in BioRender: Faria, J. https://BioRender.com/u9tacde (2026).

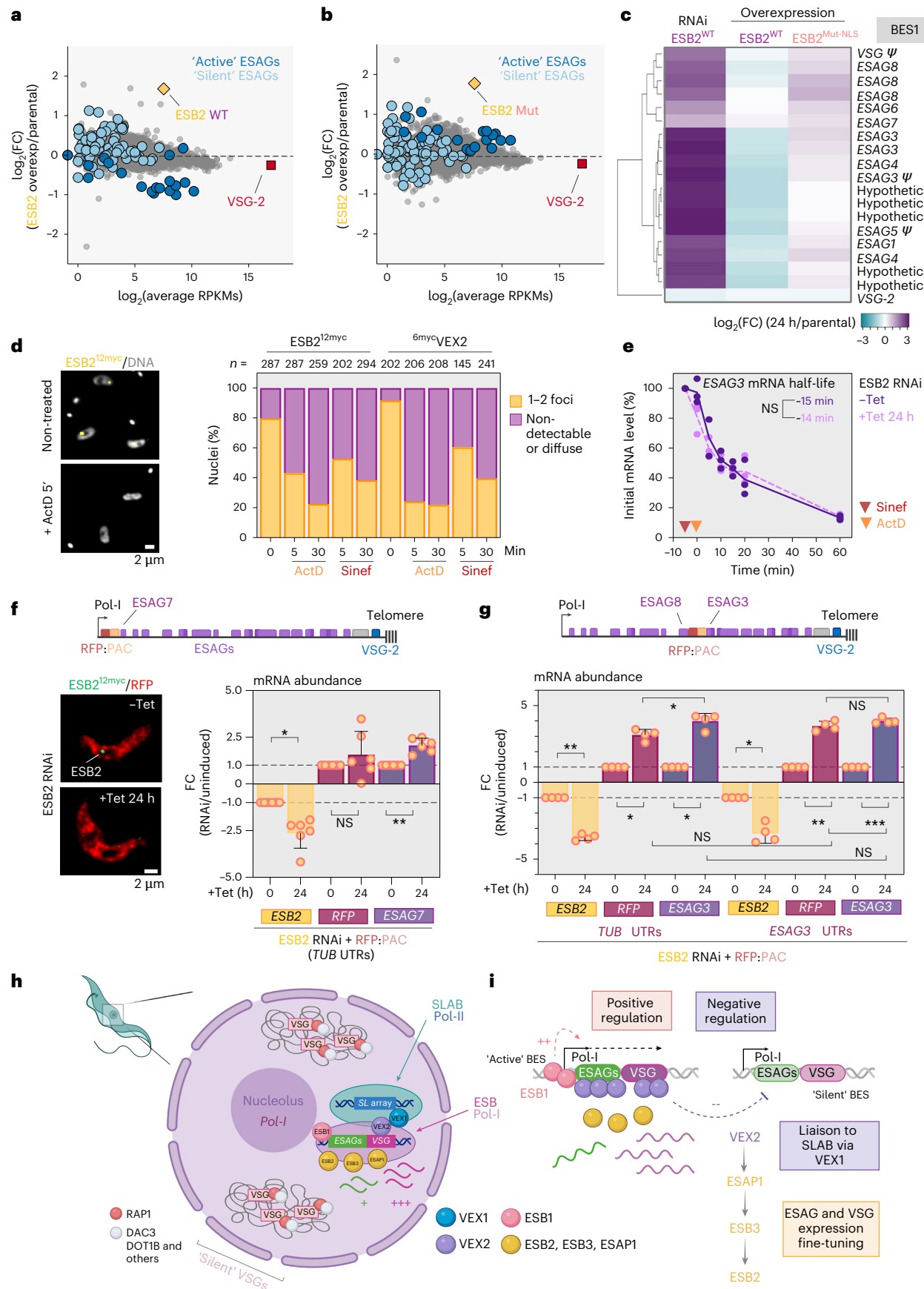

should elucidate whether ESB2 acts on precursor or mature RNA and whether undetectable pools of ESB2 might operate at 'silent' *VSG*-ESs. In addition, while the determinants of differential *ESAG* upregulation remain elusive (Extended Data Fig. 10f–i), they may include factors such as RNA secondary structure that could impact cleavage susceptibility.

The spatial restriction of ESB2 appears strictly coupled to functional activity. While recruitment is contingent upon upstream factors (VEX2, ESAP1 and ESB3), retention might rely on a continuous flux of active transcription and RNA processing. Although this sensitivity could reflect its dependence on VEX2—whose compartmentalization is itself transcription dependent—the distinct requirement for the intrinsic catalytic and/or RNA-binding capacity of ESB2 supports a model of substrate-mediated retention. Moreover, ESB2 is likely to contain post-translational modifications; both SUMOylation and phosphorylation have been previously implicated in regulating the stability and function of nucleases[54,55]. Notably, ESB2 contains a predicted SUMO-interacting motif[56]. Furthermore, we attribute the substantial fitness cost following ESB2 depletion to the consequences of the ~11-fold *ESAG* upregulation, as it is plausible that the resulting protein accumulation compromises viability through biosynthetic burden or by disrupting the stoichiometry of essential receptors and transporters.

In kinetoplastid parasites, the cellular proteome is sculpted almost exclusively by post-transcriptional mechanisms[57]; indeed, a recent screen identified potentially destabilizing motifs within *ESAG* 3′ UTRs[58]. We expand this framework by establishing ESB2 as a distinct upstream nuclear checkpoint. Unlike factors that broadly regulate mRNA cohorts (for example, RBP10, ZC3H11 (refs. [59,60])), ESB2 acts as a site-specific 'pre-filter' for the active-*VSG*-ES, pre-emptively refining stoichiometry before transcripts engage the cytoplasmic machinery. In summary, we identified a post-transcriptional mechanism of gene expression control, which is spatially regulated within a dedicated nuclear body, fine-tuning expression of proteins involved in host–pathogen interactions and immune evasion in *T. brucei*. Notably, if we consider toxin–antitoxin systems in bacteria[61] and that an exonuclease is also responsible for silencing genes linked to severe malaria[62], nucleases may indeed have a broader role in regulating pathogen-specific gene expression programmes. Overall, our work underscores how specialized RNA decay can regulate expression of specific genes.

## Methods

### *T. brucei* growth and manipulation

BSF *T. brucei*, Lister 427 (L427), 2T1 cells[63], 2T1/T7/Cas9 (ref. [64]) and T7/TetR/Cas9 (ref. [65]) cells were grown in HMI-11 medium, and PCF *T. brucei*, Lister 427 (L427) and PT1 cells were grown in SDM-79 medium and genetically manipulated using electroporation. Cytomix or human T cell nucleofector solution (Lonza) was used for all transfections.

In BSFs, puromycin, phleomycin, hygromycin, neomycin and blasticidin were used at 2 µg ml$^{-1}$, 2 µg ml$^{-1}$, 2.5 µg ml$^{-1}$, 2 µg ml$^{-1}$ and 10 µg ml$^{-1}$ for selection of recombinant clones, and at 1 µg ml$^{-1}$, 1 µg ml$^{-1}$, 1 µg ml$^{-1}$, 1 µg ml$^{-1}$ and 2 µg ml$^{-1}$ for maintaining those clones, respectively. Cumulative growth curves were generated from cultures seeded at 10$^5$ cells ml$^{-1}$, counted on a haemocytometer and diluted back to 10$^5$ cells ml$^{-1}$ as necessary.

In PCFs, puromycin, phleomycin, hygromycin, neomycin and blasticidin were used at 2 µg ml$^{-1}$, 2.5 µg ml$^{-1}$, 50 µg ml$^{-1}$, 10 µg ml$^{-1}$ and 10 µg ml$^{-1}$ for selection of recombinant clones, and at 1 µg ml$^{-1}$, 1 µg ml$^{-1}$, 1 µg ml$^{-1}$, 1 µg ml$^{-1}$ and 2 µg ml$^{-1}$ for maintaining those clones, respectively. Cumulative growth curves were generated from cultures seeded at 10$^6$ cells ml$^{-1}$, counted on a haemocytometer and diluted back to 10$^6$ cells ml$^{-1}$ as necessary.

Tetracycline was applied at 1 µg ml$^{-1}$ for RNAi, overexpression or Cas9 induction.

## Plasmids and constructs

For epitope tagging at the native locus[66], a previously generated plasmid was used: pNAT$^{TAGx}$ to add an N-terminal 6× c-myc to VEX2 (Tb927.11.13380)[12] and pNAT$^{xTAG}$ to add an C-terminal 12× c-myc to BRCA2 (Tb927.1.640)[27]. In addition, 25 new constructs were generated for our study: pNAT$^{xTAG}$ to add a C-terminal 12× c-myc to ESB1 (Tb927.10.3800), BDF6 (Tb927.1.3400), EAF6 (Tb927.9.2910), V2PL02 or ESB2 (Tb927.1.1820), V2PL03 (Tb927.10.14190), V2PL05 (Tb927.10.12030), V2PL06 (Tb927.10.15870), V2PL07 or SLAP2 (Tb927.4.3360), V2PL08 (Tb927.10.11230), KKIP2 or NufB3 (Tb927.5.1320), KKIP3 or NufB4 (Tb927.10.6700) and CPSF73 (Tb927.4.1340); a C-terminal 10× c-Ty to Pol-I RPA1 (Tb927.8.5090), RBP34 (Tb927.11.3340) and SNAP42 (Tb927.5.3910); and a C-terminal TurboID::4xGS::3xmyc to VEX1 (Tb927.11.16920) and VEX2 (Tb927.11.13380); and pNAT$^{TAGx}$ to add an N-terminal 6× c-myc to BDF2 (Tb927.10.7420), V2PL01 or ESB3 (Tb927.9.15850), V2PL04 or ESAP1 (Tb927.10.11600), KKIP2 or NufB4 (Tb927.5.1320), KKIP4 or NufB5 (Tb927.7.3080) and KKIP11 or NufB6 (Tb927.9.7820); an N-terminal 3xmyc::4xGS::TurboID to VEX2; and an N-terminal 6xHA to ESB3. TurboID sequences[20] were codon optimized for *T. brucei* expression[67].

For the RNAi experiments, a previously generated plasmid was used to target VEX2 (ref. [12]). For ESB1, ESB2, ESB3 and ESAP1 RNAi constructs, primers were selected from ORF sequences using RNAit (ref. [68]). A specific RNAi target fragment was amplified and cloned in pRPa$^{iSL}$ (ref [66]).

For tetracycline-inducible ectopic expression, the ESB2 ORFs, either wild type or containing an inactive nuclease (D240A, D330A and D353A), were cloned in the overexpression plasmid pRPa[69] containing a 12× c-myc tag at the C-terminus. A nuclear localization (NLS) sequence (La; RGHKRSRE) was added between the ORF and the 12× c-myc tag using HiFi cloning. Untagged versions of these plasmids were also generated. The ESB2 ORFs, either wild type or containing an inactive nuclease, were recoded and synthesized (GenScript) to be refractory to the RNAi construct above described. To generate constructs for dual-inducible overexpression and RNAi, the ESB2 RNAi stem-loop was then cloned into the above-mentioned pRPa ESB2$^{recoded}$ overexpression plasmids via Klenow.

The BES promoter-targeting construct, pESP::RFP::PAC, was derived from pESPi::RFP::PAC (ref. [69]) by removing the tetracycline operator. Synthetic sequences (Twist Biosciences) containing *RFP::PAC* either flanked by tubulin or *ESAG3* UTRs and homology regions for integration between *ESAG8* and *ESAG3* in BES1 were cloned into pESP::RFP::PAC via SacI/KpnI. All *RFP::PAC* cassettes were excised via SacI/PshAI or SacI/KpnI before transfection. To assess whether the reporters were successfully integrated in BES1, DNA was extracted with DNAzol and then analysed by a diagnostic PCR.

RNAi, overexpression and dual RNAi–overexpression cassettes were excised via AscI before transfection. All primers, amplicon sizes and enzymes used for either cloning or linearization before transfection are listed in Supplementary Data Sheet 9. All plasmids were confirmed by whole-plasmid sequencing using Plasmidsaurus.

## CRISPR–Cas9 endogenous tagging and precision editing

For CRISPR–Cas9-mediated endogenous tagging, sgRNAs and DNA templates were transfected into a T7/TetR/Cas9 cell line as described in (ref. [65]). ESB2 was tagged with a GFP-myc fusion at the N-terminus and all clones screened for double allele tagging by diagnostic PCR and subsequently validated by protein blotting and immunofluorescence.

For CRISPR–Cas9-mediated precision editing, the sgRNAs were cloned into pT7$^{sgRNA}$. The plasmid was sequenced for confirmation and then digested with NotI before transfection into a 2T1/T7/Cas9 parental cell line[64].

To edit ESB2 D240, dsDNA templates ESB2 D240D and D240A were generated by end-filling reactions. Templates were identical except for the D240 codon, which contained a synonymous mutation encoding

aspartate, or a mutation encoding alanine, respectively. Both templates contained 10 conservative changes in the region immediately upstream of the target codon, flanked by 50 bp of complete homology to the wild-type sequence (a schematic representation is depicted in Extended Data Fig. 9c). All substitutions were based on preferred codon usage where possible.

Tetracycline was added to induce Cas9 induction; 48 h later, 10 million cells constitutively expressing a gRNA were transfected with ~10–15 µg of repair template using Amaxa human T cell nucleofector solution (Lonza) and the programme Z-001. After 24 h, the cells were diluted to 1.5 cells ml$^{-1}$ and plated in 96-well plates in the absence of any selection drug and incubated for 5 days, then 10–15 subclones for each mutation were expanded. To assess whether successful precision editing occurred, DNA was extracted with DNAzol and then analysed first by a diagnostic PCR using a reverse primer that specifically recognized the synonymous substitutions present in the ESB2 repair templates, therefore only generating an amplicon if the template had been integrated. For the positive clones, a PCR was performed to amplify a larger fragment using primers that bind regions outside the repair template and that should amplify fragments regardless of whether editing occurred—the resulting amplicons were purified and sent for Sanger sequencing to confirm the genotypes. All primers are listed in Supplementary Data Sheet 9.

## PL–MS

**Optimization and validation.** The successful implementation of TurboID relied on rigorous validation and optimization of the generated fusion cell lines. Validation confirmed four critical aspects: (1) the fusion proteins showed the expected molecular weight by western blot; (2) they demonstrated correct localization to the anticipated nuclear compartments via fluorescence microscopy; (3) specific biotinylation could be detected within these target compartments by fluorescence microscopy; and (4) biotinylated material, including the bait protein, could be successfully affinity purified using streptavidin-coated beads, as confirmed by western blot. Throughout all validation and purification steps, controls included samples incubated without biotin and the parental wild-type cell line incubated with biotin. Furthermore, the biotinylation protocol was optimized by testing various biotin concentrations (50 µM, 150 µM and 500 µM) and incubation times (30 min and 1 h, 3 h, 6 h and 18 h), assessed via microscopy, western blot and pilot mass spectrometry (MS). While specific labelling was observed at both 30 min and 18 h, the 18-h incubation period significantly enhanced the signal intensity within the compartments of interest and provided a higher signal-to-noise ratio in the pilot MS experiments, establishing it as the optimal condition for subsequent large-scale experiments.

**Comparative PL–MS analysis.** The parasites expressing 3xmyc::4xGS::TurboID::VEX2 ($n = 6$), VEX2::TurboID::4xGS::3xmyc ($n = 4$) and VEX1::TurboID::4xGS::3xmyc ($n = 4$) were grown to $5 × 10^5$ cells ml$^{-1}$, at which point biotinylation was initiated in each cell line by adding biotin to 50 µM for 18 h. The parental line treated with biotin and TurboID cell lines treated with DMSO (biotin vehicle) were used as controls. After in vivo biotinylation, parasites were collected by centrifugation and washed three times in PBS and 2% glucose, and pellets were stored at −80 °C until lysis or processed right away.

The affinity purification of biotinylated material was adapted from refs. 20,70. Briefly, a pellet of $5 × 10^8$ parasites was used for each affinity purification, which was lysed in 1 ml of ice-cold RIPA buffer (50 mM Tris–HCl, pH 7.4; 150 mM NaCl; 1% NP-40; 0.5% sodium deoxycholate; 0.1% SDS) containing 0.1 mM PMSF, 1.5 µM Pepstatin and 0.1 mM TLCK. In addition, every 20 ml of RIPA was supplemented with 200 µl proteolytic protease inhibitor cocktail containing w/v 2.16% 4-(2-aminoethyl)benzenesulfonyl fluoride hydrochloride, 0.047% aprotinin, 0.156% bestatin, 0.049% E-64, 0.084% Leupeptin, 0.093%

Pepstatin A (Abcam) and one tablet of complete protease inhibitor EDTA free (Roche). Lysates were sonicated using Bioruptor (settings high; 30 s on and 30 s off; 3 cycles) according to the manufacturer's instructions. Subsequently, 1 µl of micrococcal nuclease was added to each lysate and digestion of nucleic acids proceeded for 10 min at room temperature (RT) followed by 50 min on ice. Lysates were clarified by centrifugation at 10,000 g for 10 min at 4 °C. For enrichment of biotinylated material, 100 µl of magnetic streptavidin bead suspension (1 mg of beads, Resyn Bioscience) was used for each affinity purification from $5 × 10^8$ parasites. Biotinylated material was affinity purified by end-over-end rotation at 4 °C overnight. Beads were washed in 1 ml of the following for 5 min each: RIPA for six washes, 4 M urea in 50 mM ammonium bicarbonate (AB) pH 8.5 for two washes, 6 M urea in 50 mM AB pH 8.5 for two washes, 1 M KCl in 50 mM AB pH 8.5 for two washes and 50 mM AB pH 8.5 for two washes. To confirm successful enrichment for biotinylated material before proteomics analysis, 4% of the material was used for protein blotting analysis. The remaining 96% was processed as follows:

Beads from each affinity purification were then resuspended in 200 µl 50 mM TEAB pH 8.5 containing 0.01% ProteaseMAX (Promega), 10 mM TCEP, 10 mM iodoacetamide, 1 mM CaCl$_2$ and 500 ng trypsin Lys-C (Promega). The on-bead digest was carried out overnight at 37 °C while shaking at 200 rpm. Supernatant from digests was retained and beads were washed for 5 min in 50 µl water, which was then added to the supernatant. Digests were acidified with trifluoroacetic acid to a final concentration of 0.5% before centrifugation for 10 min at 17,000 g. The supernatant was desalted using C$_{18}$ 0.6 µl ZipTips (Millipore); elution volume was 20 µl. Desalted peptides were dried for MS analysis.

## MS data acquisition

For PL–MS analysis, peptides were loaded onto an mClass nanoflow UPLC system (Waters) equipped with a nanoEaze M/Z Symmetry 100 Å C$_{18}$, 5 µm trap column (180 µm × 20 mm, Waters) and a PepMap, 2 µm, 100 Å, C$_{18}$ EasyNano nanocapillary column (75 µm × 500 mm, Thermo). The trap wash solvent was aqueous 0.05% (v:v) trifluoroacetic acid, and the trapping flow rate was 15 µl min$^{-1}$. The trap was washed for 5 min before switching flow to the capillary column. Separation used gradient elution of two solvents: solvent A, aqueous 0.1% (v:v) formic acid, and solvent B, acetonitrile containing 0.1% (v:v) formic acid. The flow rate for the capillary column was 330 nl min$^{-1}$ and the column temperature was 40 °C. The linear multi-step gradient profile was either 3–10% B over 7 min, 10–35% B over 30 min and 35–99% B over 5 min and then proceeded to wash with 99% solvent B for 4 min, or 2.5–10% B over 10 min, 10–35% B over 75 min and 35–99% B over 15 min and then proceeded to wash with 99% solvent B for 5 min. The same gradient was used for all samples that were directly compared. The column was returned to initial conditions and re-equilibrated for 15 min before subsequent injections.

The nanoLC system was interfaced with an Orbitrap Fusion hybrid mass spectrometer (Thermo) with an EasyNano ionization source (Thermo). Positive ESI–MS and MS2 spectra were acquired using Xcalibur software (version 4.0, Thermo). Instrument source settings were ion spray voltage, 1,900 V; sweep gas, 0 Arb; and ion transfer tube temperature, 275 °C. MS1 spectra were acquired in the Orbitrap with 120,000 resolution; scan range, m/z 375–1,500; AGC target, $4 × 10^5$; and maximum fill time, 100 ms. Data-dependent acquisition was performed in top speed mode using a fixed 1-s cycle, selecting the most intense precursors with charge states 2–5. Easy-IC was used for internal calibration. Dynamic exclusion was performed for 50 s after precursor selection, and a minimum threshold for fragmentation was set at $5 × 10^3$. MS2 spectra were acquired in the linear ion trap with scan rate, turbo; quadrupole isolation, 1.6 m/z; activation type, HCD; activation energy, 32%; AGC target, $5 × 10^3$; first mass, 110 m/z; and maximum fill time, 100 ms. Acquisitions were arranged by Xcalibur to inject ions for all available parallelizable time.

## MS data analysis

Peak lists in .raw format were imported into Progenesis QI (version 2.2., Waters) for peak picking and chromatographic alignment. A concatenated product ion peak list was exported in .mgf format for database searching against the *T. brucei brucei* L 427 subset of the TriTrypDB database[71] (11,388 sequences; 5,560,262 residues), appended with common proteomic contaminants. Mascot Daemon (version 2.6.1, Matrix Science) was used to submit searches to a locally running copy of the Mascot programme (Matrix Science, version 2.7.0.1). Search criteria specified were as follows: enzyme, trypsin; maximum missed cleavages, 1; fixed modifications, carbamidomethyl (C); variable modifications, oxidation (M), phospho (STY), acetyl (Protein N-term), biotin (Protein N-term, K); peptide tolerance, 3 ppm (# 13C = 1); MS–MS tolerance, 0.5 Da; and instrument, ESI-TRAP. Peptide identifications were passed through the percolator algorithm to achieve a 1% false discovery rate as assessed empirically by reverse database search, and individual matches were filtered to require minimum expected scores of 0.05. The Mascot.XML results file was imported into Progenesis QI, and peptide identifications associated with precursor peak areas were mapped between acquisitions. Relative protein abundances were calculated using precursor ion areas from non-conflicting unique peptides. For total proteome data, only non-modified peptides were used for protein-level quantification. Normalization and statistical testing were performed in Progenesis QI, with the null hypothesis being peptides are of equal abundance among all samples.

Normalized protein label-free peak areas were exported from Progenesis LFQ, and those proteins with at least two unique peptides identified were retained for downstream analysis. Missing values were imputed by drawing values from a left-shifted normal $\log_2$ intensity distribution to model low-abundance proteins (VEX2 C-term, mean = 16.8, s.d. = 3.5; VEX2 N-term, mean = 16.8, s.d. = 3.5; VEX1, mean = 17.5, s.d. = 3.0). Proximal proteins were determined with the limma package using options trend = TRUE and robust = TRUE for the eBayes function. Multiple testing correction was carried out according to Benjamini and Hochberg; the false discovery rate for identified proximals was 1%. GO term analysis was performed using TriTrypDB and Revigo.

## PL–MS 'hit' selection and prioritization

A total of 2,060, 2,075 and 1,885 proteins were identified in the [3myc]TurboID-VEX2, VEX2-TurboID[3myc] and VEX1-TurboID[3myc] datasets, respectively. The TurboID 'minus biotin' control samples showed significant biotinylation background, probably resulting from biotin present in the culture medium. Consequently, to effectively identify more subtle biotinylation changes, we used the parental wild-type cell line (plus biotin) as our primary negative control. Proteins significantly enriched (adj $P < 0.05$; FDR 1%) in the TurboID samples (incubated with biotin) compared with the parental cell line across the three datasets were initially collated. A stringent threshold of $\log_2(FC) > 1.5$ was applied to this combined set, which yielded 70 candidate proteins (highlighted in green in Supplementary Data Sheets 1–3).

The initial list was refined by subtracting known components of the ESB, SLAB and NUFIP body, as well as established telomere-binding proteins[9,11,12,15,17,22] and proteins classified as contaminants—highlighted in yellow and light orange, respectively, in Supplementary Data Sheets 1–3. This process narrowed the focus to 45 candidate proteins. Note that we defined 'contaminants' as proteins with verified non-nuclear localization, based on independent validation from the TrypTag database[24], and a comprehensive nuclear proteome analysis[25].

Of the 45 core candidates, 19 were further prioritized based on adherence to at least one of the following five criteria, designed to select candidates critical to novel or unique aspects of *T. brucei* biology:

- Evolutionary divergence and absence: candidates showing high sequence divergence (<30% sequence homology) in African trypanosomes or that are absent in other kinetoplastid parasites

lacking VSGs and antigenic variation (sequences sourced from TriTrypDB)
- PCF localization: localization to nuclear speckles, the nucleolar periphery or non-specific background signal (potentially indicative of low or no expression in this developmental stage) based on TrypTag data[24]
- Differential developmental expression: candidates showing >1.5-fold upregulation in BSFs versus PCFs[35], given that PCFs lack an ESB and do not express VSGs
- Functional domain prediction: presence of domains predicted to be involved in DNA and RNA binding, transcription, RNA processing and/or epigenetic regulation, as determined by structural and domain analyses using AlphaFold2, Foldseek and InterPro
- Novelty: candidates not previously screened in the medium-throughput tagging screen that initially identified ESB1 (ref. [11])

The iterative selection process described above is schematically represented in the Venn diagrams shown in Fig. 1h–j. Out of the 19 'high-priority hits' (highlighted in salmon in Supplementary Data Sheets 1–3), 17 were selected for endogenous tagging and subsequent localization confirmation via fluorescence microscopy (listed in Supplementary Data Sheet 5). Two proteins (CLP1, CSTF1) were excluded from individual tagging given their predicted involvement in the same multimeric assembly (CPSF complex) alongside CPSF3.

## Co-immunoprecipitation

BSF *T. brucei* cells ($2 \times 10^8$) with or without N-terminally GFP-myc-tagged endogenous copies of *ESB2* and an N-terminal 6xHA-tagged copy of ESB3 were washed three times in ice-cold PBS with EDTA-free protease inhibitor cocktail (Roche) and lysed in ice-cold lysis buffer (20 mM HEPES, pH 7.4, 1 mM $MgCl_2$, 10 μM $CaCl_2$, 250 mM sodium citrate, 0.1% Tween-20 plus EDTA-free protease inhibitor cocktail). Pipetting and incubation for 30 min at 4 °C facilitated lysis. Lysates were sonicated using Bioruptor (settings high; 30 s on and 30 s off; 3 cycles) according to the manufacturer's instructions. Micrococcal nuclease was added to each lysate, and digestion of nucleic acids proceeded for 10 min at RT followed by 50 min on ice. Lysates were clarified by centrifugation at 10,000 *g* for 15 min at 4 °C. The supernatant was removed and added to a low-protein-binding 1.5-ml Eppendorf tube containing GFP-Trap magnetic agarose beads (Chromotek) or Dynabeads (Invitrogen) cross-linked to a mouse anti-GFP (LifeTech) via bis(sulfosuccinimidyl) suberate (BS3, Thermo Scientific) and agitated at 4 °C for 2 h. The samples were then placed on a magnetic rack and washed 10 times with ice-cold lysis buffer. One final wash with ice-cold RIPA buffer containing EDTA-free protease inhibitor cocktail was performed. Samples were eluted with NuPAGE LDS loading buffer containing a reducing agent (Invitrogen). The resulting proteins were fractionated by sodium dodecyl sulfate-polyacrylamide gel electrophoresis (SDS–PAGE) and analysed by protein blotting (details below).

## Protein blotting

Protein samples were run according to standard protein separation procedures, using SDS–PAGE. Bis–Tris gels (4–12%) were used (NuPAGE, Invitrogen). The following primary antibodies were used: mouse α-myc (Millipore, clone 4A6, 1:10,000), mouse anti-Ty (BB2, Invitrogen, 1:5,000), mouse monoclonal anti-HA (Sigma, clone HA-7, 1:2,000), rabbit polyclonal anti-GFP (LifeTech, 1:2,000) and mouse α-EF1α (Millipore, clone CBP-KK1, 1:30,000). Streptavidin-HRP (Jackson ImmunoResearch) was used at 1:10,000 in PBS–5% BSA.

We used horseradish-peroxidase-coupled secondary antibodies (α-mouse and α-rabbit, Bio-Rad, 1:5,000). Blots were developed using an enhanced chemiluminescence kit (Amersham) according to the manufacturer's instructions. Densitometry was performed using Fiji v. 2.9.0.

## Immunofluorescence microscopy

Immunofluorescence microscopy was carried out according to standard protocols[12]. For wide-field microscopy, the cells were attached to 12-well 5-mm slides (Thermo Scientific). For super-resolution microscopy, the cells were attached to poly-L-lysine-treated high-precision coverslips (thickness $1^{1/2}$ mm), stained and then mounted onto glass slides. Cells were mounted in Vectashield with DAPI (wide field) or stained with 1 µg ml$^{-1}$ DAPI for 10 min and then mounted in Vectashield without DAPI (super-resolution). Primary antisera were rat α-VSG-2 (1:10,000), rabbit α-VSG-6 (1:10,000), rabbit α-myc (NEB, clone 71D10, 1:200), mouse α-myc (NEB, clone 9B11, 1:2,000), mouse α-Pol-I (largest subunit; 1:100 (ref. 15)), mouse monoclonal anti-HA (Sigma, clone HA-7, 1:1,000), rabbit polyclonal anti-GFP (LifeTech, 1:500) and mouse α-Ty (Invitrogen, BB2, 1:1,000). Streptavidin-Alexa fluor 488 was applied at 1:500 (Invitrogen).

The secondary antibodies were Alexa Fluor conjugated goat antibodies: α-mouse, α-rat and α-rabbit, Alexa Fluor 488, Alexa Fluor 555 Plus or Alexa Fluor 568 (1:1,000 for super-resolution microscopy or 1:2,000 for wide-field microscopy).

## Microscopy and image analysis

For wide-field microscopy, cells were analysed using a Zeiss AxioObserver Inverted Microscope equipped with a Colibri 7 narrow-band LED system and white LED for epifluorescent and white light imaging and ZEN Pro software (Carl Zeiss). Images were acquired as z-stacks (0.1–0.2 µm) and further deconvolved using the default settings ('good, medium') in ZEN Pro. For super-resolution microscopy, cells were analysed using a Zeiss LSM980 Airyscan 2 or a Zeiss Elyra 7 and the Zeiss ZEN software (Carl Zeiss). Representative images obtained by super-resolution microscopy correspond to maximum 3D projections by the brightest intensity of stacks of approximately 30 slices of 0.1 µm. Images acquired with the Zeiss LSM980 Airyscan 2 were deconvolved using Airyscan Joint Deconvolution (XY resolution ~90 nm). Super-resolution structured illumination microscopy (SR-SIM) was performed using a Zeiss Elyra 7 microscope in Lattice SIM$^2$ mode (XY resolution ~60 nm). SIM reconstruction was performed after correcting for chromatic aberrations using the channel alignment function in Zen and performing deconvolution using the default settings. Tetraspeck beads (Thermo; 100 nm) were adhered to slides and were used to determine channel alignment for each experiment. DAPI-stained T. brucei nuclear and mitochondrial DNA were used as cytological markers for cell-cycle stage; one nucleus and one kinetoplast (1N:1K) indicate G1, one nucleus and an elongated kinetoplast (1N:eK) indicate S phase, one nucleus and two kinetoplasts (1N:2K) indicate G2/M and two nuclei and two kinetoplasts (2N:2K) indicate post-mitosis[72]. All the images were processed and scored using Fiji v.2.9.0. Pearson's correlation coefficient was applied as a statistical measure of colocalization. Overlapping, adjacent and separate foci presented a Pearson's correlation coefficient in the ranges ≥0.5 to ≤1, ≥−0.5 to <0.5 and ≥−1 to <−0.5, respectively. Counts in total cells or specific cell cycle phases were typically performed using >100 nuclei. All quantifications are averages or representative of at least two biological replicates and independent experiments.

## Transcriptomics

RNA-seq analysis was performed in BSFs using 2T1 cells and uninduced or induced clones of ESB2 RNAi (24 h and 36 h), ESB3 RNAi (24 h and 36 h), ESAP1 RNAi (24 h and 36 h), ESB2 wild-type overexpression (72 h) and ESB2 'inactive nuclease' overexpression (72 h)—at least 3 biological replicates each. RNA-seq analysis was also performed in PCFs using PT1 cells and uninduced or induced clones of ESB2 RNAi (72 h), and ESAP1 RNAi (72 h)—2 biological replicates each. RNA was extracted using the RNeasy kit (Qiagen) according to the manufacturer's instructions. The RNA samples were sent for sequencing to BGI (Hong Kong). Briefly, polyadenylated transcripts were enriched using poly-dT beads and reverse-transcribed before sequencing on a DNBSeq platform. Each sample generated approximately 40 million reads (paired end, 100 bp) or 100 million reads (paired end, 150 bp).

Reads were mapped to either T. brucei L427 2018 (ref. 3) or a hybrid assembly consisting of the T. brucei 927 reference genome[73] plus the bloodstream VSG-ESs[4] and metacyclic VSG-ESs[74,75] from the Lister 427 strain. In the figures, we show the data mapped to the hybrid assembly because individual ESAG annotations are more amenable. The gene cohorts affected by the knockdowns or overexpression experiments we performed are the same and affected to the same extent irrespective of which assembly was used for the analysis. Bowtie 2 mapping was with the parameters --very-sensitive --no-discordant --phred33. Alignment files were manipulated and filtered for MapQ1 or MapQ10 in PCFs and BSFs, respectively, with SAMtools—indeed, uniquely mapping reads were used to distinguish ESAG transcripts. Bam files were inspected on IGV. Per-gene read counts were derived using subread. Reads per kilobase per million (RPKM) values were derived from normalized read counts, and differential expression analysis was conducted with edgeR. Coverage maps were generated using deepTools. For the analyses in Extended Data Fig. 10e,i, UTRs were predicted using two different approaches[76,77].

## mRNA turnover assay

ESAG3 mRNA stability was determined in uninduced or induced clones of ESB2 RNAi (24 h; 2 biological replicates) in BSFs of T. brucei. Briefly, cap methylation was inhibited by adding 5 µg ml$^{-1}$ sinefungin for 5 min to prevent trans-splicing (T-5 samples), then actinomycin D (10 µg ml$^{-1}$) was added (T0 samples) for up to 60 min to inhibit transcription. The following time points were obtained: T5, T10, T15, T20 and T60 min. Ten million cells were used per time point; RNA was extracted and analysed by RT-qPCR (details below). To determine mRNA half-life, data were fitted to a one-phase exponential decay equation using GraphPad Prism. Non-induced and RNAi samples were also collected for immunofluorescence analysis to determine ESB2 localization following transcription and splicing inhibition.

## RT-qPCR analysis

RT-qPCR analysis was performed to determine (1) RFP, ESB2, ESAG3 and ESAG7 mRNA levels in 2T1 BSF cells and uninduced or induced clones of ESB2 RNAi (24 h) with RFP:PAC reporters at the active-VSG expression site; (2) ESB2 mRNA levels in 2T1 BSF cells and uninduced or induced clones of ESB2 RNAi (24 h), overexpression of either wild type or containing an inactive nuclease (48 h, 72 h) or dual RNAi + overexpression (48 h); and (3) ESAG3 and ESAG7 mRNA stability in induced (24 h) and uninduced samples of ESB2 RNAi. Two or three biological replicates were analysed with three technical replicates per condition.

RNA was extracted using the RNeasy kit (Qiagen) according to the manufacturer's instructions. Reverse transcription of RNA and quantitative PCR was performed using the Luna Universal One-step RT-qPCR Kit using 2.5–5 ng or 25 ng of RNA per reaction, for RFP/ESAG3/ESAG7 or ESB2-related amplifications, respectively. Reactions were carried out in a QuantStudio 3 or QuantStudio 7 real time PCR system (ThermoFisher) and typically included reverse transcription at 55 °C for 10 min, followed by initial denaturation at 95 °C for 1 min followed by 40 cycles of 95 °C for 10 s and 60 °C for 30 s with a signal read at the end of each cycle plus a final melting curve to check fidelity from 60 °C to 95 °C, with a signal read every 1 °C. The data were normalized against the T. brucei actin gene, except for the mRNA stability experiments, in which 28Sβ rRNA was used (0.004 ng of RNA per reaction). Fold changes in mRNA abundance were calculated using the ΔΔCT method, normalized to the parental strain T. brucei 2T1 or uninduced samples of ESB2 RNAi and shown as relative abundance. As ESB2 'recoded' sequences were absent in the parental line and were not detected, an arbitrary Cq value of 30 cycles was applied for normalization purposes. All primers are listed in Supplementary Data Sheet 9.

## Recombinant protein expression and purification

The sequence encoding for ESB2 nuclease domain (aa 95-C) was cloned into a pET vector containing an N-terminal His-MBP solubility tag using InFusion (Clontech) cloning and expressed in *E. coli* BL21 DE3. All primers are listed in Supplementary Data Sheet 9. For large-scale purification, 1.5 l of cells was grown in LB medium to OD600 = 0.6 and expression induced adding IPTG to a final concentration of 0.5 mM and incubated at 18 °C for 16 h. The cells were collected, and the pellet resuspended in lysis buffer (50 mM HEPES pH 8.0, 20 mM imidazole, 0.5 M NaCl, 1 mM $CaCl_2$, 5 mM $MgCl_2$, 5% glycerol, 0.5 mM TCEP, 0.5 mM PMSF, 0.5 mM leupeptin and 0.5 mM pepstatin A). The cell suspension was then lysed using the CF2 Cell Disruptor (Constant Systems; 1 × 20 kPSI), treated with DNaseI (4 u $\mu l^{-1}$, 1 h on ice) and sonicated (3 × 30 s). After centrifugation (27,000 $g$, 20 min, 4 °C), the clarified lysate was subjected to nickel-affinity chromatography using a 5-ml HisTrap HP (Cytiva) column, washed with His washing buffer (20 mM HEPES pH 8.0, 20 mM imidazole, 0.5 M NaCl, 5 mM $MgCl_2$, 0.25 mM TCEP) and step eluted with His elution buffer (20 mM HEPES pH 8.0, 500 mM imidazole, 0.5 M NaCl, 5 mM $MgCl_2$, 0.25 mM TCEP). Fractions from the nickel column purifications were pooled and dialysed into heparin column loading buffer (20 mM HEPES pH 8.0, 350 mM NaCl, 5 mM $MgCl_2$, 0.25 mM TCEP). 6His-3C-protease (H3C) was added to the dialysis tubing, and the resulting His-MBP tag cleavage was allowed to proceed for 16 h. The sample was then injected into a 5-ml HiTrap Heparin HP (Cytiva) affinity column equilibrated and then washed with heparin buffer A (20 mM HEPES pH 8.0, 350 mM NaCl, 5 mM $MgCl_2$, 0.25 mM TCEP) and sample eluted in a gradient against heparin buffer B (20 mM HEPES pH 8.0, 1 M NaCl, 5 mM $MgCl_2$, 0.25 mM TCEP). The protein content of total, soluble and insoluble extracts, as well as chromatography elution fractions, flowthroughs and washes, were analysed by SDS–PAGE.

The purest fractions from the heparin purification were pooled and subsequently concentrated using Vivaspin 500 10 kDa molecular weight concentrators (MWCO) for biochemical assays. The identity of the band was confirmed to be the ESB2 nuclease domain by MS analysis.

## In vitro RNA nuclease activity assay

The assay was adapted from ref. 38. Briefly, 20 pmol of RNA (5′- ACCAUGAUUACGAAUUGCUUGGAAUCCUGACGAACUGUAGAC-CACCGAACGACCCACCAG-3′) was incubated with 150 ng or 1 μg of purified ESB2 nuclease domain in reaction buffer (20 mM HEPES pH 7.0, 75 mM NaCl, 5 mM $MgCl_2$, 5 mM $MnCl_2$, 5 mM $CaCl_2$, 10% PEG4000 (w/v), 1 mM DTT) in a total of 14 μl per reaction. The extent of RNA degradation was evaluated at different time points (10 min, 30 min and 60 min) by analysing the reaction mixtures on 15% urea-PAGE gels. RNA only, EDTA (30 mM) and RNasin (20 U) controls were performed. Note that we initially performed assays with a panel of divalent cations ($Mg^{2+}$, $Mn^{2+}$, $Ca^{2+}$) to identify potential cofactors and found the activity to be $Mg^{2+}$ dependent and, as expected, inhibited by the metal chelator EDTA. RNasin is a general RNase inhibitor used to ensure that the observed RNA degradation did not result from RNase contamination in our protein preparation. Protein controls were resolved by SDS–PAGE.

## Phylogenetic analysis and structural prediction

DNA and amino acid sequences of ESB2, ESB3 and ESAP1 and respective kinetoplastid orthologues were obtained from the TriTrypDB. The remaining protein sequences were obtained from Uniprot. Phylogenetic analysis was performed using TreeViewer. Structural predictions were performed with AlphaFold2 optimized for trypanosomatid proteins[78]. Sequence alignments were generated with Clustal W and shown in ENDscript or UCSF ChimeraX. Protein structures were visualized and superimposed in UCSF ChimeraX. Sequence motif analysis was performed using InterPro, and structure-based domain analysis was performed using FoldSeek.

## Data visualization and statistics

Heat maps and radial, volcano, scatter and violin plots were generated in RStudio using the following packages: ggplot2, RColorBrewer, ggrepel, dplyr, scales and tidyverse. Growth curves, heat maps, box plots and floating bar and stacked bar graphs were generated using GraphPad Prism Software (version 10.0). FragPipe-Analyst was used to generate the heat maps in Extended Data Fig. 1h–j. The GO term network analysis was generated using Cytoscape. All statistical analyses were performed using GraphPad Prism Software (version 10.0), except for the transcriptomic and proteomics analyses, which were conducted using R (details provided in the respective sections).

## Resources and reagents

Details of resources and reagents can be found in Supplementary Data Sheet 10, including a complete list of software and databases, their respective references and webpages. All unique materials are available on request.

## Reporting summary

Further information on research design is available in the Nature Portfolio Reporting Summary linked to this article.

## Data availability

RNA-seq data were deposited in the European Nucleotide Archive (PRJEB89423). Proteomics data were deposited in ProteomeX-change (PXD063534) and are available for download from MassIVE (MSV000097776). The deposited file-names match the sample names used in Supplementary Data Sheets 1–3. Source data are provided with this paper.

## Code availability

All scripts are available via Zenodo at https://zenodo.org/records/15357091 (ref. 79).

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

## Acknowledgements

This work was supported by a Wellcome Trust/Royal Society Sir Henry Dale Fellowship to J.R.C.F. (222573/Z/21/Z). L.W. was supported by a PhD scholarship by the Biotechnology and Biological Sciences Research Council (BBSRC) White-Rose doctoral programme in Mechanistic Biology (BB/T007222/1). We would like to thank the Imaging and Cytometry, Genomics, Protein Purification and Proteomics and Metabolomics labs in the Bioscience Technology Facility at the University of York, particularly J. Cartwright and M. Tyreman, for assistance with protein expression and purification. The York Centre of Excellence in Mass Spectrometry was created thanks to a major capital investment through Science City York, supported by Yorkshire Forward with funds from the Northern Way Initiative, and subsequent support from EPSRC (EP/K039660/1; EP/M028127/1). At the University of York, we thank V. Geoghegan for advice regarding the proximity labelling experiments, B. Spink for assistance in cell line maintenance and C. Hill for invaluable advice on nuclease activity assays. We thank S. Hutchinson and D. Horn (University of Dundee) for kindly gifting the PT1 cell line, R. McCulloch (University of Glasgow) for the BRCA2 tagging plasmid and C. Tiengwe (Imperial) for advice on the experiments involving *ESAG3*. Finally, we would like to sincerely thank K. Gull (University of Oxford) and all members of the Mottram and Cayla labs (University of York) for helpful discussions.

## Author contributions

J.R.C.F. acquired funding and conceived and supervised the study. L.I.M.L., H.M.A., L.W., S.L., M.J. and J.R.C.F. planned and performed experiments. L.I.M.L. and J.R.C.F. performed PL–MS experiments. L.I.M.L., A.D. and J.R.C.F. analysed proteomics data. L.I.M.L., H.M.A., M.J. and J.R.C.F. generated and analysed microscopy data. L.I.M.L., H.M.A. and J.R.C.F. generated and analysed endogenous tagging, RNAi and *RFP::PAC* reporter cell lines. L.I.M.L. generated and analysed overexpression cell lines. H.M.A. performed co-immunoprecipitation, transcription and splicing inhibition experiments as well as RNA stability experiments. L.I.M.L., J.L.R.-C. and J.R.C.F. analysed transcriptomics data. L.W. performed structural bioinformatics with input from L.I.M.L. and J.R.C.F. L.W. performed recombinant protein expression, purification and in vitro nuclease activity experiments. S.L. performed CRISPR–Cas9-mediated precision editing. J.R.C.F. was responsible for data visualization with input from L.I.M.L., H.M.A., L.W. and A.D. J.R.C.F. wrote the original draft; all authors revised and reviewed the paper.

## Competing interests

The authors declare no competing interests.

## Additional information

**Extended data** is available for this paper at https://doi.org/10.1038/s41564-026-02289-4.

**Correspondence and requests for materials** should be addressed to Joana R. C. Faria.

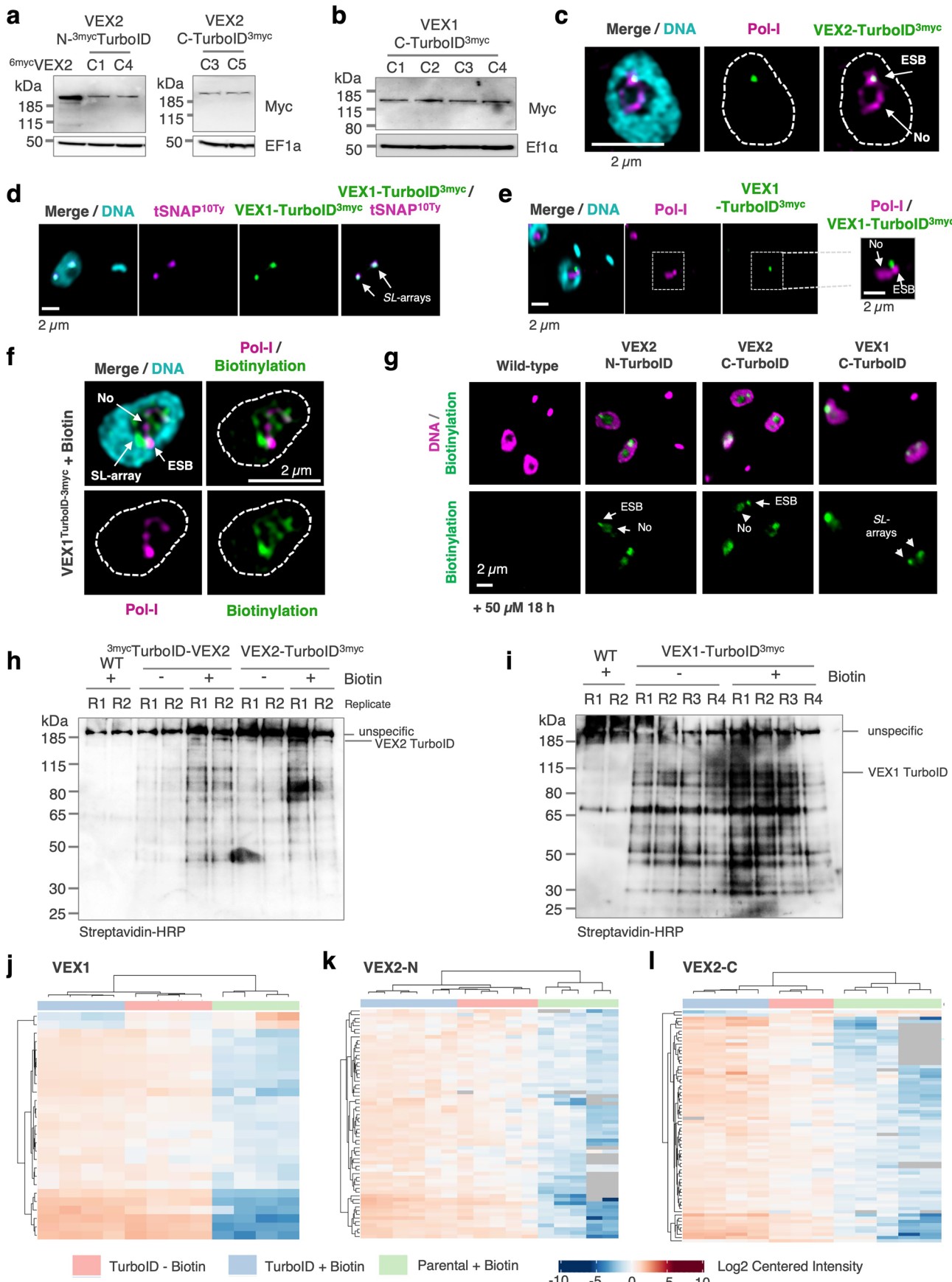

**Extended Data Fig. 1 | See next page for caption.**

**Extended Data Fig. 1 | Characterisation and validation of VEX-TurboID cell lines. a-b**, protein blotting analysis of whole cell extracts of [3myc]TurboID-VEX2 / VEX2-TurboID[3myc] (**a**) and VEX1-TurboID[3myc] (**b**) cell lines. An anti-myc antibody was used; EF1α, loading-control. *C* means clone x. **c-g**, fluorescence microscopy analysis of VEX2-TurboID[3myc] / Pol-I (**c**), VEX1-TurboID[3myc] / tSNAP[10Ty] (**d**), VEX1-TurboID[3myc] / Pol-I (**e**), Biotinylation / Pol-I in a VEX1-TurboID[3myc] cell line (**f**) and Biotinylation in [3myc]TurboID-VEX2 / VEX2-TurboID[3myc] / VEX1-TurboID[3myc] cell lines (**g**). Biotinylated material was detected using Streptavidin-A488. Images were acquired using a Zeiss LSM980 Airyscan 2 (**c/f**) or a Zeiss AxioObserver (**d/e/g**) and correspond to 3D projections by brightest intensity of 0.1 μm stacks. ESB, expression-site body; No, nucleolus. DNA was stained with DAPI (cyan or purple); scale bars: 2 μm. **h-i**, protein blotting analysis of immunoprecipitation samples of biotinylated material using streptavidin beads from [3myc]TurboID-VEX2 and VEX2-TurboID[3myc] (**h**) and VEX1-TurboID[3myc] (**i**) cell lines treated or untreated with biotin. Streptavidin-HRP was used for detection. In **a-i**, the data are representative of at least two independent experiments. **j-l**, heatmaps presenting relative MS intensities across replicates and sample groups, for proteins called as significantly enriched in TurboID (-/+ Biotin) (**j** = VEX1, **k** = VEX2-N, **l** = VEX2-C), relative to Parental (+Biotin) samples (Log2FC > 1.5; p.adj < 0.05). Significance was assessed via two-sided moderated t-tests with Benjamini-Hochberg FDR correction (limma). Heatmaps were produced using FragPipe-Analyst. Biotin was applied at 50 μM for 30 min (**f**) or 18 h (**g-l**).

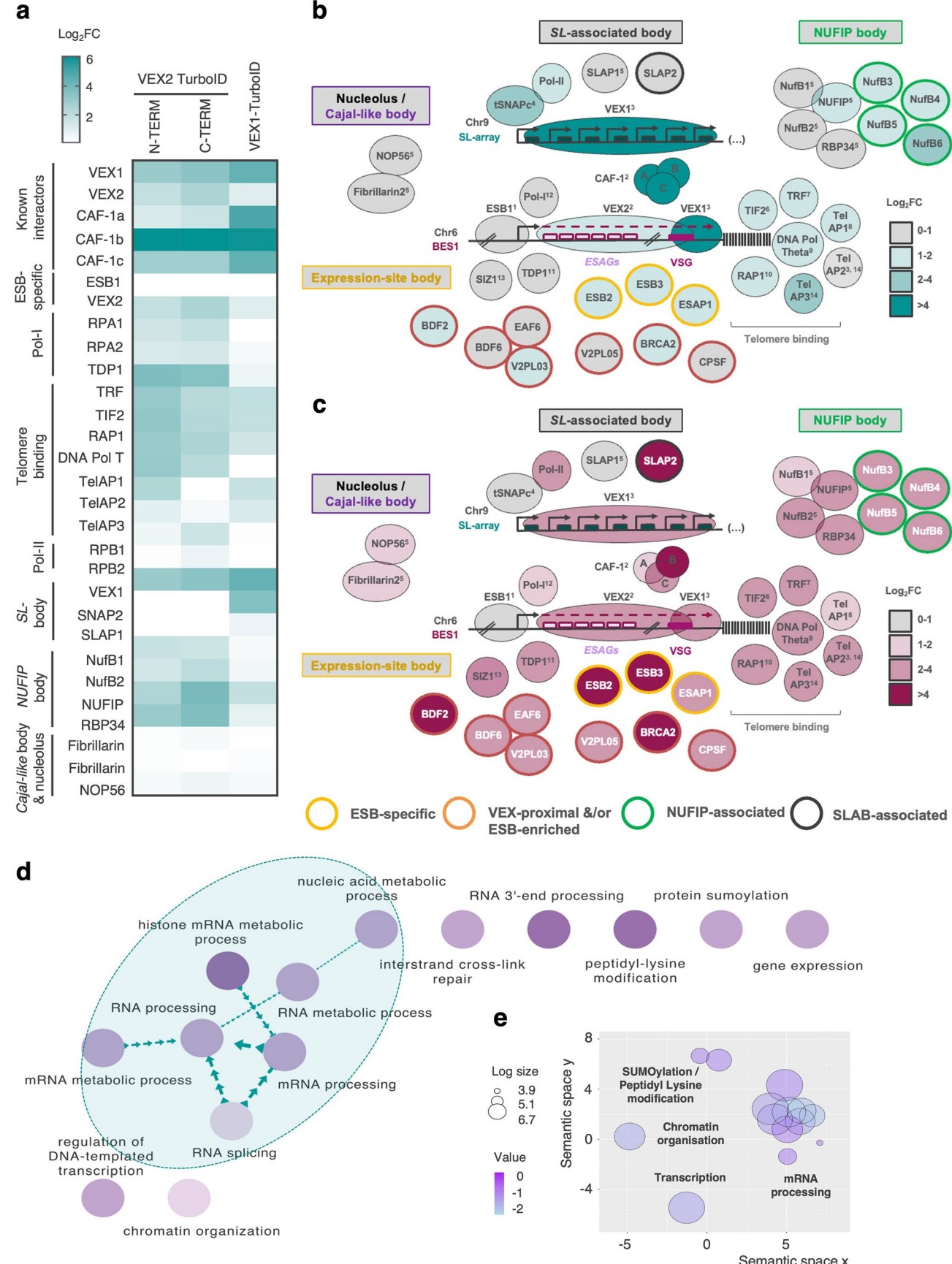

**Extended Data Fig. 2 | See next page for caption.**

**Extended Data Fig. 2 | VEX1 and VEX2 PL-MS captured most known components of the ESB and spatially proximal bodies and identified new components. a**, heatmap depicting the Log$_2$FC for ESB, SLAB, Cajal-body and NUFIP body associated factors in the $^{3myc}$TurboID-VEX2, VEX2-TurboID$^{3myc}$ and VEX1-TurboID$^{3myc}$ datasets. Log$_2$FC represents the fold change in protein abundance between the cell lines where VEX1/VEX2 were fused with TurboID and the parental line, both treated with 50 μM of biotin for 18 h. **b-c**, cartoon depicting the assembly of nuclear bodies that sustain *VSG* expression in *T. brucei* bloodstream forms. All known and newly identified factors are depicted and coloured according to their Log$_2$FC in the VEX1-TurboID$^{3myc}$ (**b**) and $^{3myc}$TurboID-VEX2 / VEX2-TurboID$^{3myc}$ (**c**) datasets. Newly identified factors are outlined in yellow, orange, green or dark grey depending on their localisation. VEX1, VEX2 and ESB1 positions on the DNA are based on available ChIP-Seq data (Faria et al., 2019 & 2023; Lopez-Escobar et al., 2022). BDF6, EAF6 and V2PL03 (Tb927.10.14190) are known to interact (Staneva et al., 2021). For the molecules previously associated with the ESB, or surrounding bodies, the superscript numbers indicate the corresponding reference as follows: 1. López-Escobar et al., 2022, PMID: 35879525. 2. Faria et al., 2019, PMID: 31289266. 3. Glover et al., 2016, PMID: 27226299. 4. Das et al., 2005, PMID: 16055739. 5. Budzak et al., 2022, PMID: 35013170. 6. Jehi et al., 2014, PMID: 24810301. 7. Jehi et al., 2016, PMID: 27258069. 8. Reis et al., 2018, PMID: 29385523. 9. Leal et al., 2020, PMID: 32890403. 10. Yang et al., 2009, PMID: 19345190. 11. Narayanan and Rudenko, 2013, PMID: 23361461. 12. Navarro and Gull, 2001, PMID: 11742402. 13. López-Farfán, Bart et al. 2014, PMID: 25474309. 14. Weisert et al., 2024, PMID: 39681615. **d-e**, GO terms analysis of the 45 proteins significantly enriched in the VEX-TurboID 'plus biotin' samples that have not been previously associated with the ESB or surrounding nuclear bodies. **d**, protein network analysis based on biological process. **e**, similar GO terms containing functionally-related proteins cluster proximally. The size and colour of the bubbles were derived from the number of proteins in the VEX interactome that cluster in each GO term and the corresponding *p* value, respectively.

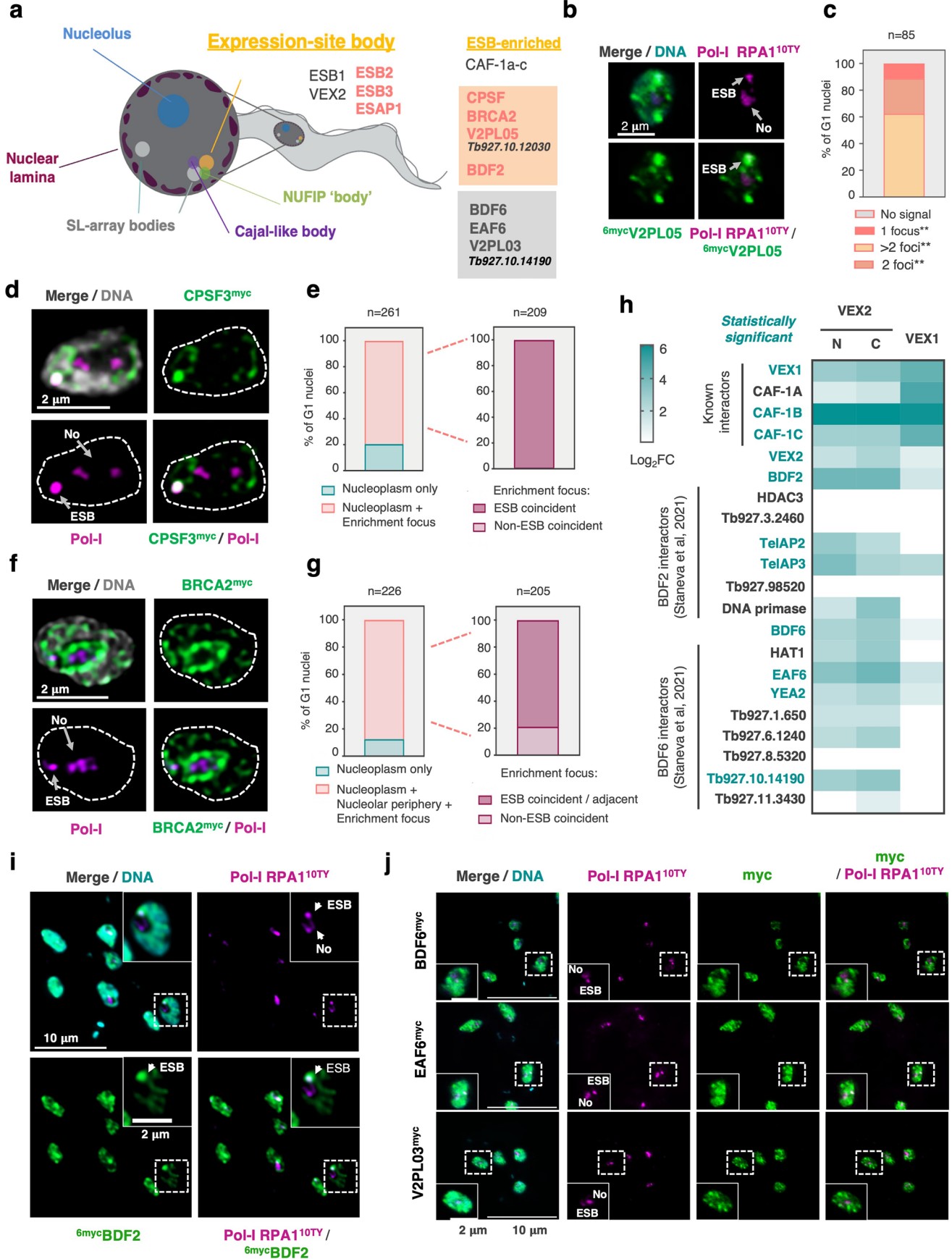

**Extended Data Fig. 3 | See next page for caption.**

**Extended Data Fig. 3 | VEX proximity labelling identified nuclear factors that are significantly enriched at the ESB, including Tb927.10.12030, CPSF, BRCA2 and BDF2. a**, cartoon summarising the main nuclear compartments in *T. brucei* bloodstream forms and highlighting the newly identified ESB-specific or ESB-enriched factors in salmon. BDF6, EAF6 and V2PL03, despite being found in proximity to VEX-proteins, did not evidently accumulate at the ESB. **b-g / i-j**, fluorescence microscopy analysis of $^{6myc}$V2PL05 (Tb927.10.12030) (**b/c**), CPSF3$^{12myc}$ (**d/e**), BRCA2$^{12myc}$ (**f/g**), $^{6myc}$BDF2 (**i**), BDF6$^{12myc}$, EAF6$^{12myc}$ and V2PL03$^{12myc}$ (Tb927.10.14190) (**j**) and Pol-I RPA1$^{10ty}$ in bloodstream forms. **c** shows the % of G1 nuclei with 1, 2 or >2 $^{6myc}$V2PL05 foci. The asterisk indicates that additional nucleoplasmic signal can be detected. **e/g** show % of G1 nuclei where a major CPSF3 or BRCA2 enrichment focus is ESB coincident or not. For more details, see section 'Microscopy and image analysis'. **c/e/g**, the graphs depict mean values of two biological replicates; >85 G1 cells per condition were analysed. The images in **b/i/j** were acquired using a Zeiss AxioObserver, whereas the images in **d/f** were acquired using a Zeiss LSM980 Airyscan 2 and a Zeiss Elyra 7, respectively. All correspond to 3D projections by brightest intensity of 0.1 µm stacks. DNA was stained with DAPI (cyan or grey); scale bars: 2 µm or 10 µm (as indicated). ESB, expression-site body; No, nucleolus. In **i-j**, the images are representative of two independent experiments. **h**, heatmap depicting the Log$_2$FC for BDF2 and BDF6 interactors in the $^{3myc}$TurboID-VEX2, VEX2-TurboID$^{3myc}$ and VEX1-TurboID$^{3myc}$ datasets. Log$_2$FC represents the fold change in protein abundance between the cell lines where VEX1/VEX2 were fused with TurboID and the parental line, both treated with 50 µM of biotin for 18 h. Schematic in **a** created in BioRender; Faria, J. https://BioRender.com/6g770jr (2026).

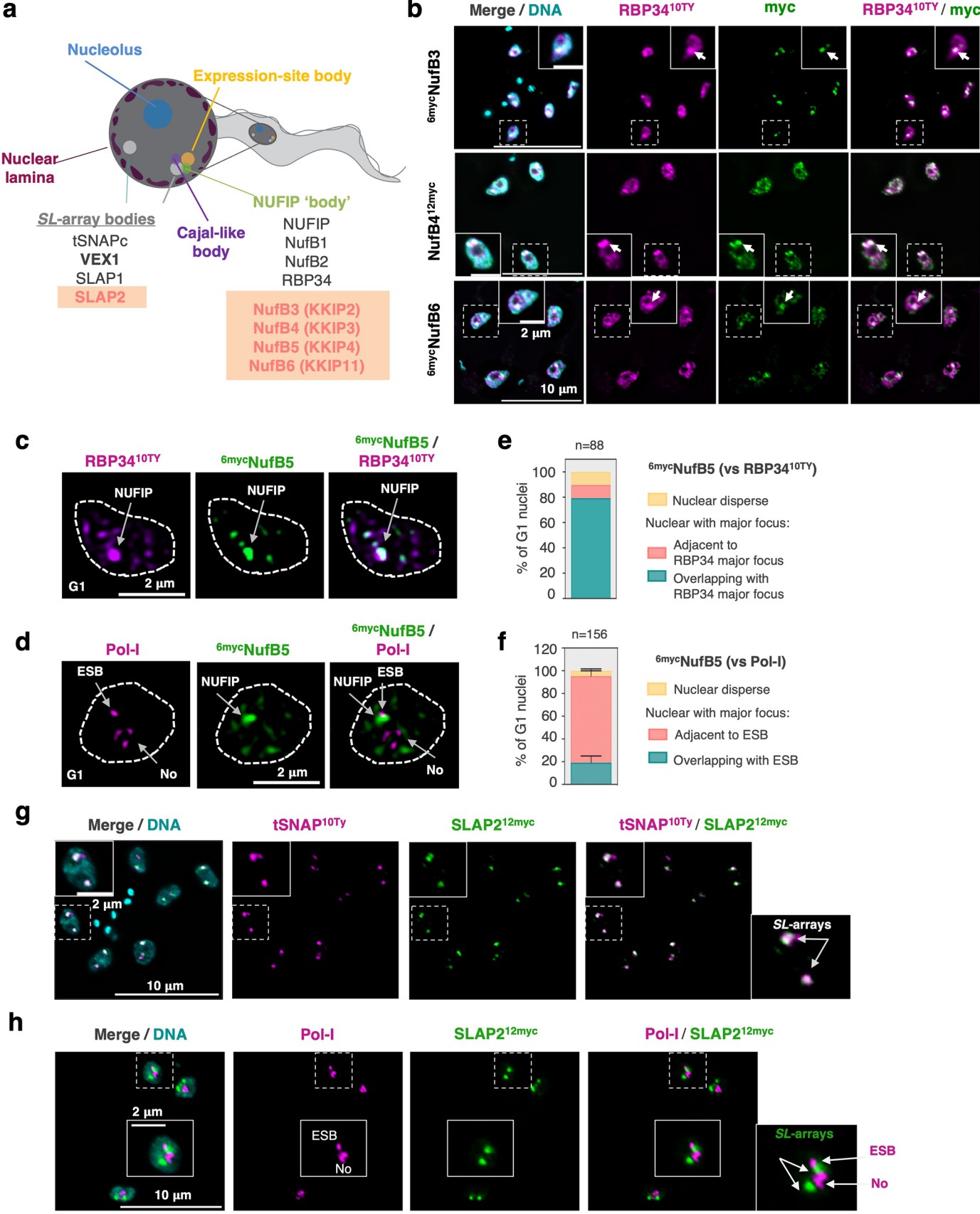

**Extended Data Fig. 4 | See next page for caption.**

**Extended Data Fig. 4 | VEX proximity labelling identified new SLAB and NUFIP body components. a**, cartoon summarising the main nuclear compartments in *T. brucei* bloodstream forms and highlighting the newly identified SLAB and NUFIP-body components in salmon. **b-f**, fluorescence microscopy analysis of $^{6myc}$NufB3 (**b**), NufB4$^{12myc}$ (**b**), $^{6myc}$NufB5 (**c/d**), $^{6myc}$NufB6 (**b**) and Pol-I RPA1$^{10ty}$ or RBP34$^{10ty}$ in bloodstream forms. **e/f**, the graphs depict mean values of two (**e**) or three (**f**) biological replicates; >88 G1 cells per condition were analysed; in **f**, error bars correspond to SD. For more details, see section 'Microscopy and image analysis'. **g-h**, fluorescence microscopy analysis of SLAP2$^{12myc}$ and tSNAP$^{10ty}$ (**g**) or Pol-I RPA1$^{10ty}$ (**h**) in bloodstream forms. The images in **b/g/h** were acquired using a Zeiss AxioObserver, whereas the images in **c/d** were acquired using a Zeiss LSM980 Airyscan 2. All correspond to 3D projections by brightest intensity of 0.1 μm stacks. DNA was stained with DAPI (cyan or grey); scale bars: 2 μm or 10 μm (as indicated). ESB, expression-site body; No, nucleolus. In **b/g-h**, the images are representative of two independent experiments. Schematic in **a** created in BioRender; Faria, J. https://BioRender.com/owzempb (2026).

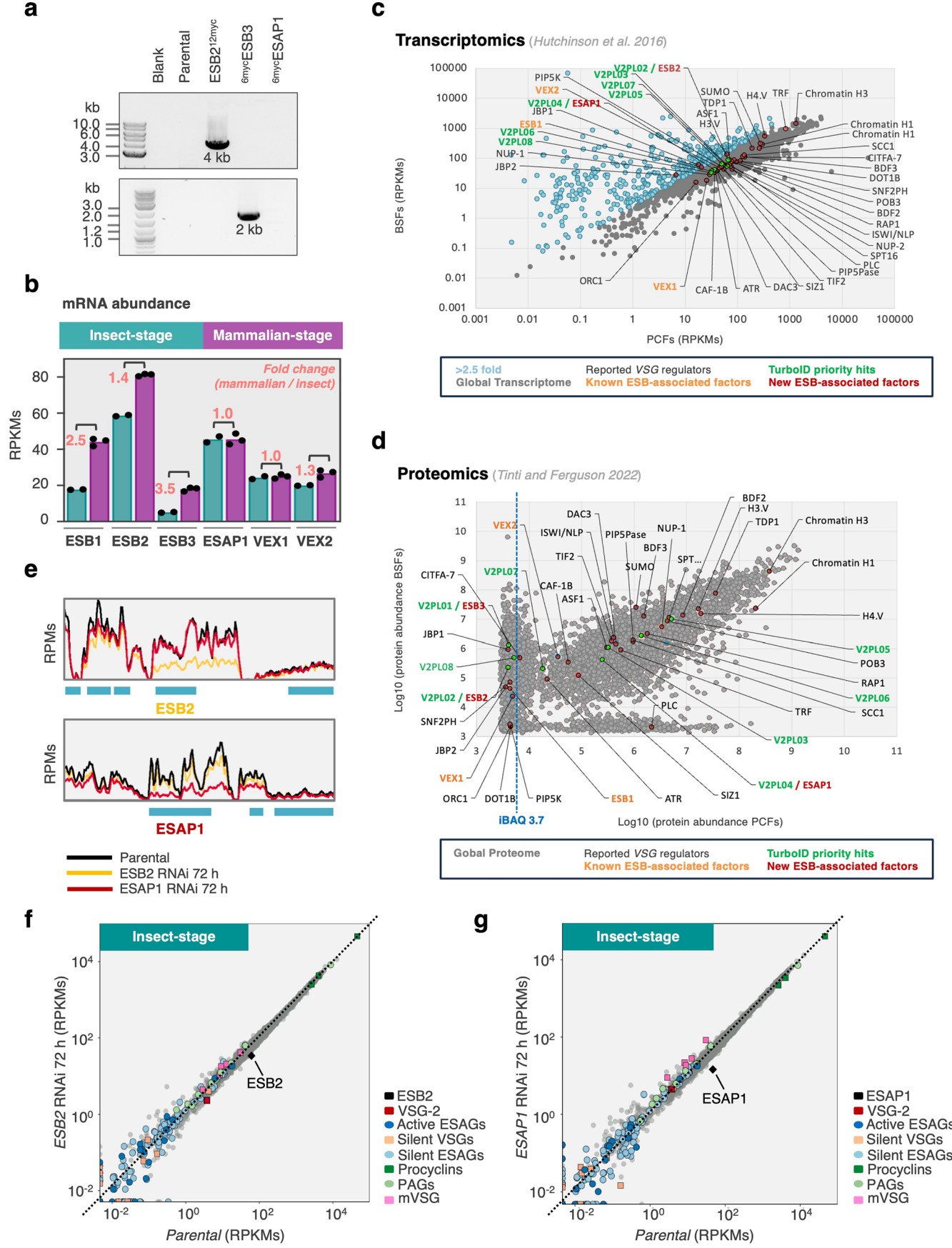

**Extended Data Fig. 5 | See next page for caption.**

**Extended Data Fig. 5 | ESB2, ESB3 and ESAP1 are developmentally regulated. a**, PCR analysis of the integration of ESB2 and ESB3 tagging constructs in the correct genomic location in procyclic forms (n = 1). **b**, mRNA abundance expressed in reads per kilobase per million (RPKMs) in bloodstream (n = 3, purple) and procyclic (n = 2, cyan) forms. Data was obtained from RNA-Seq analysis of both developmental stages of the parasite and correspond to mean values of *n* biological replicates. **c-d**, Scatter plots represent either mRNA abundance (**c**) or protein abundance (**d**) in procyclic forms (x axis) and bloodstream forms (y axis). RNA-Seq data Hutchinson et al., 2016; proteomics data from Tinti & Ferguson, 2022. Labelled in grey are all the factors that have somehow been implicated in *VSG* regulation; in orange those that had previously been shown to localise to the ESB and in red the new ones identified in this study. GeneIDs for all TurboID priority hits can be found in the methods section. The threshold of protein expression in PCFs was defined as intensity-based absolute quantification (iBAQ) > 3.7 (dashed blue line). **e-g**, RNA-Seq analysis of ESB2 and ESAP1 knockdown in procyclic forms at 72 h post-induction. Reads per kilobase per million (RPKMs) are averages of two biological replicates. **e** zooms in on the ESB2 and ESAP1 coding loci to show successful transcript depletion in the knockdowns. RPMs, reads per million.

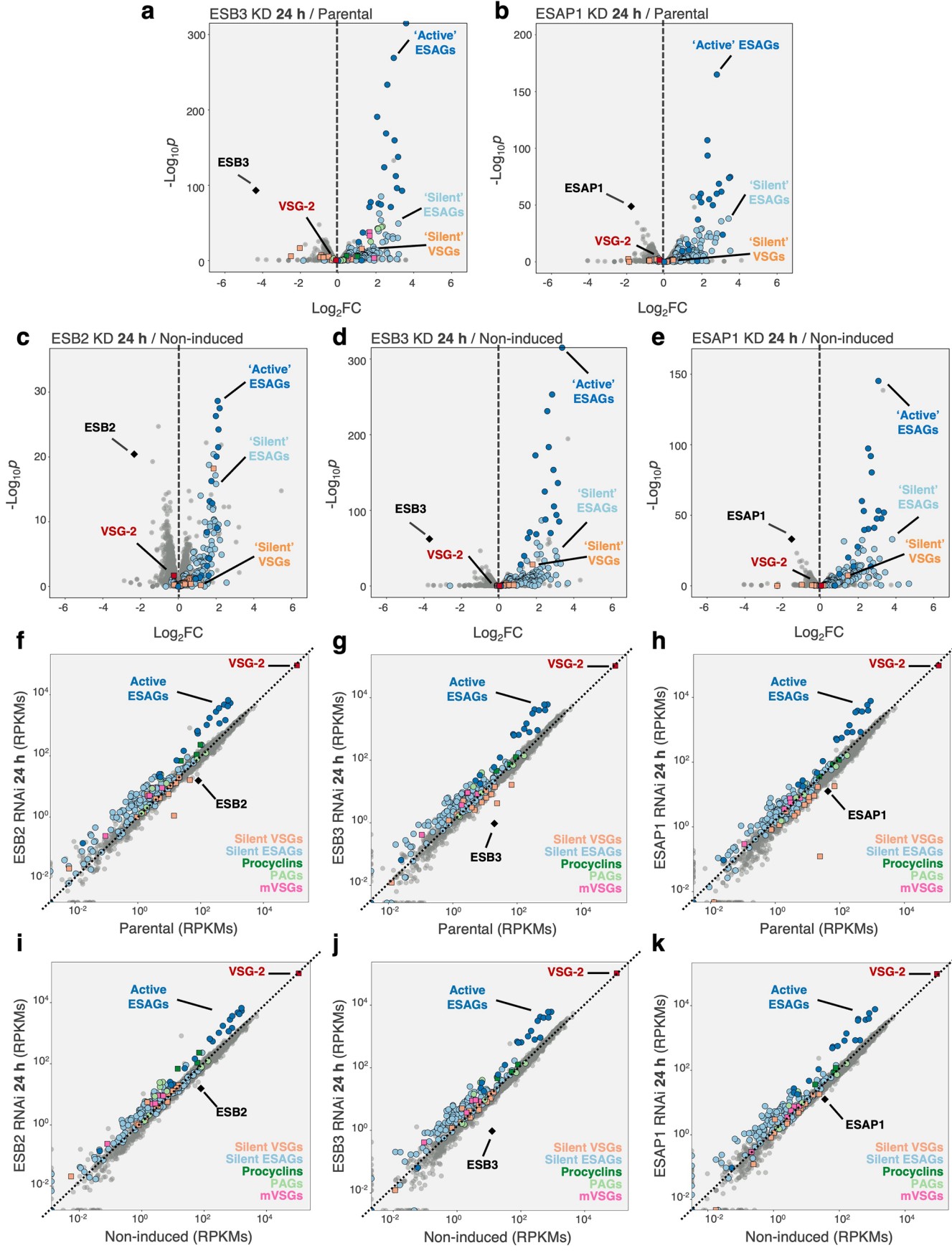

**Extended Data Fig. 6 | See next page for caption.**

**Extended Data Fig. 6 | RNA-Seq analysis of ESB2, ESB3 and ESAP1 RNAi in BSFs.**
**a-k**, RNA-Seq analysis of ESB2 (n = 5), ESB3 (n = 3) and ESAP1 (n = 3) knockdowns in bloodstream forms (BSFs) at 24 h post induction, where $n$ values correspond to the number of biological replicates. **a-e**, volcano plots depicting $Log_2FC$ (fold change) between RNAi and parental line (**a-b**) or RNAi induced *versus* non-induced samples (**c-e**) and corresponding statistical significance (-$log_{10}p$). Differential expression determined by two-sided negative binomial generalised linear models (edgeR) with Benjamini-Hochberg FDR correction. The following cohorts are highlighted: 'active' *VSG* (red) and *ESAGs* (darker blue) and 'silent' *VSGs* (orange) and *ESAGs* (lighter blue). **f-k**, scatter plots comparing induced RNAi samples with the parental line (**f-h**) or the non-induced samples (**i-k**). RPKMs, reads per kilobase per million. The following cohorts are highlighted: 'active' *VSG* (red) and *ESAGs* (darker blue); 'silent' *VSGs* (orange) and *ESAGs* (lighter blue), metacyclic *VSGs* (mVSGs, pink), procyclins (darker green), procyclin associated genes (PAGs, lighter green).

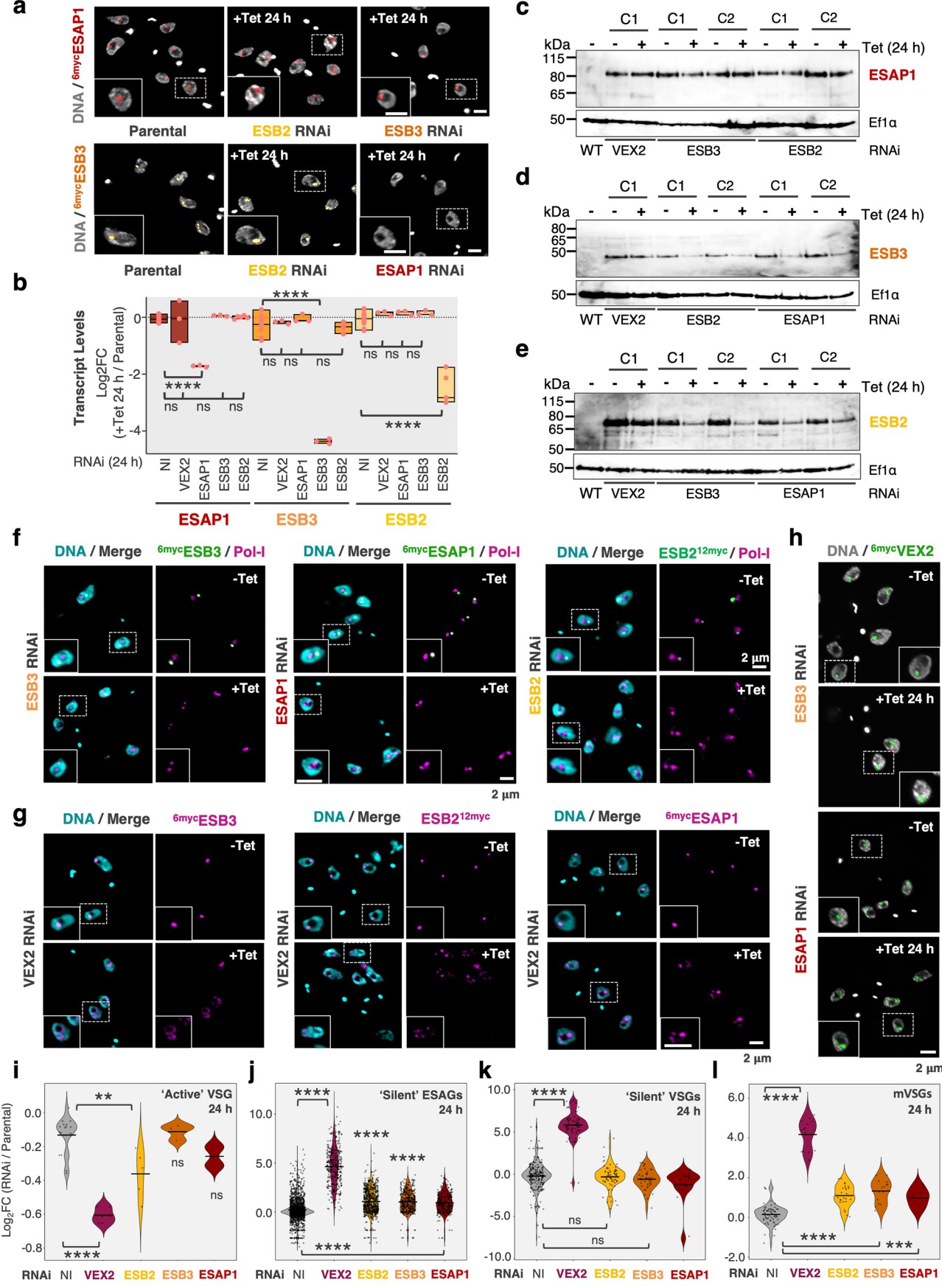

**Extended Data Fig. 7 | See next page for caption.**

**Extended Data Fig. 7 | ESB2, ESB3, ESAP1, VEX2 and Pol-I co-dependencies regarding subcellular localisation as well as mRNA / protein abundance.**
**a**, Fluorescence microscopy analysis of $^{6myc}$ESB3 localisation following ESB2 and ESAP1 RNAi and $^{6myc}$ESAP1 localisation following ESB2 and ESB3 RNAi. **b**, ESB2, ESB3 and ESAP1 mRNA levels following VEX2, ESB2, ESB3 and ESAP1 RNAi. The values were derived from RNA-Seq data (VEX2, ESB3 & ESAP1 RNAi: 3 biological replicates; ESB2 RNAi: 5 biological replicates). Bars span between minimum and maximum values; the centre line is the median; all datapoints are represented. **c-e**, protein blotting analysis of $^{6myc}$ESAP1 (**c**), $^{6myc}$ESB3 (**d**) and ESB2$^{12myc}$ (**e**) levels following VEX2, ESB2, ESB3 or ESAP1 RNAi at 24 h post induction. C1, clone 1; C2, clone 2. EF1α, loading-control. **f-h**, fluorescence microscopy analysis of Pol-I localisation following ESB2, ESB3 and ESAP1 knockdown (**f**), ESB2$^{12myc}$, $^{6myc}$ESB3 and $^{6myc}$ESAP1 localisation following VEX2 knockdown (**g**) and $^{6myc}$VEX2 localisation following ESB3 and ESAP1 knockdown (**h**). **a/f-h**, all analyses were

performed at 24 h post induction. The images were acquired using a Zeiss AxioObserver and correspond to 3D projections by brightest intensity of 0.1 μm stacks and are representative of two biological replicates. DNA was stained with DAPI (grey or cyan); scale bars: 2 μm. The blots in **c-e** are representative of 3 biological replicates (only two are depicted). **i-l**, violin plots depicting a comparative transcriptomic analysis between VEX2 (n = 3), ESB2 (n = 5), ESB3 (n = 3) and ESAP1 (n = 3) RNAi cell lines with a focus on Pol-I transcribed gene cohorts: active-*VSG* (**i**), 'silent' *ESAGs* (**j**), 'silent' *VSGs* (**k**) and metacyclic *VSGs* (*mVSGs*, **i**); *n* values correspond to biological replicates. Log$_2$FC, fold change in transcript abundance between RNAi (24 h post induction) and the parental cell line. The violins span between minimum and maximum values, centre lines correspond to the mean, all datapoints are shown. In **b/i-l**, significance was determined using One-Way ANOVA followed by Tukey's multiple comparisons test; ns, non-significant; ** $p < 0.01$; *** $p < 0.001$; **** $p < 0.0001$.

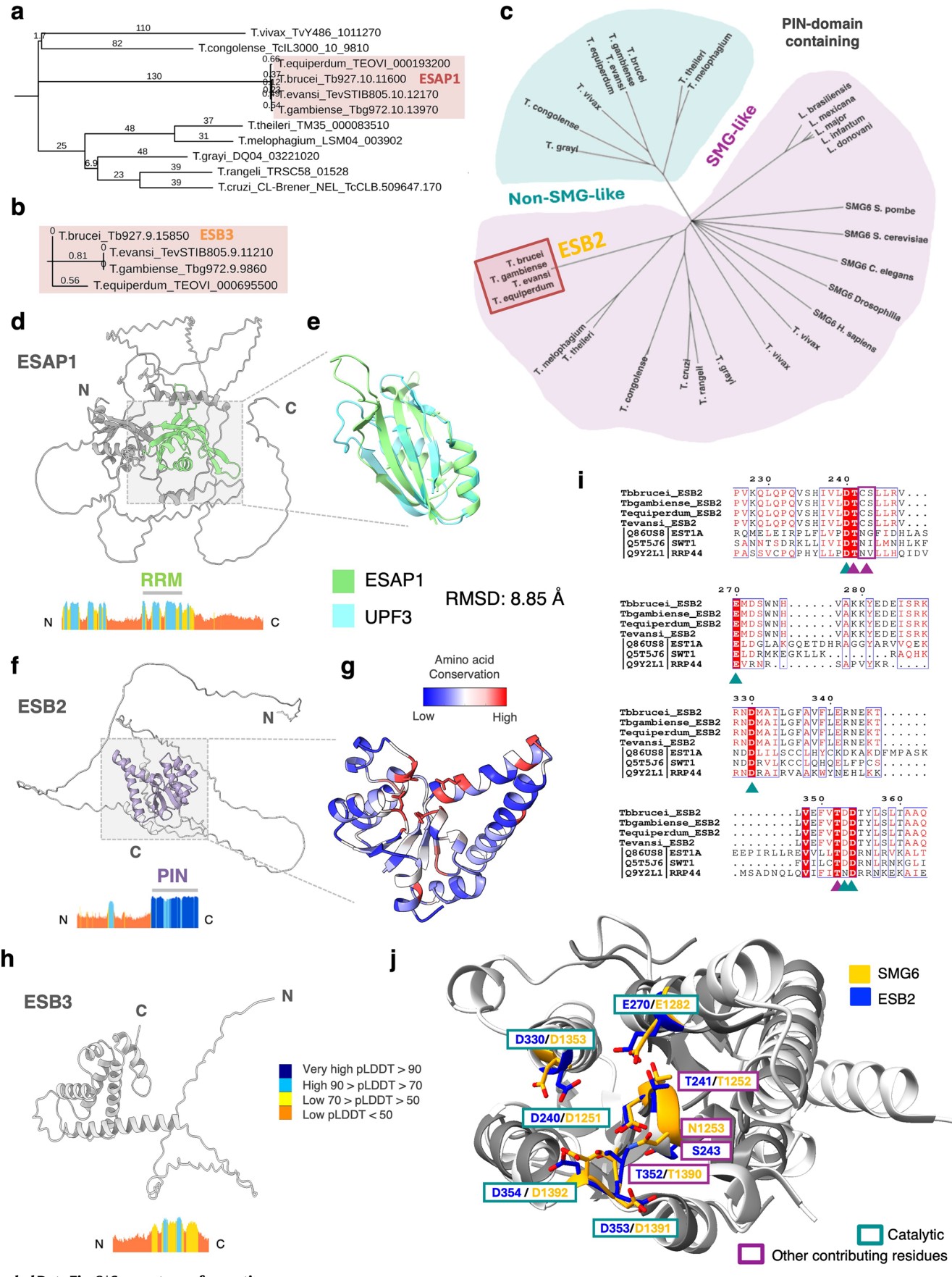

**Extended Data Fig. 8 | See next page for caption.**

**Extended Data Fig. 8 | ESB2, ESB3 and ESAP1 phylogenetic and structural prediction analyses. a-c**, phylogenetic analysis of ESAP1 (**a**), ESB3 (**b**) and ESB2 (**c**) conducted using TreeViewer. Highlighted in light red are trypanosome species that contain *ESAGs*. **d/f/h** depict 3D structural predictions of ESAP1 (RRM-like domain highlighted in green), ESB2 (PIN domain highlighted in purple) and ESB3. pLDDT plots accompany each model. **e**, Structural overlay between ESAP1 and UPF3 (PDB ID 7NWU). Root mean square deviation (RMSD) quantifies the average distance between corresponding atoms in two superimposed structures. **g** shows a structural prediction of ESB2 PIN domain overlayed with the structure of SMG6 PIN domain (PDB ID 2HWW) coloured by degree of amino acid conservation. **i-j**, conservation of the catalytic (cyan) and other important residues (purple) within the PIN domain of ESB2 compared to human SMG6. The first conserved, acidic residue, which is invariably an aspartate occurs at the end of β1 before the first helix (D240 / D1251). This is followed by another important residue, which always occurs after one turn of the first helix (α1) and often is an asparagine or a serine (Ser243 / N1253). The third residue of the active site (2nd acidic residue) occurs in the helix (α2) following β2 and is always a glutamate (E270 / E1282); structurally is the most distant of the core acidic residues. Its side chain is held in place by the residue located between the first two active site residues, which is often a threonine (T241 / T1252). The fourth residue (3rd acidic residue) is always an aspartate (D330 / D1353) and it is located on the other side of the active site at the end of α3. Finally, the active site contains one or sometimes two additional aspartic acid residues, which both occur in the loop following β4 and in the beginning following helix (D353 / D1391; D354 / D1392). Another potentially important residue in this region is often either a serine or a threonine (T352 / T1390).

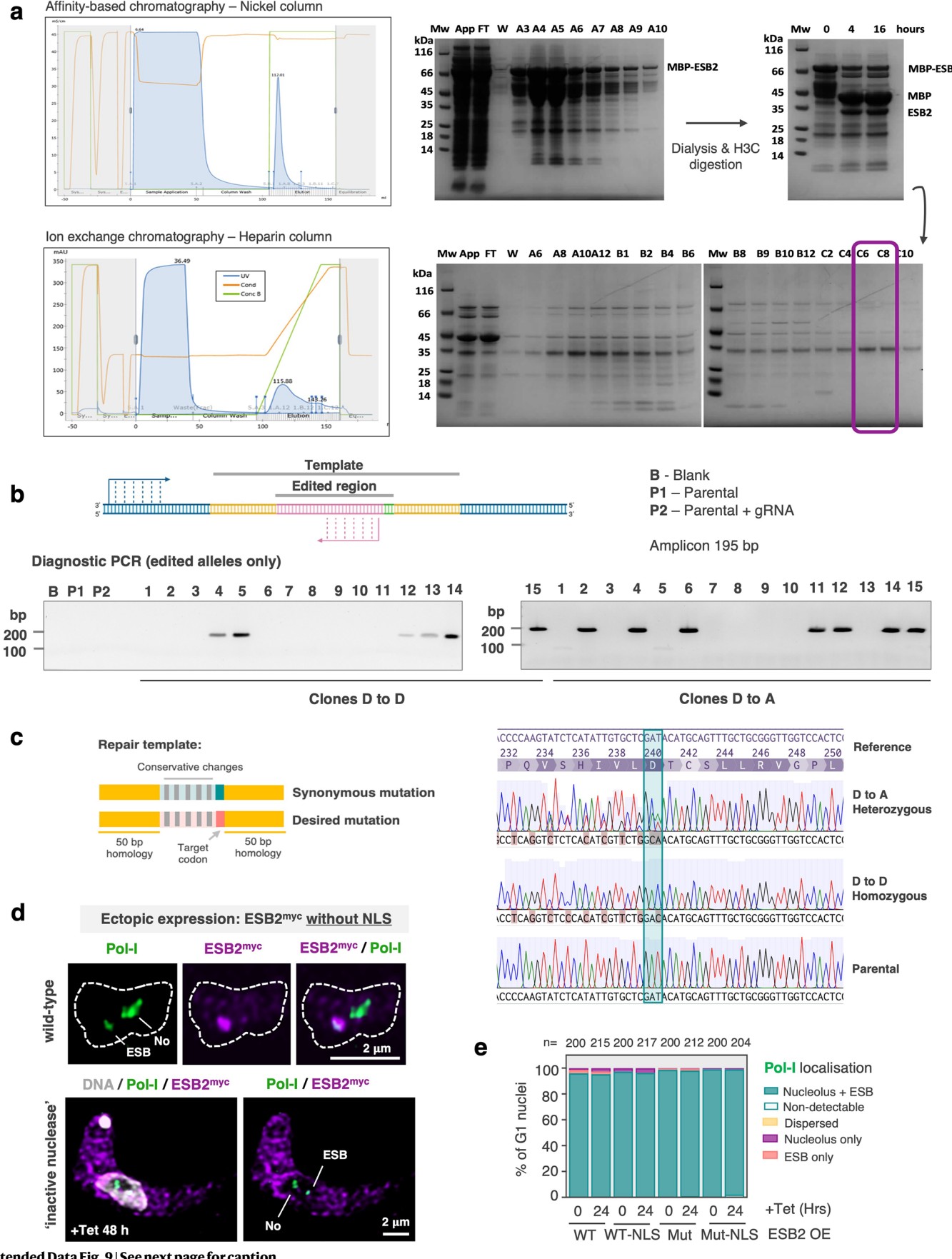

**Extended Data Fig. 9 | See next page for caption.**

**Extended Data Fig. 9 | ESB2 nuclease activity is required for parasite survival and localisation to the ESB. a**, ESB2 nuclease domain (aa 95-C) expression and purification from *E. coli* BL21 DE3 – chromatograms and an SDS-PAGE analysis of the resulting fractions are depicted. The domain was fused with a His-MBP tag at the N-terminus and affinity purified using nickel affinity chromatography. Fractions were pooled, dialysed and treated with H3C protease to cleave off the His-MBP tag. After 16 h of incubation with H3C, the samples were further purified using a Heparin column (a form of ion exchange chromatography). The fractions highlighted in the purple box were pooled, concentrated, analysed by MS and mass photometry and then used in the *in vitro* nuclease activity assays. For more details, check the methods section. MW, molecular weight; App, lysate; FT, flowthrough; W, wash; A/B/Cx correspond to different chromatography fractions. **b-c**, CRISPR/Cas9 mediated precision editing of ESB2 D240. Mutants carrying the mutation from aspartic acid to alanine were always heterozygous, however successful double allele editing was achieved when using a repair template containing a synonymous mutation. Both repair templates contained conservative changes so that edited alleles could be readily distinguished from wild-type alleles (schematics in **b**). 15 clones were screened for both synonymous and non-synonymous mutations, respectively. **b**, diagnostic PCR analysis. The Sanger sequencing profiles depicted in **c** are representative of different clones. In **a-c**, the data are representative of two independent experiments. **d-e**, fluorescence microscopy analysis of Pol-I and ectopically expressed ESB2[12myc] (tetracycline-inducible expression). Wild-type ESB2[12myc] or a variant containing 3-point mutations (D240A, D330A, D353A) that render the nuclease inactive, with or without an La-NLS sequence, were analysed. Images were acquired using a Zeiss LSM980 Airyscan 2 and correspond to 3D projections by brightest intensity of 0.1 μm stacks. ESB, expression-site body; No, nucleolus. DNA was stained with DAPI (grey); scale bars: 2 μm. The stacked graph (**e**) depicts mean values of two biological replicates; >100 G1 cells per condition were analysed. Diagram in **b** created in BioRender; Faria, J. https://BioRender.com/slijrkq (2026).

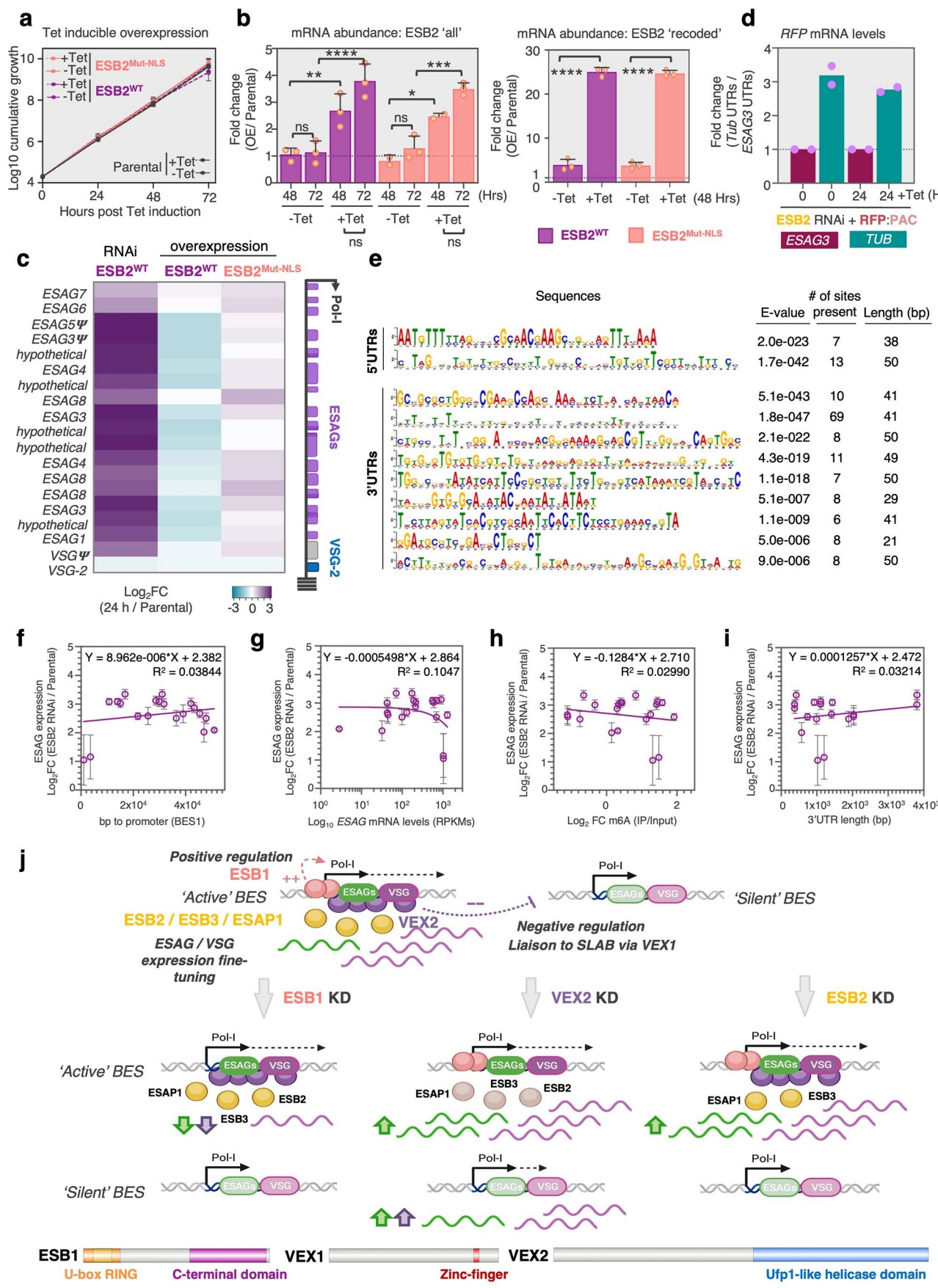

**Extended Data Fig. 10 | See next page for caption.**

**Extended Data Fig. 10 | VEX2 and ESB2 are *VSG* and *ESAG* negative regulators, respectively. a-c**, tetracycline inducible overexpression of untagged ESB2 in bloodstream forms. **a**, cumulative growth following tetracycline induced ESB2^Mut-NLS (salmon), and ESB2^WT (purple) expression. **b**, determination of *ESB2* mRNA levels by RT-qPCR. The primers annealed outside the recoded region amplifying both endogenously and ectopically expressed *ESB2* (left hand side) or in the recoded region amplifying only ectopically expressed *ESB2* (right hand side). Significance was determined using one-way ANOVA followed by Tukey's multiple comparisons test; ns, non-significant; * $p < 0.05$; ** $p < 0.01$; *** $p < 0.001$; **** $p < 0.0001$. **c**, RNA-Seq analysis of ESB2^WT and ESB2^Mut-NLS overexpression (OE) at 72 h post induction. Heatmap depicting the Log$_2$FC within the active-ES (BES1) for induced samples of ESB2^WT and ESB2^Mut-NLS overexpression compared to ESB2 RNAi – all normalised against the parental line. From top to bottom, genes are ordered as they appear in BES1 (active-ES). *ESAG2* and *ESAG11* were omitted as their expression could not be detected in all replicates in the parental line when considering uniquely mapped reads. **d**, analysis of cell lines containing an *RFP::PAC* fluorescent reporter at the active *VSG*-ES, between *ESAG8* and *ESAG3* (see schematics in Fig. 6g), before and after ESB2 RNAi. *RFP* mRNA levels were determined by RT-qPCR. Its expression was normalized against actin (housekeeper gene) and displayed as fold change between the *RFP::PAC* reporter flanked by *TUB* (tubulin) UTRs and flanked by *ESAG3* UTRs, for both induced and uninduced samples. In **b/d**, each experiment was conducted using three technical replicates per condition, which were averaged. In **a-d**, data correspond to mean values of two (**d**) or three (**a-c**) biological replicates. **e**, MEME analysis of *ESAG* UTRs from the active *VSG*-ES. The detected motifs appear as extended, low-complexity blocks (>35 bp), more likely to reflect the repetitive nature and shared ancestry of this gene family or even perhaps structural features than discrete consensus sequences indicative of specific nuclease recognition. **f-i**, Log2FC in 'active' *ESAGs* expression comparing ESB2 RNAi (24 h) to parental line *versus* distance to BES1 promoter (**f**), expression level in the parental line (RPKM, reads per kilobase per million) (**g**); enrichment for m6A modification (**h**, data extracted from Viegas et al., 2022; PMID:35355019); and 3' UTR length (**i**). Log2FC data are mean values of three biological replicates; no correlation was found. Error bars in **a-b / f-i** are SD. In **e / i**, UTRs were predicted using the approach described by Tinti & Horn (PMID:40735494); analyses with UTRs predicted using UTRme (PMID:30619487) did not change the main conclusions. **j**, summary of the gene expression profile at both active and silent-ESs following ESB1, VEX2 or ESAP1/ESB3/ESB2 knockdowns. ESB1 is a positive regulator required for transcription at the active *VSG*-ES. VEX2 is an exclusion factor, which prevents activation of silent *VSGs* and therefore a negative regulator. ESAP1 and ESB3 are required for ESB2 recruitment, which ultimately negatively regulates *ESAGs* expression. In the schematics, ESB2, ESB3 and ESAP1 are not coloured in the VEX2 knockdown because they are compartmentalised in a VEX2-dependent manner. The upregulation of 'active' *ESAG* transcripts in the VEX2 RNAi is likely to be indirect because of ESB2 mislocalisation. ESB1, VEX1 and VEX2 domain analysis is provided at the bottom. Schematic in **j** created in BioRender; Faria, J. https://BioRender.com/3uskgmu (2026).

# Reporting Summary

## Statistics

For all statistical analyses, confirm that the following items are present in the figure legend, table legend, main text, or Methods section.

| n/a | Confirmed | |
|---|---|---|
| ☐ | ☒ | The exact sample size (*n*) for each experimental group/condition, given as a discrete number and unit of measurement |
| ☐ | ☒ | A statement on whether measurements were taken from distinct samples or whether the same sample was measured repeatedly |
| ☐ | ☒ | The statistical test(s) used AND whether they are one- or two-sided<br>*Only common tests should be described solely by name; describe more complex techniques in the Methods section.* |
| ☒ | ☐ | A description of all covariates tested |
| ☐ | ☒ | A description of any assumptions or corrections, such as tests of normality and adjustment for multiple comparisons |
| ☐ | ☒ | A full description of the statistical parameters including central tendency (e.g. means) or other basic estimates (e.g. regression coefficient) AND variation (e.g. standard deviation) or associated estimates of uncertainty (e.g. confidence intervals) |
| ☐ | ☒ | For null hypothesis testing, the test statistic (e.g. *F*, *t*, *r*) with confidence intervals, effect sizes, degrees of freedom and *P* value noted<br>*Give P values as exact values whenever suitable.* |
| ☒ | ☐ | For Bayesian analysis, information on the choice of priors and Markov chain Monte Carlo settings |
| ☒ | ☐ | For hierarchical and complex designs, identification of the appropriate level for tests and full reporting of outcomes |
| ☒ | ☐ | Estimates of effect sizes (e.g. Cohen's *d*, Pearson's *r*), indicating how they were calculated |

*Our web collection on statistics for biologists contains articles on many of the points above.*

## Software and code

Policy information about availability of computer code

| Data collection | Software used for data collection (Immunofluorescence analysis):<br>Zeiss ZEN and ZEN PRO software v / https://www.zeiss.com/microscopy/int/products/microscope-software/zen.html |
|---|---|
| Data analysis | SOFTWARE / VERSION / Citation<br>Bowtie2 / v2.5.4 / Langmead & Salzberg 2012 (doi: 10.1038/nmeth.1923)<br>Samtools / v1.21 / Li et al 2009 (doi: 10.1093/bioinformatics/btp352)<br>deeptools2 / v3.5.6 / Ramirez et al 2016 (doi: 10.1093/nar/gkw257)<br>Artemis / v18.2.0 / Rutherford et al 2000 (https://doi.org/10.1093/bioinformatics/16.10.944)<br>RStudio / v4.3.1 / Available at https://github.com/rstudio/rstudio<br>IGV / v 2.18.2 / Robinson et al, 2011 https://igv.org/<br>Bedtools / v2.31.0 / Quinlan and Hall, 2010 / http://bedtools.readthedocs.io/en/latest/<br>Picard tools / v3.4.0 / https://broadinstitute.github.io/picard/<br>Progenesis QI / v2.2 / https://www.nonlinear.com/progenesis/qi/<br>Mascot Daemon / v2.6.1 / https://www.matrixscience.com/daemon.html<br>Graphpad / v10.0 / https://www.graphpad.com/scientific-software/prism/<br>Fiji / v2.9.0 / Schindelin et al 2012 (doi: 10.1038/nmeth.2019)<br>ThermoFisher Connect Platform<br>AlphaFold2 / https://alphafold.ebi.ac.uk/<br>FoldSeek / v10 / https://search.foldseek.com/search<br>ChimeraX / v1.9 / https://www.cgl.ucsf.edu/chimerax/<br>InkScape / v1.3 / https://inkscape.org |

```
BioRender
Cytoscape       / v3.10.3  / https://cytoscape.org/
Revigo          / v1.8.1
FragPipe-Analyst / http://fragpipe-analyst.nesvilab.org

Custom code:
All custom scripts are publicly available at Zendodo (https://doi.org/10.5281/zenodo.15357090 ).
```

For manuscripts utilizing custom algorithms or software that are central to the research but not yet described in published literature, software must be made available to editors and reviewers. We strongly encourage code deposition in a community repository (e.g. GitHub). See the Nature Portfolio guidelines for submitting code & software for further information.

## Data

Policy information about availability of data

All manuscripts must include a data availability statement. This statement should provide the following information, where applicable:
- Accession codes, unique identifiers, or web links for publicly available datasets
- A description of any restrictions on data availability
- For clinical datasets or third party data, please ensure that the statement adheres to our policy

Mass spectrometry data sets and associated results files are referenced in ProteomeXchange (PXD063534) and are available to download from MassIVE (MSV000097776) [doi:10.25345/C5JS9HM41], currently password protected: username and password provided in the manuscript. Data will be fully available to the public upon publication.
RNA sequencing datasets have been deposited in the European Nucleotide Archive (ENA) under primary accession number PRJEB89423. Processed data and results are available as supplementary data.

## Research involving human participants, their data, or biological material

Policy information about studies with human participants or human data. See also policy information about sex, gender (identity/presentation), and sexual orientation and race, ethnicity and racism.

| Reporting on sex and gender | N/A |
|---|---|
| Reporting on race, ethnicity, or other socially relevant groupings | N/A |
| Population characteristics | N/A |
| Recruitment | N/A |
| Ethics oversight | N/A |

Note that full information on the approval of the study protocol must also be provided in the manuscript.

# Field-specific reporting

Please select the one below that is the best fit for your research. If you are not sure, read the appropriate sections before making your selection.

☒ Life sciences        ☐ Behavioural & social sciences        ☐ Ecological, evolutionary & environmental sciences

For a reference copy of the document with all sections, see nature.com/documents/nr-reporting-summary-flat.pdf

# Life sciences study design

All studies must disclose on these points even when the disclosure is negative.

| Sample size | Sample size was not statistically predetermined for the individual experiments. The sample size is appropriate as we were able to robustly detect differences as low as % (versus control) for different biological replicates and independent experiments. |
|---|---|
| Data exclusions | N/A |
| Replication | All attempts of replication were successfull Mass spectrometry analysis following proximity labelling experiments was performed using 4 or 6 technical replicates. RNAseq experiments were performed using two to five biological replicates for each cell line. For microscopy analysis, at least 2 biological replicates were used and typically >100 cells were analysed per condition. For RT-qPCR analysis, at least 2 biological replicates were used per cell line; technical triplicates were used for each individual condition and were averaged. |
| Randomization | No randomisation applied. |

| Blinding | No blinding applied. For RNA-Seq and Proteomics data, blinding was not applicable because data acquisition and analysis were performed using automated algorithms that do not require subjective interpretation. The results are based on quantitative measurements where the investigator's knowledge of the sample group could not influence the raw output. The only analyses where a certain degree of 'subjectivity' would be expected were some of the quantitative microscopic analyses, but we have mitigated for that by having at least two investigators independently looking at the images. |
|---|---|

# Reporting for specific materials, systems and methods

We require information from authors about some types of materials, experimental systems and methods used in many studies. Here, indicate whether each material, system or method listed is relevant to your study. If you are not sure if a list item applies to your research, read the appropriate section before selecting a response.

## Materials & experimental systems

| n/a | Involved in the study |
|---|---|
| ☐ | ☒ Antibodies |
| ☐ | ☒ Eukaryotic cell lines |
| ☒ | ☐ Palaeontology and archaeology |
| ☒ | ☐ Animals and other organisms |
| ☒ | ☐ Clinical data |
| ☒ | ☐ Dual use research of concern |
| ☒ | ☐ Plants |

## Methods

| n/a | Involved in the study |
|---|---|
| ☒ | ☐ ChIP-seq |
| ☒ | ☐ Flow cytometry |
| ☒ | ☐ MRI-based neuroimaging |

## Antibodies

| Antibodies used | ANTIBODY / SOURCE / IDENTIFIER<br>Mouse anti-Myc 9B11 / New England Biolabs / Cat# 2276S<br>Mouse anti-Myc 4A6 / Merck-Millipore / Cat# 05-724 RRID:AB_568800<br>Rabbit polyclonal anti-GFP / Life Technologies (Invitrogen) / Cat# A-6455 RRID:AB_221570<br>Mouse monoclonal anti-HA / Sigma-Aldrich / Cat# H9658<br>Mouse anti-EF1α CBP-KK1 / Merck-Millipore / Cat# 05-235 RRID:AB_309663<br>Mouse anti-Ty / ThermoFisher Scientific / Cat# A-6455 RRID:AB_221570<br>Rat anti-VSG-2 / Prof. George Cross, Rockefeller University / Hoek and Cross, 1999 (doi.org/10.1006/expr.1998.4369)<br>Rabbit anti-VSG-2 / Prof. George Cross, Rockefeller University / Hoek and Cross, 1999 (doi.org/10.1006/expr.1998.4369)<br>Rabbit anti-VSG-6 / Prof. George Cross, Rockefeller University / Hoek and Cross, 1999 (doi.org/10.1006/expr.1998.4369)<br>Mouse anti-EP procyclin / VWBio-Cedarlene / Cat# CLP001AP RRID:AB_10060662<br>Rabbit anti-Pol-I / In-house / Glover et al, 2016 (doi: 10.1073/pnas.1600344113)<br>Rabbit anti-VEX2 / In-house / Faria et al, 2019 (doi: 10.1038/s41467-019-10823-8)<br>Goat anti-mouse Alexa 488 / ThermoFisher Scientific / Cat# A-11001 RRID:AB_2534069<br>Goat anti-rabbit Alexa 488 / ThermoFisher Scientific / Cat# A-11034 RRID:AB_2576217<br>Goat anti-mouse Alexa555+ / ThermoFisher Scientific / Cat# A-32727 RRID:AB_2633276<br>Goat anti-mouse Alexa 568 / ThermoFisher Scientific / Cat# A-11004 RRID:AB_2534072<br>Goat anti-rabbit Alexa 568 / ThermoFisher Scientific Cat# A-11011 RRID:AB_143157<br>Chicken anti-rat Alexa 488 / ThermoFisher Scientific / Cat# A-21470 RRID:AB_2535873<br>Goat anti-mouse HRP / Biorad / Cat# 1721011 RRID:AB_11125936<br>Goat anti-rabbit HRP / Biorad / Cat# 1706515 RRID:AB_11125142 |
|---|---|
| Validation | -There was no new antibody specifically generated for this study.<br><br>-Antibodies previously generated inhouse:<br>Rabbit anti-Pol-I (IFA + WB), validation available in Glover et al, 2016 (doi: 10.1073/pnas.1600344113)<br>Rabbit anti-VEX2 (IFA only), validation available in Faria et al, 2019 (doi: 10.1038/s41467-019-10823-8)<br><br>-Antibodies generated by other labs:<br>Rat anti-VSG-2 (IFA), validation available in Hoek and Cross, 1999 (doi.org/10.1006/expr.1998.4369)<br>Rabbit anti-VSG-2 (WB), validation available in Hoek and Cross, 1999 (doi.org/10.1006/expr.1998.4369)<br>Rabbit anti-VSG-6 (IFA & WB), validation available in Hoek and Cross, 1999 (doi.org/10.1006/expr.1998.4369)<br><br>-Commercial antibodies (validation available on manufacturer's website):<br>Mouse anti-Myc 9B11 (IFA) https://www.cellsignal.co.uk/ - Mouse mAb #2276<br>Mouse anti-Myc 4A6 (WB) https://www.merckmillipore.com/GB/en/product/Anti-Myc-Tag-Antibody-clone-4A6,MM_NF-05-724<br>Rabbit anti-GFP LifeTechnologies (IFA / WB) https://www.thermofisher.com/antibody/product/GFP-Antibody-Polyclonal/A-11122<br>Mouse monoclonal anti-HA (IFA / WB) https://www.sigmaaldrich.com/GB/en/product/sigma/h9658?srsltid=AfmBOorjpZabx94q6N6xnQaYergL4UG4cUGG8s14kA4_oJ5ZZ-KC7C7R<br>Mouse anti-EF1α CBP-KK1 (WB) https://www.merckmillipore.com/GB/en/product/Anti-EF1-Antibody-clone-CBPKK1,MM_NF-05-235?ReferrerURL=https%3A%2F%2Fwww.google.com%2F&bd=1&cid=BIOS-S-EPDF-1148-1107-RC<br>Mouse anti-EP procyclin (IFA) https://www.biocompare.com/9776-Antibodies/90525-Mouse-AntiParasite-Trypanosoma-bruceiprocyclin-EP-Monoclonal-antibody-Unconjugated-Clone-tbrp1247/ |

Goat anti-mouse Alexa 488 (IFA) https://www.thermofisher.com/antibody/product/Goat-anti-Mouse-IgG-H-L-Cross-Adsorbed-Secondary-Antibody-Polyclonal/A-11001
Goat anti-rabbit Alexa 488 (IFA) https://www.thermofisher.com/antibody/product/Goat-anti-Rabbit-IgG-H-L-Highly-Cross-Adsorbed-Secondary-Antibody-Polyclonal/A-11034
Goat anti-mouse Alexa555+ (IFA) https://www.thermofisher.com/antibody/product/Goat-anti-Mouse-IgG-H-L-Highly-Cross-Adsorbed-Secondary-Antibody-Polyclonal/A32727
Goat anti-mouse Alexa 568 (IFA) https://www.thermofisher.com/antibody/product/Goat-anti-Mouse-IgG-II-L-Cross-Adsorbed-Secondary-Antibody-Polyclonal/A-11004
Goat anti-rabbit Alexa 568 (IFA) https://www.thermofisher.com/antibody/product/Goat-anti-Rabbit-IgG-H-L-Cross-Adsorbed-Secondary-Antibody-Polyclonal/A-11011
Chicken anti-rat Alexa 488 (IFA) https://www.thermofisher.com/antibody/product/Chicken-anti-Rat-IgG-H-L-Cross-Adsorbed-Secondary-Antibody-Polyclonal/A-21470
Goat anti-mouse HRP (WB) https://www.bio-rad.com/en-jp/sku/1706516-goat-anti-mouse-igg-h-l-hrp-conjugate?ID=1706516
Goat anti-rabbit HRP (WB) https://www.bio-rad.com/en-us/sku/1706515-goat-anti-rabbit-igg-h-l-hrp-conjugate?ID=1706515

Additionally, the antibodies above were tested in Trypanosoma brucei cells or cell extracts using the appropriate controls to ensure signal specificity. For Western-Blot analysis, the antibodies were tested for their ability to generate a single band with the expected molecular weight; in the case of tagged cell lines, a non-tagged control was always included in all experiments. For IFA, in the case of the primary antibodies, these were first tested for their ability to generate a single band with the correct molecular weight following Western-Blot analysis; for tagged cell lines, a non-tagged control was always included in all IFA experiments. With exception of the mouse monoclonal anti-HA, all the antibodies above have been previously used in Faria et al, 2019 (doi: 10.1038/s41467-019-10823-8).

# Eukaryotic cell lines

Policy information about cell lines and Sex and Gender in Research

Cell line source(s)
All Trypanosoma brucei brucei Lister 427 cell lines used and generated in this study are described in the methods section. The original Trypanosoma brucei brucei Lister 427 was a kind gift by Prof George Cross (Rockefeller University). The 2T1 cell line (parental line for RNAi and overexpression mutants) was generated in Prof Horn's lab (Alsford et al, 2005; doi:10.1016/j.molbiopara.2005.08.009). The 2T1/T7/Cas9 cell line (parental line for CRISPR/Cas9 precision editing) was generated in Prof Horn's lab (Rico et al, 2018; http://doi.org/10.1038/s41598-018-26303-w). The T7/TetR/Cas9 cell line (parental line for CRISPR/Cas9 endogenous tagging) was generated in Prof Rudenko's lab (Beneke et al, 2017; https://doi.org/10.1098/rsos.170095). The PT1 cell line was a kind gift by Prof David Horn and Dr Sebastian Hutchinson.

Authentication
RNA-seq provided authentication.

Mycoplasma contamination
All parental lines used for subsequent genetic manipulation were Mycoplasma-free by PCR-based detection.

Commonly misidentified lines (See ICLAC register)
No commonly misidentified cell lines were used.

# Plants

Seed stocks
N/A

Novel plant genotypes
N/A

Authentication
N/A

