## [Peer Review File · Nature Microbiology]

Specialised RNA decay fine-tunes monogenic antigen expression in *Trypanosoma brucei*.

Corresponding Author: Dr Joana Faria

Version 0:

Reviewer comments:

Reviewer #1

(Remarks to the Author)

This manuscript presents an elegant and comprehensive study exploring the post-transcriptional mechanisms that regulate expression of *Trypanosoma brucei* Variant Surface Glycoprotein (VSG) expression site-associated genes (ESAGs). The authors identify three novel ESB-associated proteins—ESB2, ESB3, and ESAP1—and define a recruitment hierarchy dependent on VEX2. ESB2, which harbours a SMG6-like PIN domain, acts as a nuclease that downregulates ESAG transcripts in a stage-specific, nuclease-dependent manner. These findings define a new layer of spatial gene regulation within the ESB and broaden our understanding of monogenic expression control in kinetoplastids.

Through proximity proteomics, transcriptomics, super-resolution microscopy, and mutational analysis, the study elucidates the organisation and function of this nuclear subdomain and proposes an intriguing parallel with components of the nonsense-mediated decay (NMD) pathway.

This is an exciting and technically strong study with high relevance to the field. However, several points regarding data interpretation, biological context, and comparisons with known systems require clarification.

Major Points for Clarification

-The recruitment of ESB2 to the ESB is said to be dependent on ESAP1 and ESB3.

While loss of localisation is significant in RNAi conditions, the authors do not clearly show whether loss of ESB localisation is due to failed recruitment or reduced protein abundance.

Including quantification of protein levels and nuclear localisation efficiency together would help strengthen this claim. Alternatively, an IP experiment could distinguish localisation from abundance effects.

-The manuscript interprets the cytoplasmic localisation of the catalytically dead ESB2 mutant as a defect in nuclear import or partner interaction. However, the NLS-tagged mutant still reaches the nucleus in many cells but fails to accumulate at the ESB. This suggests that catalytic activity (or RNA binding) is required for ESB retention, not import. This alternative interpretation should be acknowledged and discussed. Additional data on whether the mutant retains RNA-binding capability would help clarify the mechanism.

-ESAGs are highly homologous, and transcriptomic analyses risk read cross-mapping between active and silent expression sites. The authors mention using high-stringency alignment, but further support—such as uniquely mapping read analysis or orthogonal validation (e.g., qPCR)—would strengthen the claim that silent ESAGs are also upregulated in ESB2/3/ESAP1 knockdowns.

-The manuscript's conclusion effectively highlights a novel RNA decay-based mechanism for regulating virulence gene expression. However, the parallels drawn to NMD—while structurally grounded—should be reframed to avoid implying functional conservation. A clearer distinction between NMD as a surveillance mechanism and ESB2's role in selective transcript tuning in *T. brucei* would help focus the impact of this discovery and avoid conflation with unrelated pathways. The PIN domain in ESB2 is structurally similar to SMG6, and VEX2/ESAP1 resemble NMD factors. However, these proteins act outside canonical NMD. The manuscript would benefit from softening this analogy (e.g., use "NMD-like" or "SMG6-related domain") to avoid overstating functional conservation.

The manuscript presents a novel and well-supported model for regulated RNA decay in *T. brucei*, with broad implications for antigenic variation and subnuclear compartmentalisation. The points above should be addressed to strengthen the interpretation of the data and clarify the context of the findings within broader RNA biology.

Recommendation: Minor Revision

Reviewer #2

(Remarks to the Author)

This work biochemically identifies a large number of new molecules associated with VEX2, a key molecular player in the regulation of antigenic variation in African trypanosomes. Characterization of three of these molecules - ESB2, ESB3 and ESAP1 - shows that they are specifically localized at the Expression Site body (ESB). This study also establishes the dependency of their localization and abundance. In contrast to VEX2, none of these proteins appear to regulate VSG levels. Focusing on the role of ESB2, the authors demonstrate that the localization and function of this protein are dependent on its RNA nuclease activity. Importantly, this activity is required for negative regulation of ESAG transcripts from the active bloodstream expression site.

This work represents a commendable and technically impressive advance in describing new components of the ESB and identifying, for the first time, ESB components involved in downregulating ESAG levels. The experiments are thoughtfully designed, carefully controlled and the conclusions well-supported by the data.

A few points merit clarification or further consideration:

1. How do authors reconcile the fact that VEX2 regulates VSG levels, while its associated proteins primarily affect ESAG expression? Could there be distinct VEX2 sub-complexes operating in different regions of the active bloodstream expression site? A discussion of this possibility would significantly enrich the manuscript.
2. Figure 2i implies that ESB2 downregulates all ESAGs throughout BES, yet Figure 2j shows that several ESAGs (ESAG8, ESAG4, ESAG2 or ESAG11 for example) are not significantly affected; could the authors clarify this discrepancy? Are these particular ESAGs similarly unaffected by depletion of ESB3 and ESAP1? Additionally, could there be conserved features within their UTRs that distinguish the affected ESAGs, potentially making them more susceptible to regulation by an RNA nuclease?
3. It is somewhat unexpected that the RFP reporter gene integrated near ESAG7 is not regulated by ESB2 (Figure 5f). Could the authors integrate the reporter near ESAG3 gene - a transcript markedly upregulated in the ESB2 mutant (Figure 2j) - to assess whether ESB2 regulation depend on genomic context or distance from telomere? If regulatory sequences are identified (as discussed in point 2), their functional significance could be directly tested using this reporter.
4. While the precise mechanism by which ESB2's nuclease activity regulates ESAG transcripts is beyond the scope of the present study, further insight would greatly enhance the impact of the work. The authors mention a plausible model in which ESB2 promotes RNA decay. Do ESAG transcripts exhibit altered stability following ESB2 depletion? Can the authors rule out changes in transcriptional initiation as an alternative explanation, thereby supporting a post-transcriptional regulatory role for ESB2?

Reviewer #3

(Remarks to the Author)

Lansink et al. report the discovery of a novel post-transcriptional regulatory mechanism that modulates expression of ESAGs in *Trypanosoma brucei*. Through proximity proteomics, functional genetics, and transcriptomics, they identify three previously uncharacterised proteins (ESB2, ESB3, ESAP1) that localise to the expression-site body (ESB) and negatively regulate ESAG transcript abundance. ESB2, in particular, harbours a SMG6-like PIN domain and acts as an RNA nuclease whose activity is required for function and nuclear localisation. The study provides significant mechanistic insight into how trypanosomes achieve the extreme differential expression of VSGs and ESAGs, a long-standing puzzle in the field of antigenic variation.

Highlights of the manuscript:

Identification of ESB2 as a functional nuclease that selectively destabilises ESAG transcripts is a substantial contribution, introducing a post-transcriptional decay mechanism as a critical layer of monogenic antigen expression control.

The combination of TurboID-based proximity labelling, genetic epistasis, and RNA-Seq is well executed and highly complementary.

The study rigorously contrasts bloodstream and procyclic forms, underlining the developmental specificity of the ESB-associated machinery.

The inferred dependency chain from VEX2 to ESAP1, ESB3 and ESB2 is logical and experimentally supported, integrating into existing knowledge of ESB function.

The SMG6-like domain in ESB2 hints at evolutionary convergence in specialised RNA decay mechanisms and invites parallels with NMD in metazoans.

Some more major concerns/questions:

1. Direct evidence of RNA cleavage:

While the data strongly support ESB2 as a negative regulator of ESAGs, direct biochemical demonstration of its nuclease activity (e.g. in vitro cleavage assay with ESAG-derived RNA) would considerably strengthen the central claim. If such assays were not technically feasible, this limitation should be acknowledged more explicitly in the Discussion.

2. Mechanistic specificity:

The authors convincingly show that ESB2 acts post-transcriptionally, but it remains unclear how specificity for ESAGs is achieved. Is it mediated via ESAG UTRs, RNA secondary structure, or protein cofactors such as ESB3? The RFP reporter assay suggests positional specificity rather than sequence-specific decay, but further clarification would be helpful, even as speculation.

3. Fitness cost vs. transcriptome effects:

ESB2 RNAi causes a pronounced fitness defect, while transcriptomic shifts (i.e., ESAG upregulation) are not as strong (~11-fold). Could the fitness cost reflect functions beyond ESAG regulation?

Minor Comments:

While the authors provide a generally thorough discussion, especially concerning VSG expression and nuclear architecture, the manuscript would benefit from deeper engagement with prior studies on UTR-mediated mRNA regulation in *T. brucei*, such as genome-wide UTR screens that suggested negative regulatory motifs in ESAGs. More generally, the authors could discuss how this new decay mechanism fits into broader post-transcriptional regulation paradigms in kinetoplastids (e.g., ZC3H11- or RBP10-mediated control).

Figure 3k ('Hierarchical recruitment model') is excellent; consider adding it to the graphical abstract.

A more detailed table or schematic comparing known ESB components (ESB1, VEX1/2, ESB2/3, ESAP1) with their molecular functions and interactions would be a useful reference.

Clarify whether the observed increase in ESAG expression in silent VSG-ESs reflects transcriptional leakage or increased mRNA stability.

In Extended Data Fig. 5, detection thresholds for ESB2/ESB3 protein levels in procyclics could be more clearly stated. The statistical methods for proteomics (limma with FDR correction) and RNA-Seq (biological replicates, stringent mapping) are appropriate and well-applied. However, the manuscript would benefit from explicitly reporting statistical tests for functional assays such as growth curves, RNAi rescue, and reporter expression. Including p-values or confidence intervals would improve interpretability, particularly for claims based on subtle transcriptomic shifts or partial rescue effects.

Conclusion:

This is a well-executed and conceptually important study that reveals a new facet of gene regulation in *T. brucei*, helping explain the long-standing paradox of ESAG/VSG co-transcription and differential expression. The data are rich, the conclusions are generally well supported, and the integration into the broader field of RNA regulation is thoughtful.

I support publication in *Nature Microbiology* pending minor clarifications and, if feasible, inclusion of direct biochemical evidence of ESB2's nuclease activity.

Reviewer #4

(Remarks to the Author)

The manuscript by Lansink et al. entitled "Specialised RNA decay fine-tunes monogenic antigen expression in African trypanosomes" describes a proximity labeling and quantitative mass spectrometry approach for identifying novel expression-site body (ESB) components in African trypanosomes. The manuscript also validates and mechanistically interrogates novel factors in the ESB. This work fills a key gap in knowledge, uses advanced proximity proteomics technology for localized interaction discovery, and does a good job following up on novel biology. Overall, I am excited by the work, though I have some concerns regarding the proximity labeling mass spectrometry (PL-MS) data and its representation. In addition, in some results sections imprecise language is used where exact phrasing would provide more clarity and be more impactful. While I think this work has a lot of promise, I would not recommend publication unless the following major and minor concerns are addressed.

Major Concerns:

1. I recommend adding more information to the results section regarding the PL-MS experiments. It is hard to determine some key details, and while I tried to intuit some of them, it was ambiguous and can lead to misunderstanding. It is important here to use precise language and avoid ambiguity. Specifically:
 - a. Please very clearly specify how ESB and SLAB components are determined. Are ESB hits from both VEX2 experiments combined, that are not found in VEX1 hits? Are these hits from all three PL-MS experiments? Are they just the N-term VEX2? The only thing referencing this I could find was on page 3 line 128-131 which seems to indicate VEX2 is for ESB and VEX1 is for SLAB, but this is not well defined.
 - b. Please indicate what controls are used to discriminate between hits and background proteins (for example, pg 4 line 142, "... were highly enriched in the 'plus biotin' samples"; enriched compared to what? Is this statistically significant, is it qualitative, what fold change, how was this determined?).
 - c. Include a small description of the steps involved in generating the hit list including a brief summary of the sample generation steps (e.g. biotinylation timing, control, bead enrichment), and overview of MS analysis and hit calling.
 - d. An overview figure as the first panel in Figure 1 would help immensely in understanding the experimental design, would help define key terminology (like ESB component vs SLAB), and would contextualize the results of the PL-MS experiments.
 - e. Make sure to define the dataset as a whole in this section. How many total proteins are identified from each bait, how many are significantly enriched compared to the control, what is the number that overlaps, etc... It's useful to provide some QC plots also (correlation/PCA plots, bar graphs with totals per replicate, etc...) that can go in supplement to demonstrate the

reproducibility and quality of the dataset. More detailed information of the proteomics data is needed.

f. In general, please be more precise when describing the results, try to avoid statements like "most of them were likely involved in..." and include exact numbers (eg 10/12 or 77% etc...). If statistical significance is indicated, report the values. If GO enrichment analysis highlights key pathways report the significance, fold enrichment, and components (would be good to include in a table if too many to list).

2. Figure 1 feels a little disjointed. It might be easier to reorganize the figures such that only proteomics data is represented in Figure 1. Having the first results section focus solely on the proteomics data and moving the validation results from this figure to a new one, will help with the overall flow. My recommendation is to follow a more standard 1 main figure per results section approach, rather than distributing it across multiple results sections. The new figure 1 should include an overview of the experimental design, a summary of the PL-MS experiments, a summary of the "hits" (a Venn Diagram or other chosen graph to highlight the overlapping hits; and/or incorporating some of the information from Extended Data like the heatmap in ED Figure 2a). Each panel can lead the reader through each proteomics finding, starting large with the number of proteins identified, and ending smaller, with a discrete priority hit list of ESB and SLAB components. But make sure each panel is accurately and fully described. For example, the radial plots in Figure 1 are an interesting way to summarize the PL-MS data, and are described a little in the legend, but are not well described in the main text. This panel is briefly cited on page 3 line 138-141, but then not mentioned again. Ideally every figure panel should have a conclusion that is described in the main text (not just summarized). If you move current Figure 1f-j to a new figure it will also coincide nicely with the second results section which describes that data in better detail.

3. Please update the methods sections to incorporate information on the experimental design for the PL-MS experiments. This should include the number of biological and technical replicates that were performed for each condition and control, which controls were used for quantitative and statistical calculations, and ideally reference a metadata sheet with filenames tracing back to sample information such that another researcher could download the dataset and analyze on their own platform.

4. I recommend adding figures that demonstrate the same quality control/labeling verification for all three of the baits. This is regarding the Extended Data Fig. 1, particularly panels c-e. Ideally, there would be biotinylation + myc/bait co-localization for all three tagged constructs (n- and c-terminal tagged VEX2; plus VEX1). This should be with and without exogenous biotinylation given that the "no biotin" sample is the control. The microscopy images should be biotin labeled in a similar way the data was collected (30min biotinylation is useful to show, but since the proteomics data is collected based on 18hrs of labeling, the images should be generated also at 18hrs). The 18h labeling and background control are key to interpreting the data, especially given that the 18h labeling, overnight binding, and on beads digest protocols used here can result in higher levels of background binding and background protein identification. If there are technical reasons for why this can't be done, it should be transparently addressed somewhere in the text. Furthermore, it appears in panel 1c that there is biotinylation in locations that are not adjacent to the VEX2-TurboID, I think this is the "No" labeled arrows, but this labeling is not addressed or explained in the figure legend or main text (that I can find). These labels should be explained in the legend, and the non-ESB biotinylation should be addressed somewhere in the text.

Minor Concerns:

1. Consider adding abbreviations commonly used in the proteomics field:

a. Mass Spectrometry (MS)

b. Proximity Labeling (PL); Proximity Labeling Mass Spectrometry (PL-MS)

2. The Supplemental Sheets contain coloring which is not described. If the coloring is important, please provide a legend or table of contents somewhere describing the colors. Additionally, it is now preferred to provide a table of contents/legend in proteomics datasets describing the column headings for each table such that readers unfamiliar with the chosen software can understand all the values. This is especially important in this case since Progenesis requires a software license to use.

3. Extended Data Fig 1j is not described in the legend.

Decision Letter:

20th August 2025

Dear Joana,

Thank you for your patience while your manuscript "Specialised RNA decay fine-tunes monogenic antigen expression in African trypanosomes." was under peer-review at Nature Microbiology, and while we recruited an additional referee to assess the proximity labeling mass spectrometry-related aspects of the study. It has now been seen by 4 referees, whose expertise and comments you will find at the end of this email. Although they find your work of some potential interest, they have raised a number of concerns that will need to be addressed before we can consider publication of the work in Nature Microbiology.

In particular, referee #1 lists several points regarding data interpretation, biological context, and comparisons with known systems that will require clarification. The referee also says the study does not clearly show whether loss of ESB localisation is due to failed recruitment or reduced protein abundance, and they suggest including quantification of protein levels and nuclear localisation efficiency together. Referee #1 also states that additional data on whether the ESB2 mutant retains RNA-binding capability would help clarify the mechanism, and that uniquely mapping read analysis or orthogonal validation (e.g., qPCR) would strengthen the claim that silent ESAGs are also upregulated in ESB2/3/ESAP1 knockdowns. Referee #2 suggests to integrate the RFP reporter near ESAG3 gene to assess whether ESB2 regulation depend on genomic context or distance from telomere. Furthermore, this referee says that further insight into the precise mechanism by which ESB2's nuclease activity

regulates ESAG transcripts would greatly enhance the impact of the work. Referee #3 says that while the data strongly support ESB2 as a negative regulator of ESAGs, direct biochemical demonstration of its nuclease activity (e.g. in vitro cleavage assay with ESAG-derived RNA) would considerably strengthen the central claim. The referee also says that it remains unclear how specificity for ESAGs is achieved (this could be discussed). Referee #4 has some concerns regarding the proximity labeling mass spectrometry data and its representation. The referee recommends adding more information to the results section regarding the PL-MS experiments (specify how ESB and SLAB components are determined, indicate what controls are used to discriminate between hits and background proteins, include a small description of the steps involved in generating the hit list including a brief summary of the sample generation steps, an overview figure as the first panel in Figure 1, define the dataset as a whole, be more precise when describing the results). Referee #4 also says the methods sections should be updated to incorporate information on the experimental design for the PL-MS experiments. The referee also recommends adding figures that demonstrate the same quality control/labeling verification for all three of the baits.

Should further experimental data allow you to address these criticisms, we would be happy to look at a revised manuscript.

Please include a data availability statement as a separate section after Methods but before references, under the heading "Data Availability". This section should inform readers about the availability of the data used to support the conclusions of your study. This information includes accession codes to public repositories (data banks for protein, DNA or RNA sequences, microarray, proteomics data etc...), references to source data published alongside the paper, unique identifiers such as URLs to data repository entries, or data set DOIs, and any other statement about data availability. At a minimum, you should include the following statement: "The data that support the findings of this study are available from the corresponding author upon request", mentioning any restrictions on availability. If DOIs are provided, we also strongly encourage including these in the Reference list (authors, title, publisher (repository name), identifier, year). For more guidance on how to write this section please see: <http://www.nature.com/authors/policies/data/data-availability-statements-data-citations.pdf>

* If you have not done so already we suggest that you begin to revise your manuscript so that it conforms to our Article format instructions at <http://www.nature.com/nmicrobiol/info/final-submission>. Refer also to any guidelines provided in this letter.

When submitting the revised version of your manuscript, please pay close attention to our [href="https://www.nature.com/nature-portfolio/editorial-policies/image-integrity">Digital Image Integrity Guidelines](https://www.nature.com/nature-portfolio/editorial-policies/image-integrity) and to the following points below:

EXTENDED DATA FIGURES

Link Redacted

Note: This url links to your confidential homepage and associated information about manuscripts you may have submitted or be

reviewing for us. If you wish to forward this e-mail to co-authors, please delete this link to your homepage first.

Nature Microbiology is committed to improving transparency in authorship. As part of our efforts in this direction, we are now requesting that all authors identified as 'corresponding author' on published papers create and link their Open Researcher and Contributor Identifier (ORCID) with their account on the Manuscript Tracking System (MTS), prior to acceptance. This applies to primary research papers only. ORCID helps the scientific community achieve unambiguous attribution of all scholarly contributions. You can create and link your ORCID from the home page of the MTS by clicking on 'Modify my Springer Nature account'. For more information please visit www.springernature.com/orcid.

If you wish to submit a suitably revised manuscript we would hope to receive it within 6 months. If you cannot send it within this time, please let us know.

Yours sincerely,

Reviewer Expertise:

Referee #1: Trypanosoma, omics

Referee #2: Trypanosomes, antigenic variation, microscopy

Referee #3: Super resolution microscopy

Referee #4: TurboID-mediated proximity labelling combined with quantitative mass spectrometry

Reviewer Comments:

Reviewer #1 (Remarks to the Author):

This manuscript presents an elegant and comprehensive study exploring the post-transcriptional mechanisms that regulate expression of *Trypanosoma brucei* Variant Surface Glycoprotein (VSG) expression site-associated genes (ESAGs). The authors identify three novel ESB-associated proteins—ESB2, ESB3, and ESAP1—and define a recruitment hierarchy dependent on VEX2. ESB2, which harbours a SMG6-like PIN domain, acts as a nuclease that downregulates ESAG transcripts in a stage-specific, nuclease-dependent manner. These findings define a new layer of spatial gene regulation within the ESB and broaden our understanding of monogenic expression control in kinetoplastids.

Through proximity proteomics, transcriptomics, super-resolution microscopy, and mutational analysis, the study elucidates the organisation and function of this nuclear subdomain and proposes an intriguing parallel with components of the nonsense-mediated decay (NMD) pathway.

This is an exciting and technically strong study with high relevance to the field. However, several points regarding data interpretation, biological context, and comparisons with known systems require clarification.

Major Points for Clarification

-The recruitment of ESB2 to the ESB is said to be dependent on ESAP1 and ESB3.

While loss of localisation is significant in RNAi conditions, the authors do not clearly show whether loss of ESB localisation is due to failed recruitment or reduced protein abundance.

Including quantification of protein levels and nuclear localisation efficiency together would help strengthen this claim.

Alternatively, an IP experiment could distinguish localisation from abundance effects.

-The manuscript interprets the cytoplasmic localisation of the catalytically dead ESB2 mutant as a defect in nuclear import or partner interaction. However, the NLS-tagged mutant still reaches the nucleus in many cells but fails to accumulate at the ESB. This suggests that catalytic activity (or RNA binding) is required for ESB retention, not import. This alternative interpretation should be acknowledged and discussed. Additional data on whether the mutant retains RNA-binding capability would help clarify the mechanism.

-ESAGs are highly homologous, and transcriptomic analyses risk read cross-mapping between active and silent expression sites. The authors mention using high-stringency alignment, but further support—such as uniquely mapping read analysis or orthogonal validation (e.g., qPCR)—would strengthen the claim that silent ESAGs are also upregulated in ESB2/3/ESAP1 knockdowns.

-The manuscript's conclusion effectively highlights a novel RNA decay-based mechanism for regulating virulence gene expression. However, the parallels drawn to NMD—while structurally grounded—should be reframed to avoid implying functional conservation. A clearer distinction between NMD as a surveillance mechanism and ESB2's role in selective transcript tuning in *T. brucei* would help focus the impact of this discovery and avoid conflation with unrelated pathways. The PIN domain in ESB2 is structurally similar to SMG6, and VEX2/ESAP1 resemble NMD factors. However, these proteins act outside canonical NMD. The manuscript would benefit from softening this analogy (e.g., use "NMD-like" or "SMG6-related domain") to avoid overstating functional conservation.

The manuscript presents a novel and well-supported model for regulated RNA decay in *T. brucei*, with broad implications for

antigenic variation and subnuclear compartmentalisation. The points above should be addressed to strengthen the interpretation of the data and clarify the context of the findings within broader RNA biology.

Recommendation: Minor Revision

Reviewer #2 (Remarks to the Author):

This work biochemically identifies a large number of new molecules associated with VEX2, a key molecular player in the regulation of antigenic variation in African trypanosomes. Characterization of three of these molecules - ESB2, ESB3 and ESAP1 - shows that they are specifically localized at the Expression Site body (ESB). This study also establishes the dependency of their localization and abundance. In contrast to VEX2, none of these proteins appear to regulate VSG levels. Focusing on the role of ESB2, the authors demonstrate that the localization and function of this protein are dependent on its RNA nuclease activity. Importantly, this activity is required for negative regulation of ESAG transcripts from the active bloodstream expression site.

This work represents a commendable and technically impressive advance in describing new components of the ESB and identifying, for the first time, ESB components involved in downregulating ESAG levels. The experiments are thoughtfully designed, carefully controlled and the conclusions well-supported by the data.

A few points merit clarification or further consideration:

1. How do authors reconcile the fact that VEX2 regulates VSG levels, while its associated proteins primarily affect ESAG expression? Could there be distinct VEX2 sub-complexes operating in different regions of the active bloodstream expression site? A discussion of this possibility would significantly enrich the manuscript.
2. Figure 2i implies that ESB2 downregulates all ESAGs throughout BES, yet Figure 2j shows that several ESAGs (ESAG8, ESAG4, ESAG2 or ESAG11 for example) are not significantly affected; could the authors clarify this discrepancy? Are these particular ESAGs similarly unaffected by depletion of ESB3 and ESAP1? Additionally, could there be conserved features within their UTRs that distinguish the affected ESAGs, potentially making them more susceptible to regulation by an RNA nuclease?
3. It is somewhat unexpected that the RFP reporter gene integrated near ESAG7 is not regulated by ESB2 (Figure 5f). Could the authors integrate the reporter near ESAG3 gene - a transcript markedly upregulated in the ESB2 mutant (Figure 2j) - to assess whether ESB2 regulation depends on genomic context or distance from telomere? If regulatory sequences are identified (as discussed in point 2), their functional significance could be directly tested using this reporter.
4. While the precise mechanism by which ESB2's nuclease activity regulates ESAG transcripts is beyond the scope of the present study, further insight would greatly enhance the impact of the work. The authors mention a plausible model in which ESB2 promotes RNA decay. Do ESAG transcripts exhibit altered stability following ESB2 depletion? Can the authors rule out changes in transcriptional initiation as an alternative explanation, thereby supporting a post-transcriptional regulatory role for ESB2?

Reviewer #3 (Remarks to the Author):

Lansink et al. report the discovery of a novel post-transcriptional regulatory mechanism that modulates expression of ESAGs in *Trypanosoma brucei*. Through proximity proteomics, functional genetics, and transcriptomics, they identify three previously uncharacterised proteins (ESB2, ESB3, ESAP1) that localise to the expression-site body (ESB) and negatively regulate ESAG transcript abundance. ESB2, in particular, harbours a SMG6-like PIN domain and acts as an RNA nuclease whose activity is required for function and nuclear localisation. The study provides significant mechanistic insight into how trypanosomes achieve the extreme differential expression of VSGs and ESAGs, a long-standing puzzle in the field of antigenic variation.

Highlights of the manuscript:

Identification of ESB2 as a functional nuclease that selectively destabilises ESAG transcripts is a substantial contribution, introducing a post-transcriptional decay mechanism as a critical layer of monogenic antigen expression control.

The combination of TurboID-based proximity labelling, genetic epistasis, and RNA-Seq is well executed and highly complementary.

The study rigorously contrasts bloodstream and procyclic forms, underlining the developmental specificity of the ESB-associated machinery.

The inferred dependency chain from VEX2 to ESAP1, ESB3 and ESB2 is logical and experimentally supported, integrating into existing knowledge of ESB function.

The SMG6-like domain in ESB2 hints at evolutionary convergence in specialised RNA decay mechanisms and invites parallels with NMD in metazoans.

Some more major concerns/questions:

1. Direct evidence of RNA cleavage:

While the data strongly support ESB2 as a negative regulator of ESAGs, direct biochemical demonstration of its nuclease activity (e.g. in vitro cleavage assay with ESAG-derived RNA) would considerably strengthen the central claim. If such assays were not technically feasible, this limitation should be acknowledged more explicitly in the Discussion.

2. Mechanistic specificity:

The authors convincingly show that ESB2 acts post-transcriptionally, but it remains unclear how specificity for ESAGs is achieved. Is it mediated via ESAG UTRs, RNA secondary structure, or protein cofactors such as ESB3? The RFP reporter assay suggests positional specificity rather than sequence-specific decay, but further clarification would be helpful, even as speculation.

3. Fitness cost vs. transcriptome effects:

ESB2 RNAi causes a pronounced fitness defect, while transcriptomic shifts (i.e., ESAG upregulation) are not as strong (~11-fold). Could the fitness cost reflect functions beyond ESAG regulation?

Minor Comments:

While the authors provide a generally thorough discussion, especially concerning VSG expression and nuclear architecture, the manuscript would benefit from deeper engagement with prior studies on UTR-mediated mRNA regulation in *T. brucei*, such as genome-wide UTR screens that suggested negative regulatory motifs in ESAGs. More generally, the authors could discuss how this new decay mechanism fits into broader post-transcriptional regulation paradigms in kinetoplastids (e.g., ZC3H11- or RBP10-mediated control).

Figure 3k ('Hierarchical recruitment model') is excellent; consider adding it to the graphical abstract.

A more detailed table or schematic comparing known ESB components (ESB1, VEX1/2, ESB2/3, ESAP1) with their molecular functions and interactions would be a useful reference.

Clarify whether the observed increase in ESAG expression in silent VSG-ESs reflects transcriptional leakage or increased mRNA stability.

In Extended Data Fig. 5, detection thresholds for ESB2/ESB3 protein levels in procyclics could be more clearly stated. The statistical methods for proteomics (limma with FDR correction) and RNA-Seq (biological replicates, stringent mapping) are appropriate and well-applied. However, the manuscript would benefit from explicitly reporting statistical tests for functional assays such as growth curves, RNAi rescue, and reporter expression. Including p-values or confidence intervals would improve interpretability, particularly for claims based on subtle transcriptomic shifts or partial rescue effects.

Conclusion:

This is a well-executed and conceptually important study that reveals a new facet of gene regulation in *T. brucei*, helping explain the long-standing paradox of ESAG/VSG co-transcription and differential expression. The data are rich, the conclusions are generally well supported, and the integration into the broader field of RNA regulation is thoughtful.

I support publication in *Nature Microbiology* pending minor clarifications and, if feasible, inclusion of direct biochemical evidence of ESB2's nuclease activity.

Reviewer #4 (Remarks to the Author):

The manuscript by Lansink et. al. entitled "Specialised RNA decay fine-tunes monogenic antigen expression in African trypanosomes" describes a proximity labeling and quantitative mass spectrometry approach for identifying novel expression-site body (ESB) components in African trypanosomes. The manuscript also validates and mechanistically interrogates novel factors in the ESB. This work fills a key gap in knowledge, uses advanced proximity proteomics technology for localized interaction discovery, and does a good job following up on novel biology. Overall, I am excited by the work, though I have some concerns regarding the proximity labeling mass spectrometry (PL-MS) data and its representation. In addition, in some results sections imprecise language is used where exact phrasing would provide more clarity and be more impactful. While I think this work has a lot of promise, I would not recommend publication unless the following major and minor concerns are addressed.

Major Concerns:

1. I recommend adding more information to the results section regarding the PL-MS experiments. It is hard to determine some key details, and while I tried to intuit some of them, it was ambiguous and can lead to misunderstanding. It is important here to use precise language and avoid ambiguity. Specifically:
 - a. Please very clearly specify how ESB and SLAB components are determined. Are ESB hits from both VEX2 experiments combined, that are not found in VEX1 hits? Are these hits from all three PL-MS experiments? Are they just the N-term VEX2? The only thing referencing this I could find was on page 3 line 128-131 which seems to indicate VEX2 is for ESB and VEX1 is for SLAB, but this is not well defined.
 - b. Please indicate what controls are used to discriminate between hits and background proteins (for example, pg 4 line 142, "... were highly enriched in the 'plus biotin' samples"; enriched compared to what? Is this statistically significant, is it qualitative, what fold change, how was this determined?).

- c. Include a small description of the steps involved in generating the hit list including a brief summary of the sample generation steps (e.g. biotinylation timing, control, bead enrichment), and overview of MS analysis and hit calling.
- d. An overview figure as the first panel in Figure 1 would help immensely in understanding the experimental design, would help define key terminology (like ESB component vs SLAB), and would contextualize the results of the PL-MS experiments.
- e. Make sure to define the dataset as a whole in this section. How many total proteins are identified from each bait, how many are significantly enriched compared to the control, what is the number that overlaps, etc... It's useful to provide some QC plots also (correlation/PCA plots, bar graphs with totals per replicate, etc...) that can go in supplement to demonstrate the reproducibility and quality of the dataset. More detailed information of the proteomics data is needed.
- f. In general, please be more precise when describing the results, try to avoid statements like "most of them were likely involved in..." and include exact numbers (eg 10/12 or 77% etc...). If statistical significance is indicated, report the values. If GO enrichment analysis highlights key pathways report the significance, fold enrichment, and components (would be good to include in a table if too many to list).

2. Figure 1 feels a little disjointed. It might be easier to reorganize the figures such that only proteomics data is represented in Figure 1. Having the first results section focus solely on the proteomics data and moving the validation results from this figure to a new one, will help with the overall flow. My recommendation is to follow a more standard 1 main figure per results section approach, rather than distributing it across multiple results sections. The new figure 1 should include an overview of the experimental design, a summary of the PL-MS experiments, a summary of the "hits" (a Venn Diagram or other chosen graph to highlight the overlapping hits; and/or incorporating some of the information from Extended Data like the heatmap in ED Figure 2a). Each panel can lead the reader through each proteomics finding, starting large with the number of proteins identified, and ending smaller, with a discrete priority hit list of ESB and SLAB components. But make sure each panel is accurately and fully described. For example, the radial plots in Figure 1 are an interesting way to summarize the PL-MS data, and are described a little in the legend, but are not well described in the main text. This panel is briefly cited on page 3 line 138-141, but then not mentioned again. Ideally every figure panel should have a conclusion that is described in the main text (not just summarized). If you move current Figure 1f-j to a new figure it will also coincide nicely with the second results section which describes that data in better detail.

3. Please update the methods sections to incorporate information on the experimental design for the PL-MS experiments. This should include the number of biological and technical replicates that were performed for each condition and control, which controls were used for quantitative and statistical calculations, and ideally reference a metadata sheet with filenames tracing back to sample information such that another researcher could download the dataset and analyze on their own platform.

4. I recommend adding figures that demonstrate the same quality control/labeling verification for all three of the baits. This is regarding the Extended Data Fig. 1, particularly panels c-e. Ideally, there would be biotinylation + myc/bait co-localization for all three tagged constructs (n- and c-terminal tagged VEX2; plus VEX1). This should be with and without exogenous biotinylation given that the "no biotin" sample is the control. The microscopy images should be biotin labeled in a similar way the data was collected (30min biotinylation is useful to show, but since the proteomics data is collected based on 18hrs of labeling, the images should be generated also at 18hrs). The 18h labeling and background control are key to interpreting the data, especially given that the 18h labeling, overnight binding, and on beads digest protocols used here can result in higher levels of background binding and background protein identification. If there are technical reasons for why this can't be done, it should be transparently addressed somewhere in the text. Furthermore, it appears in panel 1c that there is biotinylation in locations that are not adjacent to the VEX2-TurboID, I think this is the "No" labeled arrows, but this labeling is not addressed or explained in the figure legend or main text (that I can find). These labels should be explained in the legend, and the non-ESB biotinylation should be addressed somewhere in the text.

Minor Concerns:

1. Consider adding abbreviations commonly used in the proteomics field:
 - a. Mass Spectrometry (MS)
 - b. Proximity Labeling (PL); Proximity Labeling Mass Spectrometry (PL-MS)
2. The Supplemental Sheets contain coloring which is not described. If the coloring is important, please provide a legend or table of contents somewhere describing the colors. Additionally, it is now preferred to provide a table of contents/legend in proteomics datasets describing the column headings for each table such that readers unfamiliar with the chosen software can understand all the values. This is especially important in this case since Progenesis requires a software license to use.
3. Extended Data Fig 1j is not described in the legend.

Version 1:

Reviewer comments:

Reviewer #2

(Remarks to the Author)

The authors have clearly addressed my previous reservations.

1. Although RNA stability could not be reliably assayed (Figure 6d-e), the new analysis presented in Figure 3f and the additional data shown in Figure 6f-g provide compelling evidence that ESB2 acts via a post-transcriptional mechanism.

2. The new in vitro data shown in Figure 5c are also an excellent addition to the revised manuscript, as they demonstrate that ESB2 functions as an endonuclease. For this specific panel (urea gels), it would be helpful for the general reader if the bands corresponding to “undigested RNA” (upper band) and “cleaved RNAs” (lower bands) were clearly labeled on the panel itself, or at least described explicitly in the figure legend.

Reviewer #3

(Remarks to the Author)

My major concerns have been satisfactorily addressed. In particular, the inclusion of new in vitro biochemical data supporting intrinsic RNA endonuclease activity of ESB2 substantially strengthens the central mechanistic claim.

Reviewer #4

(Remarks to the Author)

The authors have done a great job updating the manuscript, especially in describing the proteomics datasets. I appreciate the efforts taken to address my comments and the new version reads very well. All major and minor comments were addressed, and I have only found a few very minor details to address in the final version. I would recommend the manuscript for publication.

Some minor details/typos/edits:

1. Some abbreviations were not defined the first time they are introduced (or I might have missed them, apologies if so). Abbreviations: SL-array, PCF, BSFs (bloodstream forms is defined in the extended data fig 6 legend; it would be good to also define in the main text/methods).
2. This is a bit in the weeds, but in the Fig1 panel a schematic lower right image “on bead digest” region, the peptides still have biotin on them, this is a super technical detail, but with on beads digest, if there is no avidin elution used (which methods seem to indicate no avidin elution), then the peptides with biotin will remain stuck to the beads, and only the peptides without biotin will digest/elute off into the supernatant. Again, this is a super minor technical detail (and I like the figure as it is now), but if easily edited, it would be more accurate to have no biotin in that panel.

Decision Letter:

Our ref: NMICROBIOL-25051882A

22nd January 2026

Dear Joana,

Thank you for submitting your revised manuscript "Specialised RNA decay fine-tunes monogenic antigen expression in African trypanosomes." (NMICROBIOL-25051882A). It has now been seen by the original referees and their comments are below. The reviewers find that the paper has improved in revision, and therefore we'll be happy in principle to publish it in Nature Microbiology, pending minor revisions to satisfy the referees' final requests and to comply with our editorial and formatting guidelines.

Thank you again for your interest in Nature Microbiology. Please do not hesitate to contact me if you have any questions.

Sincerely,

Reviewer #2 (Remarks to the Author):

The authors have clearly addressed my previous reservations.

1. Although RNA stability could not be reliably assayed (Figure 6d–e), the new analysis presented in Figure 3f and the additional data shown in Figure 6f–g provide compelling evidence that ESB2 acts via a post-transcriptional mechanism.
2. The new in vitro data shown in Figure 5c are also an excellent addition to the revised manuscript, as they demonstrate that ESB2 functions as an endonuclease. For this specific panel (urea gels), it would be helpful for the general reader if the bands corresponding to “undigested RNA” (upper band) and “cleaved RNAs” (lower bands) were clearly labeled on the panel itself, or at least described explicitly in the figure legend.

Reviewer #3 (Remarks to the Author):

My major concerns have been satisfactorily addressed. In particular, the inclusion of new in vitro biochemical data supporting intrinsic RNA endonuclease activity of ESB2 substantially strengthens the central mechanistic claim.

Reviewer #4 (Remarks to the Author):

The authors have done a great job updating the manuscript, especially in describing the proteomics datasets. I appreciate the efforts taken to address my comments and the new version reads very well. All major and minor comments were addressed, and I have only found a few very minor details to address in the final version. I would recommend the manuscript for publication.

Some minor details/typos/edits:

1. Some abbreviations were not defined the first time they are introduced (or I might have missed them, apologies if so).

Abbreviations: SL-array, PCF, BSFs (bloodstream forms is defined in the extended data fig 6 legend; it would be good to also define in the main text/methods).

2. This is a bit in the weeds, but in the Fig1 panel a schematic lower right image "on bead digest" region, the peptides still have biotin on them, this is a super technical detail, but with on beads digest, if there is no avidin elution used (which methods seem to indicate no avidin elution), then the peptides with biotin will remain stuck to the beads, and only the peptides without biotin will digest/elute off into the supernatant. Again, this is a super minor technical detail (and I like the figure as it is now), but if easily edited, it would be more accurate to have no biotin in that panel.

Version 2:

Decision Letter:

10th February 2026

Dear Joana,

I am pleased to accept your Article "Specialised RNA decay fine-tunes monogenic antigen expression in *Trypanosoma brucei*." for publication in Nature Microbiology. Thank you for having chosen to submit your work to us and many congratulations.

You may wish to make your media relations office aware of your accepted publication, in case they consider it appropriate to organize some internal or external publicity. Once your paper has been scheduled you will receive an email confirming the publication details. This is normally 3-4 working days in advance of publication. If you need additional notice of the date and time of publication, please let the production team know when you receive the proof of your article to ensure there is sufficient time to coordinate. Further information on our embargo policies can be found here:

<https://www.nature.com/authors/policies/embargo.html>

Authors may need to take specific actions to achieve compliance with funder and institutional open access mandates. If your research is supported by a funder that requires immediate open access (e.g. according to [a href="https://www.springernature.com/gp/open-science/plan-s-compliance"> Plan S principles](https://www.springernature.com/gp/open-science/plan-s-compliance) or the [a href="https://www.springernature.com/gp/open-science/us-federal-agency-compliance"> NIH public access policy](https://www.springernature.com/gp/open-science/us-federal-agency-compliance)) then you should select the gold OA route, and we will direct you to the compliant route where possible. Because authors warrant under our subscription licensing terms that they haven't committed to licensing any version of their article under a licence inconsistent with the terms of our agreement – including the applicable embargo period – publication under the subscription model isn't suitable for authors whose funders require no embargo.

Congratulations once again and I look forward to seeing the article published.

With kind regards,

P.S. Click on the following link if you would like to recommend Nature Microbiology to your librarian
<http://www.nature.com/subscriptions/recommend.html#forms>

** Visit the Springer Nature Editorial and Publishing website at http://editorial-jobs.springernature.com?utm_source=ejP_NMicro_email&utm_medium=ejP_NMicro_email&utm_campaign=ejP_NMicro for more information about our career opportunities. If you have any questions please click [here](mailto:editorial.publishing.jobs@springernature.com). **

Open Access This Peer Review File is licensed under a Creative Commons Attribution 4.0 International License, which permits use, sharing, adaptation, distribution and reproduction in any medium or format, as long as you give appropriate credit to the original author(s) and the source, provide a link to the Creative Commons license, and indicate if changes were made. In cases where reviewers are anonymous, credit should be given to 'Anonymous Referee' and the source.

"Specialised RNA decay fine-tunes monogenic antigen expression in African trypanosomes"

We are grateful for the opportunity to revise our work and for the Reviewers' encouraging feedback and constructive critique. In response, we have conducted comprehensive additional analyses—both experimental and bioinformatic—to address the points raised. We have significantly modified the manuscript and display items to incorporate these new data while adhering to the journal's guidelines.

We are pleased to resubmit a revised version (changes highlighted in red) alongside a point-by-point response (below). Key additions include:

1. PL-MS & Candidate Selection (Addressing Rev. 4):

- Microscopy data confirming the localisation and biotinylation activity of all TurboID protein fusions (**Extended Data Fig. 1c-g**).
- A comprehensive QC of our bioinformatic pipeline to ensure robust candidate selection (**Fig. 1a, e-j, Extended Data Fig. 1j-l, Supplementary File 1-5**).

2. ESB2 Function & Mechanism (Addressing Revs. 1-3):

- Enzymatic activity & interactions:** *In vitro* assays with a recombinant nuclease domain confirming ESB2's intrinsic endonuclease activity (**Fig. 5c, Extended Data Fig. 9a**) and co-immunoprecipitation assays verifying its interaction with ESB3 (**Fig. 2g, Fig. 4l/m**).
- Transcriptional vs. RNA stability dynamics:** Analysis of unprocessed transcripts and mRNA turnover assays (**Fig. 6e**) supporting a role for ESB2 in post-transcriptional regulation rather than transcription initiation.
- Spatial regulation / substrate specificity:** Inhibition studies linking ESB2 compartmentalisation to active transcription and splicing (**Fig. 6d**), alongside reporter assays demonstrating that regulation is spatially driven rather than sequence-dependent (**Fig. 6g, Extended Data Fig. 10d-i**).

We believe these additions have resulted in a more robust study, which we hope is now suitable for publication in *Nature Microbiology*.

Sincerely,
Joana Faria
On behalf of all authors

Reviewer #1

1. The recruitment of ESB2 to the ESB is said to be dependent on ESAP1 and ESB3. While loss of localisation is significant in RNAi conditions, the authors do not clearly show whether loss of ESB localisation is due to failed recruitment or reduced protein abundance. Including quantification of protein levels and nuclear localisation

efficiency together would help strengthen this claim. Alternatively, an IP experiment could distinguish localisation from abundance effects.

R1.1: Good point. To address it, we have performed co-immunoprecipitation experiments between ESB2 and ESB3 and now demonstrate a direct interaction between them (**Figure 4l–m**).

To ensure rigorous validation, we generated a homozygous tagged ESB2 cell line. Given that ESB2 depletion causes a significant fitness cost, the successful generation of this line—which displays no fitness defect—confirms that the tag does not disrupt protein function. Furthermore, as these proteins are of low abundance, this approach ensured maximal recovery efficiency. Characterisation of this cell line is now provided in **Figure 2g**. After confirming colocalisation with ESB3, we proceeded with the co-immunoprecipitation.

Manuscript Updates:

Lines 175-177 & 260–264: Added text includes “(...) *Notably, this direct interaction is consistent with a model whereby ESB2 mislocalisation upon ESB3 depletion is a consequence of failed recruitment, not reduced protein levels.*”

Lines 426–433: We expanded the discussion to address the potential significance of this interaction for ESB2 regulation.

2. The manuscript interprets the cytoplasmic localisation of the catalytically dead ESB2 mutant as a defect in nuclear import or partner interaction. However, the NLS-tagged mutant still reaches the nucleus in many cells but fails to accumulate at the ESB. This suggests that catalytic activity (or RNA binding) is required for ESB retention, not import. This alternative interpretation should be acknowledged and discussed. Additional data on whether the mutant retains RNA-binding capability would help clarify the mechanism.

R1.2: Good point, thank you, we modified the text to include this alternative explanation.

Manuscript Updates:

Lines 311-314: “(...) *ESB2’s ability to interact with a binding partner that either transports it into the nucleus through a piggyback mechanism or enables its retention at the ESB.*”

Lines 438-443: Expanded the Discussion to cover the factors governing ESB2 recruitment to / retention at the ESB, particularly in light of our new data showing that active transcription and splicing are required (**Fig. 6d**).

Please note that we attempted to address whether the mutant retained RNA binding capability experimentally but encountered significant expression challenges. While we successfully purified the wild-type domain to confirm nuclease activity (**Fig. 5c, Extended Data Fig. 9a**; please also consult **R3.1**), attempts to produce full-length ESB2 or the catalytically inactive domain were unfortunately unsuccessful. Importantly, even if the inactive mutant domain had been purified, it would likely offer limited insight, as RNA binding interfaces frequently extend beyond the nuclease domain. Optimisation of full-length expression is a priority for future work but is beyond the scope of the current manuscript.

3. ESAGs are highly homologous, and transcriptomic analyses risk read cross-mapping between active and silent expression sites. The authors mention using high-stringency alignment, but further support—such as uniquely mapping read analysis or orthogonal

validation (e.g., qPCR)—would strengthen the claim that silent ESAGs are also upregulated in ESB2/3/ESAP1 knockdowns.

R1.3: Our apologies, the way we described our analysis in the methods as ‘*high stringency mapping*’ was misleading. We **did** use uniquely mapping reads and have now clearly stated that (**Line 821**).

4. The manuscript’s conclusion effectively highlights a novel RNA decay-based mechanism for regulating virulence gene expression. However, the parallels drawn to NMD—while structurally grounded—should be reframed to avoid implying functional conservation. A clearer distinction between NMD as a surveillance mechanism and ESB2’s role in selective transcript tuning in *T. brucei* would help focus the impact of this discovery and avoid conflation with unrelated pathways. The PIN domain in ESB2 is structurally similar to SMG6, and VEX2/ESAP1 resemble NMD factors. However, these proteins act outside canonical NMD. The manuscript would benefit from softening this analogy (e.g., use “NMD-like” or “SMG6-related domain”) to avoid overstating functional conservation.

R1.4: We agree that while the structural parallels to NMD factors are compelling, it is crucial to distinguish between shared protein architecture and functional conservation; we modified the text accordingly.

Manuscript Updates:

Lines 282-287: “For ESB2, we found that the closest experimentally determined **structural relative** to be human SMG6 (...) Notably, **while these two proteins are not homologues**, the catalytic residues required for the nuclease activity are strictly conserved”

Lines 410-419: “(...) While VEX2, ESAP1, and ESB2 share structural features with NMD regulators (Extended Data Fig. 8d-j), they are not homologues. **Rather than participating in NMD surveillance**, which is dispensable in *T. brucei* [76-79], **these proteins function distinctively in selective transcript tuning.**”

Reviewer #2

1. How do authors reconcile the fact that VEX2 regulates VSG levels, while its associated proteins primarily affect ESAG expression? Could there be distinct VEX2 sub-complexes operating in different regions of the active bloodstream expression site? A discussion of this possibility would significantly enrich the manuscript.

R2.1: Good point. We added co-immunoprecipitation data between ESB2 and ESB3 (**Fig. 4l-m**, please refer to **R1.1**) and the following sentence to the Discussion (**Lines 401-404**): “The absence of this ‘module’ in previous VEX-affinity purifications [30,33] suggests its VEX2-dependent localisation is linked to a structural role, implying spatial proximity over stable interaction.

However, future work should clarify if distinct VEX2 sub-complexes operate along the active-VSG-ES.”

2. Figure 2i implies that ESB2 downregulates all ESAGs throughout BES, yet Figure 2j shows that several ESAGs (ESAG8, ESAG4, ESAG2 or ESAG11 for example) are not significantly affected; could the authors clarify this discrepancy? Are these particular ESAGs similarly unaffected by depletion of ESB3 and ESAP1? Additionally, could there

be conserved features within their UTRs that distinguish the affected ESAGs, potentially making them more susceptible to regulation by an RNA nuclease?

R2.2: Thank you for this observation. We reviewed the data and realised that the presentation of Figs. 2i and 2j (now **Figs. 3e and 3f**) was misleading. Specifically, the use of RPKMs is not appropriate for comparisons between genes with differing mappability and tends to mask expression changes in transcripts with lower abundance. To address this, we have recalculated the data as fold change (RNAi vs. Parental, **Figure 3f**), based on CDS features. This accurately captures the behaviour of the five independent biological replicates used in the analysis. Additionally, we reduced the bin size in **Figure 3e** to provide a higher-resolution representation.

While this clarifies the general impact on *ESAG* regulation, some heterogeneity persists, albeit less drastically than the original depiction would imply. We investigated potential drivers—including distance to promoter / telomere, m6A modification status, and sequence motifs—but found no significant correlations (**Extended Data Fig. 10e–i**). We also updated the Discussion (**Lines 434–437**) to suggest that intrinsic factors, such as RNA secondary structure, may influence cleavage susceptibility. We address sequence-specificity and the contribution of UTR sequences in the point below.

Finally, we note that *ESB3* and *ESAP1* depletion resulted in similar gene expression changes (**Figure 3k**), which prompted the co-dependency experiments in **Figure 4**.

3. It is somewhat unexpected that the RFP reporter gene integrated near *ESAG7* is not regulated by *ESB2* (**Figure 5f**). Could the authors integrate the reporter near *ESAG3* gene - a transcript markedly upregulated in the *ESB2* mutant (**Figure 2j**) - to assess whether *ESB2* regulation depend on genomic context or distance from telomere? If regulatory sequences are identified (as discussed in point 2), their functional significance could be directly tested using this reporter.

R2.3: We sincerely thank the Reviewer for this insightful suggestion, which prompted an experiment that has significantly clarified the regulatory mechanism. We analysed two distinct reporter constructs expressing *RFP::PAC*, placed between *ESAG8* and *ESAG3* flanked by either *tubulin* or *ESAG3* UTRs. In both cases, *ESB2* depletion resulted in the upregulation of *RFP* mRNA, mirroring the behaviour of endogenous *ESAG3*. **This demonstrates that the regulation occurs independently of both coding and UTR sequences and therefore appears to be largely spatially regulated (Figure 6g; Extended Data Fig. 10d).**

Moreover, we performed MEME analysis on ESAGs (CDSs and UTRs from the active VSG-ES). The detected motifs appeared as extended, low-complexity blocks (>35 bp), which likely reflect the repetitive nature and shared ancestry of this gene family or perhaps structural features. To ensure the robustness of our UTR analysis, we compared sequences predicted by the Tinti & Horn approach (PMID: 40735494, **Extended Data Fig. 10e**) against those predicted by UTRme (PMID: 30619487) and 3'UTR sequences benchmarked using unpublished Oxford Nanopore long-read RNA-Seq data that we generated. We observed no significant differences in the outcome; ultimately, we identified no discrete consensus sequences indicative of specific nuclease recognition. While we acknowledge the inherent limitations of MEME analysis, this lack of sequence specificity is consistent with our reporter assay results. Some of these technical details were omitted from the main text to adhere to length constraints and maintain accessibility for a broad audience.

Manuscript Updates: We have updated the Results section to detail these findings (Lines 353-369) and expanded the Discussion to integrate these insights along with the points raised in R2.4 (Lines 420-426).

4. While the precise mechanism by which ESB2's nuclease activity regulates ESAG transcripts is beyond the scope of the present study, further insight would greatly enhance the impact of the work. The authors mention a plausible model in which ESB2 promotes RNA decay. Do ESAG transcripts exhibit altered stability following ESB2 depletion? Can the authors rule out changes in transcriptional initiation as an alternative explanation, thereby supporting a post-transcriptional regulatory role for ESB2?

R2.4: We agree that distinguishing transcriptional from post-transcriptional effects is critical. Accordingly, we provide multiple lines of evidence supporting that ESB2 regulates mRNA stability rather than transcriptional initiation:

1. We have now demonstrated that ESB2 is a functional endonuclease *in vitro* (Fig. 5c, Extended Data Fig. 9a). Biologically, it is highly improbable that an enzyme with RNA-cleavage activity would function to increase transcriptional initiation. We note that we could not accurately quantify ESB2's contribution to ESAG mRNA stability as transcription / splicing inhibition causes rapid ESB2 dispersal from the ESB (Fig. 6d-e; Lines 346-352).

2. ESAGs are upregulated while the co-transcribed downstream VSG remains unchanged. If initiation was increased at the promoter, the entire polycistronic unit—including the VSG—should be upregulated.

3. We analysed intergenic reads at the active VSG-ES as a proxy for unprocessed precursor transcripts. If transcription initiation had increased, we would expect a concurrent rise in intergenic reads; however, we observed no such increase upon ESB2 depletion (Figure R1 below). This stands in contrast to the significant accumulation of mature mRNA (Figure 3d-k) and **supports a mechanism of post-transcriptional regulation.** We decided not to include this specific analysis in the main text because, although consistent with the model we propose: 1) PolyA-enrichment strategies are suboptimal for accurately quantifying unprocessed transcripts; 2) we believe the data currently in the paper to be sufficiently compelling. Furthermore, omitting these data allowed us to maintain a focused narrative within the required word limits.

Figure R1. Analysis of intergenic ‘precursor’ reads at the active VSG-ES before and after ESB2 RNAi (24 hours). No statistically significant difference was found between conditions. R, Biological Replicate. Results derived from short read Illumina sequencing (transcriptomic data presented in the paper) and long read Oxford Nanopore Sequencing (unpublished) are depicted on the left and right hand-side panels, respectively. Please note that given the uncertainty around some of the UTR annotations (see also **R2.3**), we were conservative on what we considered ‘intergenic’ regions.

Manuscript Updates (to consolidate the narrative):

Lines 221-224: “*The uncoupling of this upregulation from unchanged VSG levels argues against increased transcription initiation, pointing instead to specific post-transcriptional regulation, which we later explore.*”

Lines 344-352: “*These results, supported by in vitro data (Fig. 5c), show that ESB2 negatively regulates ESAG transcripts through nuclease-dependent RNA decay. Therefore, to assess the impact of ESB2 on ESAG mRNA stability (...)*”

Reviewer #3

Major concerns/questions:

1. Direct evidence of RNA cleavage:

While the data strongly support ESB2 as a negative regulator of ESAGs, direct biochemical demonstration of its nuclease activity (e.g. in vitro cleavage assay with ESAG-derived RNA) would considerably strengthen the central claim. If such assays were not technically feasible, this limitation should be acknowledged more explicitly in the Discussion.

R3.1: We appreciate this was a limitation in our original submission and are pleased to now be able to include *in vitro* biochemical data supporting the model we propose in the revised version. Due to challenges in expressing full-length ESB2 in *E. coli* or *L. tarentolae*, we focused on the nuclease domain, which we successfully purified from *E. coli* (**Extended Data Fig. 9a**) and confirmed intrinsic RNA endonuclease activity (**Fig. 5c**). Note that the domain was purified through fusion with a solubility tag (His-MBP) which was cleaved prior to biochemical assays. Although we attempted to purify a catalytically inactive mutant (triple Asp replacement) to be used as a negative control, technical limitations in achieving sufficient yield and purity could not be overcome within the revision period, as we would need to apply a combination of different expression systems, constructs and purification pipelines. However, we validated the results through a set of other rigorous controls, as the observed activity was 1) time- and concentration-dependent, 2) Mg²⁺-dependent and therefore inhibited by EDTA, 3) refractory to RNase inhibitors, ruling out contaminant activity. Please note that we utilised a generic RNA substrate here; we address substrate specificity in the point below.

Manuscript Updates: We have updated the Results section to detail these findings (**Lines 288-293**)

2. Mechanistic specificity:

The authors convincingly show that ESB2 acts post-transcriptionally, but it remains unclear how specificity for ESAGs is achieved. Is it mediated via ESAG UTRs, RNA

secondary structure, or protein cofactors such as ESB3? The RFP reporter assay suggests positional specificity rather than sequence-specific decay, but further clarification would be helpful, even as speculation.

R3.2: We agree and have significantly improved this part of the manuscript. Please kindly refer to **R2.3 and R2.4**, as well as **Lines 420-450** of the Discussion.

3. Fitness cost vs. transcriptome effects:

ESB2 RNAi causes a pronounced fitness defect, while transcriptomic shifts (i.e., ESAG upregulation) are not as strong (~11-fold). Could the fitness cost reflect functions beyond ESAG regulation?

R3.3: Regarding specificity, the strict spatial confinement of ESB2 to the ESB argues against a broad role in global RNA regulation.

Regarding the magnitude of the defect, we argue that an ~11-fold upregulation can be sufficient to explain the fitness cost due to several compounding factors:

-Absolute mRNA burden: ‘active’ ESAGs are generally abundant transcripts. An 11-fold increase is a significant quantity of additional transcripts. Note that the reported abundance of specific ESAG mRNAs is likely an underestimate given the necessity of using uniquely mapped reads, precluding absolute quantification.

-Biosynthetic protein burden: mRNA and protein levels often display non-linear correlations; the resulting protein load may be even higher.

-Stoichiometry: Several ESAGs encode transporters or receptors—protein classes that are highly sensitive to dosage. Even minor stoichiometric imbalances in these complexes can be toxic or disrupt function.

-Underestimation: technically, the measured 11-fold increase might be an underestimation as transcriptomic analysis is inherently biased towards the surviving cells (*‘survivorship bias’*).

Manuscript Update: given the word limit and the fact that this point remains speculative without experimental validation, we opted to add a brief statement to the Discussion.

Lines 447-450: *“Furthermore, we attribute the substantial fitness cost following ESB2 depletion to the consequences of the ~11-fold ESAG upregulation, as it is plausible the resulting protein accumulation compromises viability through biosynthetic burden or by disrupting the stoichiometry of essential receptors and transporters.”*

Minor Comments:

1. While the authors provide a generally thorough discussion, especially concerning VSG expression and nuclear architecture, the manuscript would benefit from deeper engagement with prior studies on UTR-mediated mRNA regulation in *T. brucei*, such as genome-wide UTR screens that suggested negative regulatory motifs in ESAGs. More generally, the authors could discuss how this new decay mechanism fits into broader post-transcriptional regulation paradigms in kinetoplastids (e.g., ZC3H11- or RBP10-mediated control).

R3.1 (minor): Good point. We wish we could expand it further but given the strict word limit, we could only add a small paragraph to the Discussion.

Manuscript Update:

Lines 451-457: *“In kinetoplastid parasites, the cellular proteome is sculpted almost exclusively by post-transcriptional mechanisms [85]; indeed, a recent screen identified potentially destabilising motifs within ESAG 3' UTRs [86]. We expand this framework by establishing ESB2 as a distinct upstream nuclear checkpoint. Unlike factors that broadly regulate mRNA cohorts (e.g., RBP10, ZC3H11 [87,88]), ESB2 acts as a site-specific ‘pre-filter’ for the active-VSG-ES, pre-emptively refining stoichiometry before transcripts engage the cytoplasmic machinery.”*

2. Figure 3k ('Hierarchical recruitment model') is excellent; consider adding it to the graphical abstract.

R3.2 (minor): Thank you, we incorporated it into the final model (**Fig. 6i**).

3. A more detailed table or schematic comparing known ESB components (ESB1, VEX1/2, ESB2/3, ESAP1) with their molecular functions and interactions would be a useful reference.

R3.3 (minor): We improved our final model to illustrate the distinct functions and co-dependencies between different ESB factors (**Fig. 6j-i**) and provide a schematic representation of the gene expression phenotypes resulting from individual knockdowns as well as ESB1 / VEX domain analysis (**Extended Data Fig. 10j**). For a more comprehensive overview, we refer readers to a recent review (PMID: 40780972; **Line 400**)

4. Clarify whether the observed increase in ESAG expression in silent VSG-ESs reflects transcriptional leakage or increased mRNA stability.

R3.4 (minor): Manuscript Update:

Lines 219-226 – *“This suggests that microscopically undetectable pools of ESB2 might operate at ‘silent’ VSG-ESs. (...) The uncoupling of this upregulation from unchanged VSG levels argues against increased transcription initiation, pointing instead to specific post-transcriptional regulation, which we later explore.”*

5. In Extended Data Fig. 5, detection thresholds for ESB2/ESB3 protein levels in procyclics could be more clearly stated.

R3.5 (minor): Thank you, it was an oversight. Detection thresholds are now indicated in the plot and clarified in the legend.

The statistical methods for proteomics (limma with FDR correction) and RNA-Seq (biological replicates, stringent mapping) are appropriate and well-applied. However, the manuscript would benefit from explicitly reporting statistical tests for functional assays such as growth curves, RNAi rescue, and reporter expression. Including p-values or confidence intervals would improve interpretability, particularly for claims based on subtle transcriptomic shifts or partial rescue effects.

We have improved that throughout the manuscript as much as we could, balancing the added detail with the required conciseness.

Reviewer #4

We appreciate the detailed feedback regarding the PL-MS data. To address these points, we have implemented substantial revisions to **Figure 1, Extended Data Figures 1–2, and the Supplementary Information (Sheets 1–5)**.

As the PL-MS screen served primarily as a discovery platform to enable the study's main focus—the **functional characterisation** of novel ESB factors—we have balanced the request for detail with the manuscript's length constraints by:

1. **Expanding the Methods section** to include detailed optimisation, QC, and hit selection criteria, ensuring technical rigor without disrupting the narrative flow of the Results.
2. **Providing a comprehensive Supplementary Data File (Sheets 1–5)** that includes all essential metrics (intensity values, peptide counts, and statistical analyses) to ensure full transparency and reproducibility.

Given the 4,500-word limit and the new mechanistic data added in response to Reviewers 1–3, this structure ensures the manuscript remains focused while still providing full access to the underlying proteomic datasets.

Major points:

1. I recommend adding more information to the results section regarding the PL-MS experiments. It is hard to determine some key details, and while I tried to intuit some of them, it was ambiguous and can lead to misunderstanding. It is important here to use precise language and avoid ambiguity. Specifically:
a. Please very clearly specify how ESB and SLAB components are determined. Are ESB hits from both VEX2 experiments combined, that are not found in VEX1 hits? Are these hits from all three PL-MS experiments? Are they just the N-term VEX2? The only thing referencing this I could find was on page 3 line 128-131 which seems to indicate VEX2 is for ESB and VEX1 is for SLAB, but this is not well defined.

R4.1a: Given the established spatial proximity between the ESB (VEX2) and SLAB (VEX1), we observed a cross-enrichment of their proximity proteomes. By combining the three PL-MS datasets, we identified and validated 19 'high priority' candidates as specific components of the ESB, SLAB, or NUFIP bodies.

To clarify these points in the manuscript, we:

1. Added a section entitled '*PL-MS hit selection and prioritisation*' to the Methods, which explains the process in detail. **Lines 677-720**

2. Edited the Supplementary File (sheets 1-3), now containing a colour code to indicate proteins enriched in the TurboID datasets compared with the parental 'plus biotin' control (cut off: $\text{LogFC} > 1.5$ & $\text{adj } p < 0.05$), known ESB / SLAB / NUFIP / Telomeric components, predicted contaminants and 'high priority' hits.

3. Added Venn diagrams (**Fig. 1h-j**) and modified the radial plots (**Fig. 1e-g**) to highlight the hits validated as ESB / SLAB / NUFIP components in the three PL-MS datasets.

4. Updated the Results section (*within the available word count*)

Lines 122-125: "We integrated the three TurboID datasets (VEX1, VEX2 N- and C-terminal) and identified 70 significantly enriched proteins in comparison with the parental control ($\text{Log}_2\text{FC} > 1.5$; $\text{adj } p < 0.05$ FDR 1%) (...)"

Lines 133-140: *“The 70 significantly enriched proteins were subsequently refined to a set of 45 candidates by excluding known ESB, SLAB, NUFIP components, telomere-binding and non-nuclear proteins (...) These were further refined to a final set of 19 prioritised candidates based on (...)”*

Lines 166-169: *“Notably, the reciprocal enrichment of ESB/SLAB components following VEX1 and VEX2 PL-MS profiling strongly supports the previously proposed VEX-mediated inter-chromosomal bridge linking these nuclear compartments”*

b. Please indicate what controls are used to discriminate between hits and background proteins (for example, pg 4 line 142, “...were highly enriched in the ‘plus biotin’ samples”; enriched compared to what? Is this statistically significant, is it qualitative, what fold change, how was this determined?).

R4.1b: As mentioned above, we added an extensive section on ‘PL-MS hit selection and prioritisation’ to the Methods (**Lines 677-720**), including the following statement: *“The TurboID ‘minus biotin’ control samples exhibited significant biotinylation background, likely resulting from biotin present in the culture medium. Consequently, to effectively identify more subtle biotinylation changes, we used the parental wild-type cell line (plus biotin) as our primary negative control.”*

While space constraints precluded a lengthy expansion in the Results section itself, we have ensured clarity by specifying the reference controls for the TurboID ‘plus biotin’ samples. We also updated the text to include specific Log2FC and adjusted *p*-values for these comparisons.

c. Include a small description of the steps involved in generating the hit list including a brief summary of the sample generation steps (e.g. biotinylation timing, control, bead enrichment), and overview of MS analysis and hit calling.

R4.1c: Now clarified in the results section: *“To ensure maximal recovery of biotinylated material, the labelling reaction was allowed to proceed for 18 hours, biotinylated material was then affinity purified using streptavidin coated beads and submitted to MS analysis”*-
Lines 118-121 Please see also some of the statements included in **R4.1a** and **R4.1b**.

d. An overview figure as the first panel in Figure 1 would help immensely in understanding the experimental design, would help define key terminology (like ESB component vs SLAB), and would contextualize the results of the PL-MS experiments.

R4.1d: This was a good suggestion, thank you. We included a cartoon – **Fig. 1a**. We also added key details to the legend: *“Schematic of proximity labelling mass spectrometry (PL-MS). VEX1 and VEX2, fused to the biotin ligase TurboID, serve as markers for the SLAB and ESB respectively. Proximal proteins were biotinylated, affinity purified using streptavidin and identified via MS. MS analysis showed shared enrichment of known SLAB, ESB, and NUFIP components in both VEX1 and VEX2 samples, suggesting dynamic interplay between these nuclear bodies. Protein candidates from combined VEX datasets were selected and prioritised as detailed in the methods section and subsequently endogenously tagged to confirm their localisation via colocalisation fluorescence microscopy with known protein markers.”* **Lines 1332-1340**

e. Make sure to define the dataset as a whole in this section. How many total proteins are identified from each bait, how many are significantly enriched compared to the

control, what is the number that overlaps, etc... It's useful to provide some QC plots also (correlation/PCA plots, bar graphs with totals per replicate, etc...) that can go in supplement to demonstrate the reproducibility and quality of the dataset. More detailed information of the proteomics data is needed.

R4.1e: Given space constraints, we have included the number of total detected proteins per bait in the Methods section ('**PL-MS hit selection and prioritisation**', Lines 677-678) rather than in the Results. As mentioned in **R4.1a**, we included Venn diagrams (Fig. 1h-j), reflecting both our hit selection / prioritisation approach and the overlap between the three TurboID datasets. Further, we added heatmaps to **Extended Data Fig. 1 (j-l)**, as they provide a direct visualisation of protein abundance differences between replicates and speak to the quality / reproducibility of our datasets for proteins called as significantly more abundant than in the parental 'plus biotin' control ($\text{Log}_2\text{FC} > 1.5$; $\text{adj } p < 0.05$ FDR 1%). Please note that, besides the detailed statistical analysis, raw intensity values and relative protein abundances for each individual replicate are also provided in **Supplementary Information file (sheets 1-3)**.

f. In general, please be more precise when describing the results, try to avoid statements like "most of them were likely involved in..." and include exact numbers (eg 10/12 or 77% etc...). If statistical significance is indicated, report the values. If GO enrichment analysis highlights key pathways report the significance, fold enrichment, and components (would be good to include in a table if too many to list).

R4.1f: As mentioned in **R4.1b**, we have added Log_2FC and $\text{adj } p$ values information throughout this section of the Results. We have been clearer about the exact number of proteins selected as hits, prioritised, validated, predicted to be involved in RNA processing, etc. We have also included an additional sheet to **Supplementary Information file (sheet 4)** containing a summary of the GO enrichment analysis.

2. Figure 1 feels a little disjointed. It might be easier to reorganize the figures such that only proteomics data is represented in Figure 1. Having the first results section focus solely on the proteomics data and moving the validation results from this figure to a new one, will help with the overall flow. My recommendation is to follow a more standard 1 main figure per results section approach, rather than distributing it across multiple results sections. The new figure 1 should include an overview of the experimental design, a summary of the PL-MS experiments, a summary of the "hits" (a Venn Diagram or other chosen graph to highlight the overlapping hits; and/or incorporating some of the information from Extended Data like the heatmap in ED Figure 2a). Each panel can lead the reader through each proteomics finding, starting large with the number of proteins identified, and ending smaller, with a discrete priority hit list of ESB and SLAB components. But make sure each panel is accurately and fully described. For example, the radial plots in Figure 1 are an interesting way to summarize the PL-MS data, and are described a little in the legend, but are not well described in the main text. This panel is briefly cited on page 3 line 138-141, but then not mentioned again. Ideally every figure panel should have a conclusion that is described in the main text (not just summarized). If you move current Figure 1f-j to a new figure it will also coincide nicely with the second results section which describes that data in better detail.

R4.2: Thank you for the suggestion. We have now split the original Fig. 1 into two figures. Current Fig 1 is solely focused on the PL-MS data, contains a cartoon with an overview of

the approach (addressed in **R4.1d**) and Venn diagrams (addressed in **R4.1a** and **R4.1e**). We also extensively modified the Results' section to improve clarity on both our experimental approach and our findings (**Lines 108-140**). Due to space limitations, we have ensured that the figure legend provides all necessary details to interpret the radial plots, rather than expanding the main text – we now highlighted the new ESB / SLAB / NUFIP components in the plots as well.

3. Please update the methods sections to incorporate information on the experimental design for the PL-MS experiments. This should include the number of biological and technical replicates that were performed for each condition and control, which controls were used for quantitative and statistical calculations, and ideally reference a metadata sheet with filenames tracing back to sample information such that another researcher could download the dataset and analyze on their own platform.

R4.3: We edited the methods section as follows:

Lines 579-580: “3xmyc::4xGS::TurboID::VEX2 (n=6), VEX2::TurboID::4xGS::3xmyc (n=4) and VEX1::TurboID::4xGS::3xmyc (n=4) (...)”.

Please refer to **R4.1b** as well.

As for the filenames, we added the following statement: “*The deposited filenames match the titles used in Supplementary Information - Sheets 1–3*” (**Line 941-942**).

4. I recommend adding figures that demonstrate the same quality control/labeling verification for all three of the baits. This is regarding the Extended Data Fig. 1, particularly panels c-e. Ideally, there would be biotinylation + myc/bait co-localization for all three tagged constructs (n- and c-terminal tagged VEX2; plus VEX1). This should be with and without exogenous biotinylation given that the “no biotin” sample is the control. The microscopy images should be biotin labeled in a similar way the data was collected (30min biotinylation is useful to show, but since the proteomics data is collected based on 18hrs of labeling, the images should be generated also at 18hrs). The 18h labeling and background control are key to interpreting the data, especially given that the 18h labeling, overnight binding, and on beads digest protocols used here can result in higher levels of background binding and background protein identification. If there are technical reasons for why this can't be done, it should be transparently addressed somewhere in the text. Furthermore, it appears in panel 1c that there is biotinylation in locations that are not adjacent to the VEX2-TurboID, I think this is the “No” labeled arrows, but this labeling is not addressed or explained in the figure legend or main text (that I can find). These labels should be explained in the legend, and the non-ESB biotinylation should be addressed somewhere in the text.

R4.4: As part of the development of our PL-MS pipeline, we tried different biotin concentrations (50, 150, 500 μ M) and incubation time (30 min, 1 h, 3 h, 6 h, 18 h). The readout was not only Western-blot and fluorescence microscopy, but we also conducted pilot MS experiments. For instance, biotinylation signals could be detected in the respective nuclear compartments at both 30 min and 18 h without significant additional nuclear background but the signal intensity in the compartments of interest was higher at 18 h. Additionally, the signal to noise in the pilot MS experiments was significantly higher at 18 h of incubation and therefore used in the subsequent experiments. Given the limit of both main and supplementary figures, we cannot include all the optimisation leading up to the final conditions used in the PL-MS experiments. However, **we did add:**

1. A paragraph to the Methods on *PL-MS optimisation and validation* lines 561-577
2. Colocalisation data for VEX1-TurboID / SLAB and VEX2-TurboID / ESB (Extended Data Fig 1c-e). Colocalisation data for TurboID-VEX2 / ESB is now in Fig. 1b.
3. Microscopy data for biotinylation patterns found in all 3 cell lines at 18 h post incubation with biotin, including parental lines (our primary background control as previously addressed R4.1b and R4.3) (Extended Data Fig 1g). We kept the data for TurboID-VEX2 and VEX1-TurboID biotinylation patterns obtained at 30 min (Fig. 1c-d, Extended Data Fig. 1f) as the images were obtained using super resolution microscopy, facilitating visualisation.
4. Clarification about the observed sub-nuclear biotinylation patterns (Lines 108-118)
5. Information on the biotinylation conditions & defined 'No' (nucleolus) in the relevant figure legends.

Minor points:

1. Consider adding abbreviations commonly used in the proteomics field:

- a. Mass Spectrometry (MS)

- b. Proximity Labeling (PL); Proximity Labeling Mass Spectrometry (PL-MS)

R4.1 (minor): We have adopted the suggested terminology.

2. The Supplemental Sheets contain coloring which is not described. If the coloring is important, please provide a legend or table of contents somewhere describing the colors. Additionally, it is now preferred to provide a table of contents/legend in proteomics datasets describing the column headings for each table such that readers unfamiliar with the chosen software can understand all the values. This is especially important in this case since Progenesis requires a software license to use.

R4.2 (minor): Apologies, we have now added a key. Please note that we have changed the colour code (R4.1a) but is now described at the top of the sheet.

3. Extended Data Fig 1j is not described in the legend.

R4.3 (minor): Thank you for noting, we have however removed that panel to accommodate the new heatmaps (R4.1e) and the additional microscopy data (R4.4).

"Specialised RNA decay fine-tunes monogenic antigen expression in *Trypanosoma brucei*"

Reviewer #2

Remarks to the Author: The authors have clearly addressed my previous reservations.

1. Although RNA stability could not be reliably assayed (Figure 6d–e), the new analysis presented in Figure 3f and the additional data shown in Figure 6f–g provide compelling evidence that ESB2 acts via a post-transcriptional mechanism.

We are pleased that the additional data provides compelling evidence that ESB2 acts via a post-transcriptional mechanism. We thank the Reviewer once again for helpful comments and suggestions.

2. The new in vitro data shown in Figure 5c are also an excellent addition to the revised manuscript, as they demonstrate that ESB2 functions as an endonuclease. For this specific panel (urea gels), it would be helpful for the general reader if the bands corresponding to “undigested RNA” (upper band) and “cleaved RNAs” (lower bands) were clearly labelled on the panel itself, or at least described explicitly in the figure legend.

Thank you. We have added labels to the panels depicting urea gels: upper band ‘intact’ RNA and lower band ‘cleaved’ RNA.

Reviewer #3:

Remarks to the Author: My major concerns have been satisfactorily addressed. In particular, the inclusion of new in vitro biochemical data supporting intrinsic RNA endonuclease activity of ESB2 substantially strengthens the central mechanistic claim.

We thank the Reviewer once again for helpful comments and suggestions.

Reviewer #4:

Remarks to the Author: The authors have done a great job updating the manuscript, especially in describing the proteomics datasets. I appreciate the efforts taken to address my comments and the new version reads very well. All major and minor comments were addressed, and I have only found a few very minor details to address in the final version. I would recommend the manuscript for publication.

We are pleased that the revised version has addressed all the points raised and we thank the Reviewer once again for helpful comments and suggestions.

Some minor details/typos/edits:

1. Some abbreviations were not defined the first time they are introduced (or I might have missed them, apologies if so). Abbreviations: SL-array, PCF, BSFs (bloodstream forms is defined in the extended data fig 6 legend; it would be good to also define in the main text/methods).

Thank you for noticing. We have now defined BSF (bloodstream forms) and PCF (procyclic forms) in the main text (**lines 161 and 170**) and methods (**line 679**). The Spliced Leader (SL) arrays are now more clearly defined in the main text (**lines 70-71**).

2. This is a bit in the weeds, but in the Fig1 panel a schematic lower right image “on bead digest” region, the peptides still have biotin on them, this is a super technical detail, but with on beads digest, if there is no avidin elution used (which methods seem to indicate no avidin elution), then the peptides with biotin will remain stuck to the beads, and only the peptides without biotin will digest/elute off into the supernatant. Again, this is a super minor technical detail (and I like the figure as it is now), but if easily edited, it would be more accurate to have no biotin in that panel.

Not at all, that is a good point – we have amended the panel. Thank you again for the thorough revision of our manuscript.